# NABench: Large-Scale Benchmarks of Nucleotide Foundation Models for Fitness Prediction

## Abstract

Nucleotide sequence variation can induce significant shifts in functional fitness. Recent nucleotide foundation models promise to predict such fitness effects directly from sequence, yet heterogeneous datasets and inconsistent preprocessing make it difficult to compare methods fairly across DNA and RNA families. We introduce NABench, a large-scale, systematic benchmark for nucleic acid fitness prediction. NABench aggregates 162 high-throughput assays and curates 2.6 million mutated sequences spanning diverse DNA and RNA families, with standardized splits and rich metadata. We show that NABench surpasses prior nucleotide fitness benchmarks in scale, diversity, and data quality. Under a unified evaluation suite, we rigorously assess 29 representative sequence models across zero-shot, few-shot prediction, transfer learning, and supervised settings. The results quantify performance heterogeneity across tasks and nucleic-acid types, demonstrating clear strengths and failure modes for different modeling choices and establishing strong, reproducible baselines. We release NABench to catalyze progress in nucleic-acid modeling and to support downstream applications in nucleotide molecular design, synthetic biology, and biochemistry. Our code is available at https://anonymous.4open.science/r/NABench-20CB.

## 1 Introduction

Sequence variation in nucleic acids, such as single-nucleotide substitutions and small insertions or deletions, can profoundly alter molecular structure and function, thereby influencing fitness (Sanjuán et al., 2004; Orr, 2009; Cuevas et al., 2012; Huang et al., 2021). Accurately modeling this complex, high-dimensional sequence–fitness landscape is essential for identifying pathogenic variants (Riessel­man et al., 2018; Agarwal et al., 2023; Pucci et al., 2024; Ito et al., 2025) and for guiding the rational engineering of DNA and RNA elements (Ward et al., 2023; Li et al., 2024b; Ma et al., 2025), with broad implications for synthetic biology and therapeutic development (Yi & Dean, 2019; Wagner, 2023; Gosai et al., 2024).

Recently, nucleotide foundation models (NFMs) (He et al., 2025; Dalla-Torre et al., 2025; Nguyen, 2025; Wu et al., 2025) have introduced a new paradigm for nucleic acid fitness prediction. Pre-trained in a self-supervised manner on massive collections of DNA and RNA sequences, NFMs learn comprehensive and transferable representations, which capture complex long-range dependencies and subtle evolutionary signals that are often overlooked by traditional methods relying on local features or hand-crafted descriptors. Consequently, these advances have enabled more accurate prediction of fitness directly from raw nucleotide sequences.

However, evaluating the effectiveness of these models remains challenging, which in turn constrains the development and application of NFMs. Firstly, existing evaluations are typically conducted on diverse and contrived datasets, making direct and fair comparison difficult. Moreover, prior studies (Arora et al., 2025; Ren et al., 2024) have shown that model performance can vary substantially across nucleic acid families and prediction tasks. This heterogeneity hinders researchers from accurately assessing the true capabilities and failure modes of different architectures, and prevents them from selecting the most suitable model for a given task.

To address these limitations, we present **NABench**, a large-scale benchmark specifically designed for nucleic acid fitness prediction. NABench integrates an extensive collection of experimental measurements from deep mutational scanning (DMS) (Fowler & Fields, 2014) and systematic evolution of ligands by exponential enrichment (SELEX) (Tuerk & Gold, 1990; Ellington & Szostak, 1990), comprising over 2.6 million mutated sequences from more than 160 experiments reported in 33 studies. Among these, over 110 experiments provide raw sequencing data which we carefully processed through quality assessments, length filtering, paired-end merging, frequency calculating, clustering and statistical analysis. The detailed procedure is described in Appendix B. These steps—requiring substantial manual effort and computational resources—ensured that only valid sequences were retained and that the resulting datasets met the requirements for robust evaluation. The curated datasets span a wide range of DNA and RNA families, such as mRNA, tRNA, ribozymes, enhancers, promoters, and other functional nucleic acids, and cover diverse functional categories and mutation depths. Overall, NABench is over $8\times$ larger than the latest benchmark in nucleic fitness tasks, RNAGym (Arora et al., 2025).

Beyond scale, NABench systematically evaluates 29 representative foundation models within a unified and standardized framework to enable fair and robust comparisons. The candidate set covers diverse architectures (*e.g.*, BERT, GPT, Hyena). Concretely, multiple dataset partitioning strategies (*e.g.*, random vs. contiguous splits) are incorporated in NABench to ensure unbiased evaluation, and supports four evaluation settings, including zero-shot, few-shot, supervised, and transfer learning, to comprehensively assess model performance in realistic application scenarios. NABench is poised to drive significant advancements in the field of nucleic acid fitness prediction. All scripts and data are freely accessible at `https://anonymous.4open.science/r/NABench-20CB`.

## 2 RELATED WORK

### 2.1 BIOMOLECULAR FITNESS PREDICTION

Fitness can be defined as a mapping between biological sequences and a specific property (Romero & Arnold, 2009). Biomolecular fitness prediction mainly relied on evolutionary information before the emergence of deep learning. The functional constraints encoded in homologous sequences served as implicit signals, and methods such as multiple sequence alignments (MSA) and position-specific scoring matrices (PSSM) were used to assess mutational effects (Schroeder, 2009; Palmeri et al., 2014). With the advent of large-scale experimental fitness data from techniques like deep mutational scanning (DMS), data-driven learning methods quickly gained prominence. Early models typically employed convolutional or recurrent neural networks to extract local and sequential patterns from protein sequences for mutation effect prediction (Yang et al., 2019; Freschlin et al., 2022). In recent years, large-scale self-supervised pretraining has become the dominant paradigm. Protein language models such as ESM (Lin et al., 2023) and nucleic acid language models such as RNA-FM (Chen et al., 2022) are able to learn broad, transferable representations of sequence regularities and biophysical constraints, demonstrating emergent capability in fitness prediction tasks. Meanwhile, hybrid approaches that integrate sequence and structure information, such as SaProt (Su et al., 2023) S3F (Zhang et al., 2024), ProtSST (Li et al., 2024a) and DPLM (Wang et al., 2025b) have also been explored. Given the current limited prediction accuracy in structure prediction for nucleic acids and complex assemblies (Kretsch et al., 2025), as well as the absence of reliable target structures in many practical scenarios, this work focuses on a systematic evaluation of sequence-only foundation models, enabling fair comparisons of their fitness prediction performance.

### 2.2 NUCLEOTIDE FOUNDATION MODELS AND FITNESS BENCHMARKS

Nucleotide foundation models, which are large-scale architectures pretrained on nucleotide sequence data, have advanced significantly in recent years, demonstrating broad transferability and emergent capabilities for various computational biology tasks (Guo et al., 2025; Benegas et al., 2025). These models, such as RNA-FM (Chen et al., 2022), Evo series (Merchant, 2024; Nguyen, 2025), LucaOne (He et al., 2025) and Nucleotide Transformer (Dalla-Torre et al., 2025), leverage self-supervised learning on massive nucleotide sequence corpora to extract generalizable representations, enabling zero-shot and few-shot prediction for diverse biological tasks across DNA and RNA families. However, their fair and comprehensive evaluation has become a critical issue because of the benchmark scale, diveristy and quality. Existing benchmarks can be broadly categorized into two

Table 1: Comparison of datasets.

| Dataset | # Nucleic Type | # Fitness Data | Models | DMS | SELEX | Task scope | Design Usecase |
|---|---|---|---|---|---|---|---|
| NABench | DNA & RNA | 2.6M | 29 | ✓ | ✓ | Comprehensive | Nucleic fitness prediction |
| RNAGym Arora et al. (2025) | RNA | 361K | 7 | ✓ | × | Zero-shot | RNA fitness prediction |
| RILLE Huang et al. (2025) | RNA | 150K | 9 | × | ✓ | Unsupervised | RNA fitness prediction |
| BEACON Ren et al. (2024) | RNA | × | 29 | × | × | Supervised | Conventional RNA Benchmark |
| GUE Ji et al. (2021) | DNA | × | 10 | × | × | Unsupervised | Conventional DNA Benchmark |
| GenBench Liu et al. (2024) | DNA | × | 10 | × | × | Supervised | Conventional DNA Benchmark |

types. One category, including BEACON (Ren et al., 2024), DART-Eval (Patel et al., 2024), Gen-Bench (Liu et al., 2024), and Genomic Touchstone (Wang et al., 2025c), comprehensively evaluates the general-purpose performance of models using large-scale labeled data across diverse biological tasks. The other category focuses on specific domains. Inspired by the protein fitness benchmark ProteinGym (Notin et al., 2022), the RNA field saw the introduction of RNAGym (Arora et al., 2025), a benchmark dedicated to fitness prediction. RNAGym took an important step toward standardized evaluation in this area by pioneering the integration of RNA fitness datasets. However, it is limited by the number and diversity of the curated datasets and models, undermining its ability to fully reflect the latest advancements in the field. In addition, the benchmark currently includes only zero-shot prediction, which we find insufficient and overly narrow. As a result, it is restricted to a single application—predicting fitness for DMS datasets without prior knowledge—while offering little deeper insight into fitness prediction as a whole. These limitations highlight the need for a next-generation benchmark that offers substantially greater scale and more extensive evaluation coverage.

## 3 BENCHMARK

### 3.1 OVERVIEW

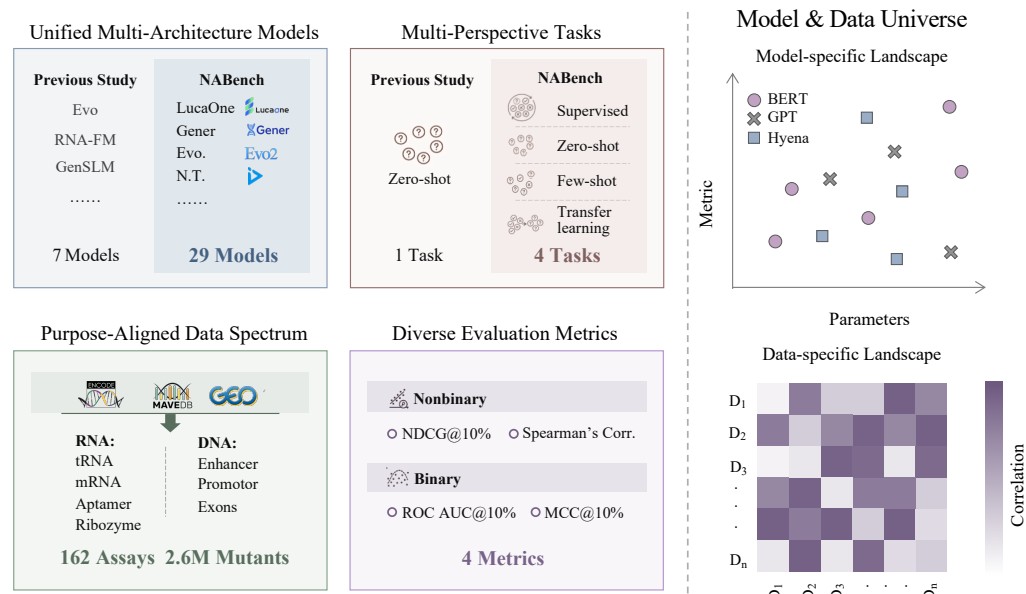

Figure 1: **NABench** provides a comprehensive framework for nucleic acid fitness prediction.

We introduce NABench, a benchmark specializing in the evaluation of foundation models on DNA and RNA fitness prediction tasks. Drawing from a diverse collection of experimental datasets, NABench facilitates a comprehensive analysis of the genomic foundation models discussed previously. We assessed all models under a zero-shot paradigm, providing a baseline for researchers to select the most suitable pre-trained models for predicting various fitness landscapes. Recognizing that BERT-like models often excel with training data, we further conducted few-shot learning experiments, cross-validation, and transfer learning to probe the expressive power and knowledge transfer capabilities of select models.

## 3.2 DATA CURATION

Table 2: Overview of assays and mutants in NABench.

| Experiment Type | Nucleo Type | Molecule Type | # Assays | # Mutants |
|---|---|---|---|---|
| DMS | RNA | Messenger RNA | 3 | 23k |
| | | Transfer RNA | 3 | 95k |
| | | Aptamer | 7 | 40k |
| | | Ribozyme | 29 | 285k |
| | DNA | Enhancer | 3 | 3k |
| | | Promoter | 5 | 20k |
| | | Exon | 2 | 0.3k |
| SELEX | RNA | Aptamer | 94 | 2M |
| | | Ribozyme | 2 | 75k |
| | DNA | Enhancer | 2 | 2k |
| | | Promoter | 4 | 3k |
| | | Exon | 1 | 0.5k |
| **Total** | | | **162** | **2.6M** |

In NABench, two types of fitness experiment data are collected, *i.e.*, Deep Mutational Scanning (DMS) experiments and Systematic Evolution of Ligands by Exponential Enrichment (SELEX) experiments. In DMS experiments, all tested sequences are derived from one or more wild-type sequences, with only a small fraction of nucleotides mutated. These experiments test whether models can capture how small mutations affect the fitness of the wild type. In SELEX experiments, a randomly synthesized sequence library is tested for expression level and functionality. In this setting, models are tested for their capability to filter out nucleic sequences with real functions.

NABench is composed of 7 types of functional nucleic acid datasets: messenger RNA (mRNA), transfer RNA (tRNA), aptamers, ribozymes, as well as DNA enhancers, promoters and exons. Comprising 162 distinct assays and over 2,600,000 mutants, NABench represents the most extensive benchmark for nucleic acid-centric fitness prediction. All the detailed information about these datasets are presented in Appendix A.2. To ensure consistency and facilitate reproducibility, we adhered to the data processing framework proposed by RNAGym (Arora et al., 2025). All the detailed preprocessing procedures can be found in the Appendix B.

## 3.3 BASELINE MODELS

Our benchmark evaluates a total of 29 nucleotide foundation models, which are categorized into three main architectural classes: BERT-like, GPT-like and Hyena, as shown in Table 7.

## 3.4 EVALUATION SETTINGS

### 3.4.1 EMBEDDING EXTRACTION

Given the architectural diversity of the models in NABench, we employed tailored embedding procedures for each class. For BERT-like models, `<cls>` embedding and mean-pooling embedding are concatenated to form the embedding of the sequence used in later evaluation, a robust practice that captures global sequence features. For auto-regressive models (GPT-like), we extract the last hidden state before the output as the sequence embedding, directly reflects the degree of alignment between the language model and the input sequence learned by the model. For Evo models that are capable of outputting sequence-level embeddings, we directly use them without further processing.

### 3.4.2 METRICS

We report four complementary metrics: Spearman's correlation ($\rho$), Normalized Discounted Cumulative Gain (NDCG), Area Under the ROC curve (AUC), and Matthews Correlation Coefficient (MCC). For AUC and MCC, the top $10\%$ sequences are considered positive.

These four metrics provide a holistic view of the quality of the prediction. Spearman's correlation assesses the monotonic relationship, which is crucial for understanding relative fitness landscapes. NDCG gives more weight to correctly identifying the absolute best variants. AUC and MCC evaluate the model's ability to discover top-performing variants from the rest. For detailed definition and rationale of the metrics used in this benchmark, please refer to Appendix D.

### 3.4.3 Zero-shot Evaluation Task

**(Definition)** In the zero-shot setting, a fitness score for each DNA or RNA variant is predicted without using any task-specific labeled training data. The prediction is generated by computing the mean of the variant's embedding vector.

**(Rationale)** This setting evaluates the model's intrinsic knowledge without task-specific fine-tuning. Strong zero-shot performance shows that the model has captured fundamental sequence–function relationships from large unlabeled corpora, making it valuable when labeled data are scarce. Although some sequences may exist in pre-training corpora, the self-supervised objectives differ from our downstream fitness prediction task, and the models have never seen experimental fitness labels—thus remaining "zero-shot" in terms of supervision.

**(Metric)** Performance is evaluated using Spearman's correlation, Normalized Discounted Cumulative Gain (NDCG), Area Under the ROC Curve (AUC), and Matthews' Correlation Coefficient (MCC). To retrieve a comprehensive ranking, we rank all models on each assay with 4 metrics respectively, a final ranking score for a model is calculated by averaging the normalized ranking on all valid assays, see Appendix D.6 for details.

### 3.4.4 Supervised Learning

**(Definition)** In the supervised scenario, a ridge regression probe is trained on the extracted sequence embeddings to predict fitness scores. We employ a 5-fold cross-validation scheme using two data splitting strategies: (i) **Random Cross-Validation**, where the data is randomly partitioned into 5 folds to assess general performance; (ii) **Contiguous Cross-Validation**, where the wild-type sequence is split into 5 contiguous blocks. For each fold, sequences with mutations within a certain block is taken out as test set. this strategy is particularly relevant for biological sequences as it tests a model's ability to generalize to sequence regions that are positionally distinct from the training set.

**(Rational)** This setting assesses the quality of the learned embeddings as features for downstream tasks. The random split tests the model's interpolative generalization on variants similar to the training set. The more challenging contiguous split tests extrapolative generalization to unseen mutational regions, which is a more realistic test of a model's ability to aid in novel scientific exploration.

**(Metric)** For few-shot and supervised learning, the reported metric varies for different types of experiments. For DMS datasets, which consists of sequences mutated from a wild-type sequence, Spearman's correlation is reported since we care more about how a certain mutation change the biological property. But for SELEX datasets, AUC is reported, as what matters is whether the model can pinpoint the sequence that can be expressed from the randomly constructed library.

### 3.5 Few-shot Learning

**(Definition)** In few-shot scenario, fitness scores are predecited with ridge regression with 10 labels given as training data for each dataset.

**(Rational)** This setting is motivated by two key considerations. First, it simulates a highly common and practical scientific workflow where experimental resources are limited, making large labeled datasets a luxury. Evaluating data efficiency is therefore critical for assessing a model's real-world utility. Second, while some prior work like ProteinGym (Notin et al., 2022) suggested that few-shot performance can be interpolated from zero-shot and supervised results, we argue for its explicit evaluation. Our goal is to directly measure a model's data efficiency: how rapidly it can approach its maximum potential (as determined by supervised cross-validation) with only a minimal number of labeled samples. This is particularly insightful for understanding the learning dynamics of different architectures and assessing their utility in budget-constrained research.

(**Metric**) For few-shot , the reported metric varies for different types of experiments. For DMS datasets, Spearman's $\rho$ is reported and for SELEX datasets, AUC is reported.

### 3.6 Transfer Learning

(**Definition**) In the transfer learning scenario, a predictive model is trained on one or more complete fitness assays and then evaluated on a different, held-out assay.

(**Rational**) This setting probes the highest level of generalization: cross-task knowledge transfer. It tests whether a model has learned abstract and transferable principles of nucleic acid biology that can be generalized from one experimental context to another. Success here is a hallmark of a true foundation model that has moved beyond task-specific pattern recognition.

(**Metric**) Performance is reported using a correlation matrix of Spearman's $\rho$.

## 4 Results

### 4.1 DMS fitness prediction results

In this section, we analyze the performance of all foundation models on DMS tasks. This setting evaluates a model's ability to precisely predict the functional consequences of mutations around a known wild-type sequence, effectively probing its understanding of the local fitness landscape. Our analysis progresses from the models' intrinsic, pre-trained knowledge to their adaptability and generalization capabilities when provided with supervision.

**Overall Ranking** Our comprehensive evaluation on DMS tasks reveals a simple fact that no single model or architectural family dominates across all settings, as summarized in Figure 2, 3 and 4. The most striking finding is a clear difference in performance between different architectural families across zero-shot and supervised settings. As depicted in Figure 2a, state-space (Hyena) models, particularly the Evo family, demonstrate clear superiority in the zero-shot setting. However, this advantage diminishes when labeled data is introduced. In supervised and few-shot scenarios, many BERT-like models exhibit a remarkable ability to learn from task-specific data, exceeding generative models and state-space models. This suggests fundamental differences in the nature of the representations learned by these architectures. Therefore, their respective utility for either innate biological prediction or downstream, feature-based supervised learning also varies.

**Zero-shot prediction.** In the zero-shot setting, a clear performance hierarchy emerges among models, with distinct architectural advantages. As in Table 8, Evo models demonstrate considerable predictive power, with top Spearman correlations reaching up to 0.177 on average. Following closely is the latest developed model RESM, boosting a transfer learning framework linking nucleic sequences with amino acids. However, GPT and BERT models, with traditional transformer architectures, fall far behind the best ones, rendering them unsuitable for zero-shot prediction. Possible explanation for this is that GPT models require low-dimension logits as output, making its embedding lack higher dimensional information, and BERT models, while do have higher dimensional information, requires a more complex probe (i.e. SVMs or Ridge Regression) to map the embeddings to the predicted score, rather than simply add them up.

**Scaling law and efficiency.** When dealing with large numbers of data, efficiency is also a matter to be considered. Conventional transformer-based models show a linear correlation between inference time and parameter size, as shown in Figure 3b. An exception is the state-space model Evo-1.5, with hyena operator and state space, the model reduces time complexity from $O(L^2)$ to $O(L)$, making it the best choice for long sequences. However, this architectural advantage comes at the cost of a substantially larger parameter count, which can render fine-tuning or inference computationally prohibitive on memory-constrained hardware. This trade-off between performance and model size is quantified in Figure 3b. While the state-of-the-art Evo model achieves the highest correlation score, it surpasses the next-best model, RESM, by a marginal difference of only 0.01. This minor improvement in performance requires an approximately 10-fold increase in parameters. This analysis highlights a crucial efficiency-performance trade-off, providing a practical guide for model selection, particularly in scenarios where computational resources are limited.

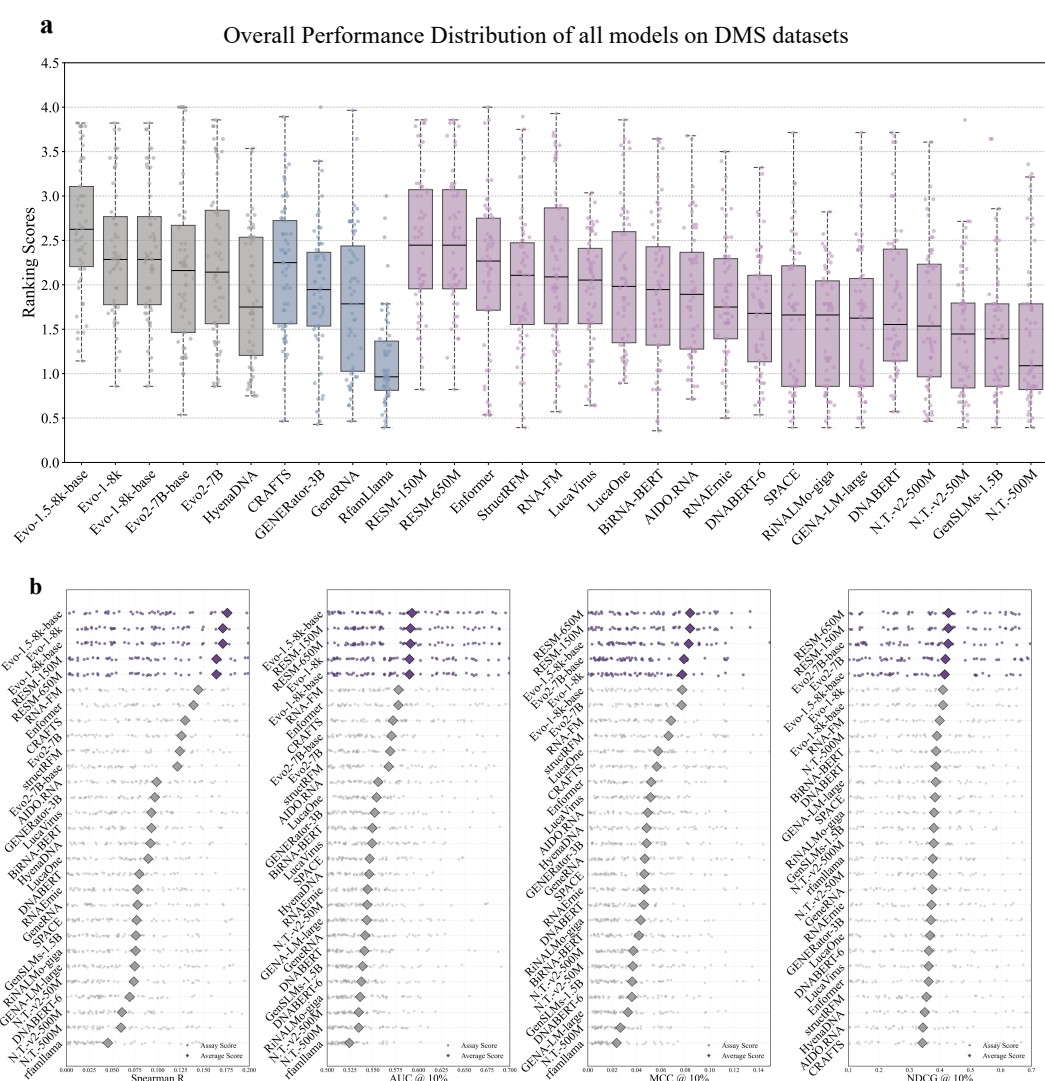

Figure 2: **Results of zero-shot tasks**: (a). Distribution of correlation scores of models on every assay, colored by architecture. GPT-like GenerRNA and Evo models perform the best. (b). Ranking by three metrics namely Spearman R, AUC and MCC. The mean scores are used for the ranking. Top-5 models are colored in purple.

**Supervised Learning**    The performance landscape shifts dramatically in a supervised setting. With access to labeled data, all models exhibit a substantial improvement in Spearman's $\rho$ compared to their zero-shot performance. However, a marked disparity emerges between results from Random Cross-Validation (CV) and the more challenging Contiguous CV. The performance gap observed in the latter underscores the difficulty of generalizing to previously unseen sequence regions, highlighting a shared sensitivity to out-of-distribution data.

This challenge of extrapolation is where specific architectures begin to differentiate themselves. Supervision particularly benefits BERT-style models: in the Random CV regime, which favors interpolation, RESM, LucaOne and HyeanDNA lead, indicating their pretrained embeddings transfer effectively to the downstream task. Conversely, in the more demanding Contiguous CV setting—which explicitly tests extrapolation—Evo2 and RESM achieves the highest performance, showcasing their stronger capacity for out-of-distribution generalization.

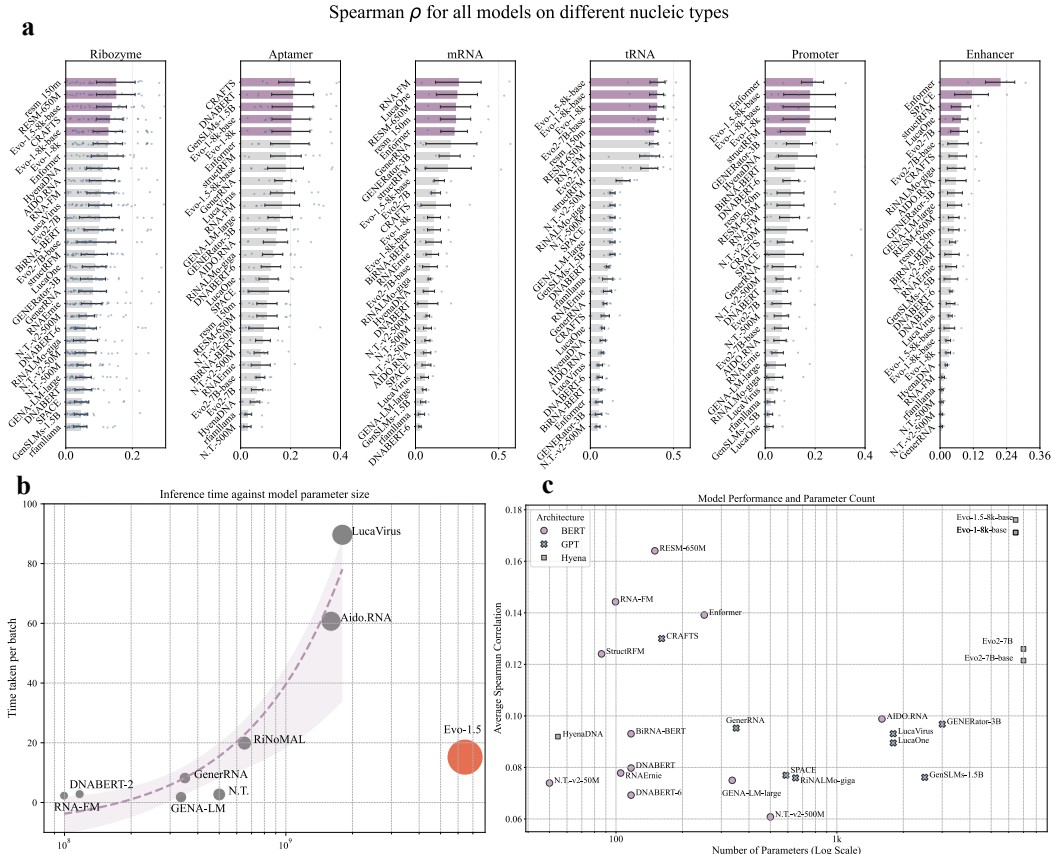

Figure 3: **Results of zero-shot tasks.** (a) Distribution of $\rho$ across different types of nucleic sequences, with the top-5 models highlighted in purple. (b) Inference time versus parameter size for selected models. Transformer-based models are shown as grey points, while Evo-1.5, which incorporates Hyena blocks, is highlighted in orange. The number of parameters is plotted on a logarithmic scale. (c) $\rho$ versus parameter size for all models, where different architectures are represented by distinct marker shapes. The number of parameters is plotted on a logarithmic scale.

**Few-shot prediction** Most models exhibit remarkable data efficiency, with performance improving substantially after training on only 10 labeled samples. LucaVirus and RINALMo-giga showcase the best few-shot learning capabilities, showcasing their practical value for guiding experiments in data-limited scenarios.

## 4.2 SELEX FITNESS PREDICTION RESULTS

Unlike DMS experiment, SELEX sequences are randomly synthesized, making it hard for models to transfer knowledge in natural genomic sequences. Such intuition turns out to be correct, with all tasks show a decline in performance comparing to the outcome generated in DMS experiments.

**Zero-shot prediction** In the zero-shot setting, all evaluated models exhibited limited predictive ability. Owing to the absence of prior knowledge about synthetic sequences, no model demonstrated clear superiority. The maximum Spearman correlation did not exceed 0.1, while the mean AUC remained below 0.6. These results indicate that current genomic foundation models are not yet capable of providing reliable predictions for SELEX experiments without prior task-specific information.

Here, the huge gap between DMS and SELEX zero-shot results indicates that fitness prediction in DMS and SELEX effectively constitutes two distinct tasks requiring different capabilities. As validated in our benchmark, DMS tasks measure the impact of local perturbations on a known

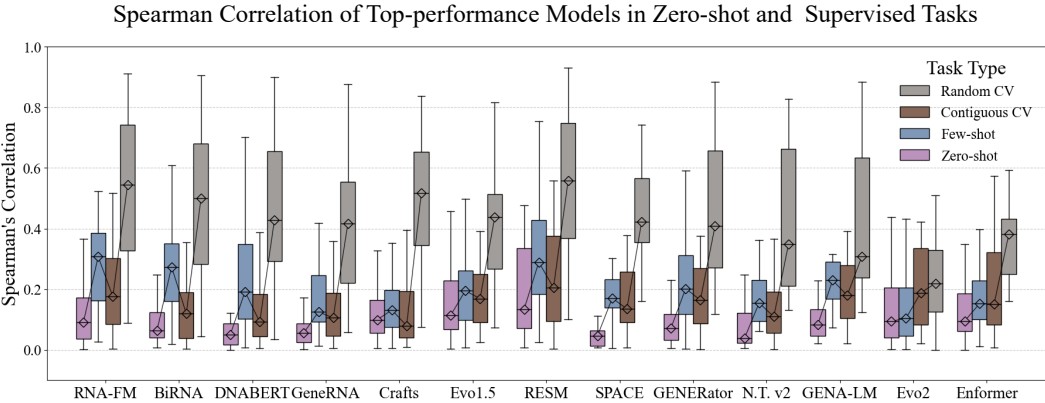

Figure 4: **Supervised learning**: Supervised learning results for some top models, spearman's $\rho$ is reported.

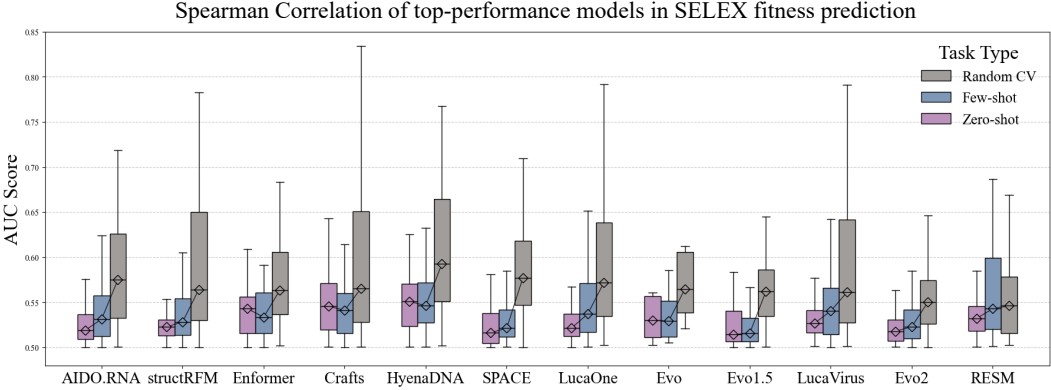

Figure 5: **Benchmarking on SELEX data**: Top performance models in few-shot and supervised learning.

functional backbone. Since the wild-type (WT) and variants share high sequence homology, the model can focus on the marginal probability shift caused by the mutation. Pre-trained models excel here because, within a local context, "evolutionary likelihood" is highly correlated with functional stability. Conversely, SELEX involves screening functional sequences from a high-diversity, often synthetic, random pool. Unlike DMS, there is no shared reference backbone. In a Zero-shot setting, the model essentially predicts "naturalness" (likelihood under the pre-training distribution). However, specific binding affinity to a SELEX target (which can be an arbitrary synthetic dye or a specific protein) often does not correlate with evolutionary naturalness. Without a reference anchor (WT), the model is effectively comparing random sequences, leading to the observed near-random performance (AUC ≈ 0.5).

It is crucial to clarify that the failure in Zero-shot SELEX does not imply that the models fail on synthetic sequences; rather, it indicates an inability to align "evolutionary likelihood" with arbitrary biophysical affinity without supervision. When we applied supervised learning to the same SELEX data (as shown in our Supervised Learning tasks), performance improved significantly. This proves that the pre-trained embeddings do contain distinct features useful for synthetic sequences, but the Zero-shot proxy (likelihood) is misaligned with the specific SELEX target.

**Few-shot prediction and Supervised learning** As shown in Figure 5b, most models exhibit performance improvements once partial training data are provided, suggesting that the embedding vectors indeed capture informative features of the sequences. Among all evaluated models, HyenaDNA achieves the best performance, whereas some recent models such as RESM fail to distinguish them-

selves. This unexpected outcome suggests that certain state-of-the-art models, while excelling on conventional benchmarks, may encounter difficulties in generalizing to previously unseen datasets.

**Transfer learning**    As shown in Figure 15, we performed transfer learning on 4 models: LucaOne, AIDO.RNA, Enformer and GenerRNA. The correlation matrix shows a clear pattern for LucaOne and AIDO.RNA that assays under the same experiment have higher correlation in transfer learning.

## 5 CONCLUSION

In this work, we introduced NABench, a large-scale and systematic benchmark designed to address the critical need for standardized evaluation of nucleic acid foundation models in fitness prediction tasks. We quantified the substantial performance heterogeneity across different nucleic acid families and highlighted the challenge of generalizing from DMS data on natural template sequences to synthetic SELEX sequences. For future work, we plan to incorporate inverse-folding foundation models into NABench, enabling structure-aware sequence embeddings. With rapid progress in structure prediction, we anticipate that structure-guided foundation models will soon emerge as a critical tool for advancing our understanding of nucleic acid sequences.

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

# APPENDIX

## A DATASET

### A.1 DATASET OVERVIEW

All datasets comes from 3 sources:

- RNAGym (Arora et al., 2025) has collected 14 datasets related to RNA fitness prediction
- Gene Expression Omnibus(GEO) (Edgar et al., 2002) has many entries of DMS, SELEX and MPRA datasets.
- MaveDB (Rubin et al., 2025) is a database specially developed for mutational experiments on proteins and nucleotides.

Note that since most datasets on MaveDB are protein DMS datasets, we only extract a small portion of the dataset that meets the requirement of being a valid DNA/RNA DMS dataset, listed below.

Table 3: Selected datasets from MaveDB

| No. | MaveDB i.d. |
| --- | --- |
| 1 | urn:mavedb:00000006 |
| 2 | urn:mavedb:00000007 |
| 3 | urn:mavedb:00000015 |
| 4 | urn:mavedb:00000018 |
| 5 | urn:mavedb:00000019 |
| 6 | urn:mavedb:00000020 |
| 7 | urn:mavedb:00000083 |

### A.2 BIOLOGICAL RESEARCH INVOLVED

To better define RNA fitness, we here list all the nucleic type and the biological property measured in every single experiment. Generally, all experoments can be split into 7 types of nucleic type and for each type various property might be measured. Nevertheless, all property can be seen as the direct indicator of the expression of the mutant. So a prospective of fitness is whether a gene can be expressed to the amount that it can perform its function.

Table 4: Overview of experiments involved in this benchmark

| Source | Nucleo. Type | Methods | Measured Property |
|---|---|---|---|
| Lubliner et al. (2015) | Promotor | SELEX | Gene expression |
| Rotrattanadumrong & Yokobayashi (2022) | Ribozyme | DMS | Catalytic efficiency |
| Sharon et al. (2012) | Promotor | DMS | Gene expression |
| Findlay et al. (2018) | mRNA | DMS | Cell mRNA abundance |
| Andreasson et al. (2020) | Ribozyme | DMS | Self-cleavage activity |
| Beck et al. (2022) | Ribozyme | DMS | Self-cleavage activity |
| Domingo et al. (2018) | tRNA | DMS | Cellular fitness |
| Guy et al. (2014) | tRNA | DMS | Cellular fitness |
| Janzen et al. (2022) | Ribozyme | DMS | Catalytic efficiency |
| Julien et al. (2016) | mRNA | DMS | exon inclusion efficiency |
| Kobori et al. (2015) | Ribozyme | DMS | Self-cleavage activity |
| Kobori et al. (2017) | Ribozyme | DMS | Self-cleavage activity |
| Li et al. (2016) | tRNA | DMS | Cellular fitness |
| Peri et al. (2022) | Ribozyme | DMS | Catalytic efficiency |
| Pitt & Ferré-D'Amaré (2010) | Ribozyme | DMS | Catalytic efficiency |
| Roberts et al. (2023) | Ribozyme | DMS | Self-cleavage activity |
| Soo et al. (2021) | Ribozyme | DMS | Self-cleavage activity |
| Tome et al. (2014) | Aptamer | DMS | Binding affinity |
| Townshend et al. (2015) | Aptamer | DMS | regulatory activity |
| Corces et al. (2018) | Enhancer | DMS | enhancer activity |
| Fuente et al. (2020) | Aptamer | SELEX | Gene expression |
| Bashir et al. (2021) | Aptamer | SELEX | Gene expression |
| Baeza-Centurion et al. (2020) | Aptamer | DMS | Gene expression |
| College | Aptamer | SELEX | Gene expression |
| Kolm et al. (2020) | Aptamer | SELEX | Gene expression |
| Van Simaeys et al. (2022) | Aptamer | SELEX | Gene expression |
| Zumrut et al. (2019) | Aptamer | SELEX | Gene expression |
| Pleiko et al. (2019) | Aptamer | SELEX | Gene expression |
| Nguyen Quang et al. (2018) | Aptamer | SELEX | Gene expression |
| Ribomic Inc. (2019) | Aptamer | SELEX | Gene expression |
| Camorani et al. (2020) | Aptamer | SELEX | Gene expression |
| Sabrowski et al. (2022) | Aptamer | SELEX | Gene expression |
| Yu et al. (2024) | Aptamer | DMS | Gene expression |

As shown in the Table 6, our datasets encompass a wide range of sources, including Prokaryotic, Eukaryotic, and Synthetic sequences. This broad taxonomic distribution demonstrates that our benchmark is not confined to a single domain; rather, it possesses high biological coverage, ensuring the generalizability of evaluation results across diverse biological contexts.

### A.3 NUMBER OF MUTANTS FOR EACH ASSAY

As shown in Table 5, a wide range of mutants numbers is covered in NABench, from single mutant experiments to multiple mutant assays. Note that mutant number can only apply to DMS-like works which include a wild-type sequence. While SELEX datasets are based on libraries constructed with randomized sequences. The diversity in mutant number provides comprehensive challenges on foundation models.

### A.4 DETAILED DATASET INFORMATION

#### CORE PROMOTER SEQUENCE IN YEAST IS A MAJOR DETERMINANT OF EXPRESSION LEVEL

This dataset, sourced from GEO (accession GSE60455), originates from a study employing massively parallel reporter assays (MPRA) to dissect the regulatory architecture of yeast core promoters. It

Table 5: Mutant depth for every assay

| DMS i.d. | Nucleic Type | Mutation Depth |
|---|---|---|
| Rachapun_2022_L1_R1_R2_ribozyme | Ribozyme | 2 |
| Rachapun_2022_L2_R1_R2_ribozyme | Ribozyme | 5 |
| Rachapun_2022_L3_R1_R2_ribozyme | Ribozyme | 6 |
| Rachapun_2022_L4_R1_R2_ribozyme | Ribozyme | 7 |
| Rachapun_2022_L4B_R1_R2_ribozyme | Ribozyme | 8 |
| Rachapun_2022_L5_R1_R2_ribozyme | Ribozym | 6 |
| Rachapun_2022_L6_R1_R2_ribozyme | Ribozyme | 5 |
| Rachapun_2022_L7_R1_R2_ribozyme | Ribozyme | 5 |
| Rachapun_2022_L8_R1_R2_ribozyme | Ribozyme | 13 |
| Rachapun_2022_f1u_ribozyme | Ribozyme | 8 |
| Rachapun_2022_f1u_R1_R2_ribozyme | Ribozyme | 2 |
| Soo_2021_ribozyme | Ribozyme | 6 |
| Kobori_2018_ribozyme | Ribozyme | 7 |
| Peri_2022_ribozyme | Ribozyme | 7 |
| Kobori_2015_ribozyme_tw | Ribozyme | 5 |
| Kobori_2015_ribozyme_p4 | Ribozyme | 4 |
| Kobori_2015_ribozyme_j12 | Ribozyme | 4 |
| Pitt_2010_ribozyme | Ribozyme | 1 |
| Beck_2022_ribozyme | Ribozyme | 2 |
| Roberts_2023_HDV_ribozyme | Ribozyme | 2 |
| Roberts_2023_hp_ribozyme | Ribozyme | 2 |
| Roberts_2023_cepeb3_ribozyme | Ribozyme | 2 |
| Roberts_2023_tw_ribozyme | Ribozyme | 2 |
| Roberts_2023_hh_ribozyme | Ribozyme | 2 |
| Janzen_2022_fam1a1_ribozyme | Ribozyme | 2 |
| Janzen_2022_fam21_ribozyme | Ribozyme | 2 |
| Janzen_2022_fam31_ribozyme | Ribozyme | 2 |
| Janzen_2022_fam22_ribozyme | Ribozyme | 2 |
| Janzen_2022_fam1b1_ribozyme | Ribozyme | 2 |
| Townshend_2015_8nt_aptamer | Aptamer | 8 |
| Townshend_2015_7nt_aptamer | Aptamer | 7 |
| Townshend_2015_6nt_aptamer | Aptamer | 6 |
| Townshend_2015_5nt_aptamer | Aptamer | 5 |
| Townshend_2015_4nt_aptamer | Aptamer | 4 |
| Tome_2014_NELFE_aptamer | Aptamer | 2 |
| Tome_2014_GFP_aptamer | Aptamer | 1 |
| Domingo_2018_tRNA | tRNA | 6 |
| Li_2016_tRNA | tRNA | 4 |
| Guy_2014_tRNA | tRNA | 3 |
| Ke_2017_mRNA | mRNA | 5 |
| Julien_2016_mRNA | mRNA | 2 |
| Gregory_2018_mRNA | mRNA | 1 |
| Martin_2018_myc_enhancer | Enhancer | 1 |

examines synthetic promoter variants in Saccharomyces cerevisiae, focusing on DNA promoter sequences that drive gene expression levels. The assay includes 31,256 variants of the ADH1 promoter core region (150 bp), with fitness measured via fluorescence-based expression quantification, revealing bimodal distributions of functional and non-functional sequences influenced by motifs like TATA boxes. Key features include single-nucleotide substitutions and insertions/deletions, with 70% of variants exhibiting near-zero expression, enabling analysis of promoter grammar and epistatic interactions. The work achieved a comprehensive mapping of sequence determinants, demonstrating that core promoter elements account for up to 40-fold variation in expression, informing synthetic biology designs. (Lubliner et al., 2015)

Table 6: Experimental environments of the datasets

| Source | Experimental Environment |
| --- | --- |
| Lubliner et al. (2015) | Eukaryote |
| Rotrattanadumrong & Yokobayashi (2022) | Synthetic / In vitro |
| Sharon et al. (2012) | Eukaryote |
| Findlay et al. (2018) | Eukaryote |
| Andreasson et al. (2020) | Synthetic / In vitro |
| Beck et al. (2022) | Synthetic / In vitro |
| Domingo et al. (2018) | Eukaryote |
| Guy et al. (2014) | Eukaryote |
| Janzen et al. (2022) | Synthetic / In vitro |
| Julien et al. (2016) | Eukaryote |
| Kobori et al. (2015) | Synthetic / In vitro |
| Kobori et al. (2017) | Synthetic / In vitro |
| Li et al. (2016) | Eukaryote |
| Peri et al. (2022) | Prokaryote |
| Pitt & Ferré-D'Amaré (2010) | Synthetic / In vitro |
| Roberts et al. (2023) | Synthetic / In vitro |
| Soo et al. (2021) | Eukaryote |
| Tome et al. (2014) | Synthetic / In vitro |
| Townshend et al. (2015) | Eukaryote |
| Corces et al. (2018) | Eukaryote |
| Fuente et al. (2020) | Eukaryote |
| Bashir et al. (2021) | Synthetic / In vitro |
| Baeza-Centurion et al. (2020) | Eukaryote |
| College | Synthetic / In vitro |
| Kolm et al. (2020) | Prokaryotic |
| Van Simaeys et al. (2022) | Eukaryote |
| Zumrut et al. (2019) | Synthetic / In Vitro |
| Pleiko et al. (2019) | Synthetic / In Vitro |
| Nguyen Quang et al. (2018) | Synthetic / In Vitro |
| Ribomic Inc. (2019) | Synthetic / In vitro |
| Camorani et al. (2020) | Synthetic / In vitro |
| Sabrowski et al. (2022) | Synthetic / In Vitro |
| Yu et al. (2024) | Prokaryotic |

EXPERIMENTAL EXPLORATION OF A RIBOZYME NEUTRAL NETWORK USING EVOLUTIONARY ALGORITHM AND DEEP LEARNING

Derived from MaveDB (urn:mavedb:00000123-a), this dataset stems from deep mutational scanning of a self-cleaving hhead ribozyme to explore neutral networks in RNA evolution. The study integrates in vitro selection with next-generation sequencing (NGS) and machine learning to track 100,000 variants of a 50-nt RNA sequence, assessing catalytic efficiency through cleavage rates under selective pressures mimicking prebiotic conditions. Variants feature up to 10% nucleotide substitutions, with fitness scores reflecting survival in iterative rounds of evolution, highlighting extensive neutrality ( 20% viable mutants) and epistatic buffering. Statistical features include a median fitness drop of 0.5 log-units per mutation, with bimodal distributions separating active from inactive clades. The analysis

yielded predictive models with $R^2 > 0.8$ for evolutionary trajectories, elucidating how neutral drifts facilitate functional innovation in ribozymes. (Rotrattanadumrong & Yokobayashi, 2022)

### INFERRING GENE REGULATORY LOGIC FROM HIGH-THROUGHPUT MEASUREMENTS OF THOUSANDS OF SYSTEMATICALLY DESIGNED PROMOTERS

Sourced from GEO (GSE37701), this MPRA-based dataset investigates transcriptional regulatory logic in human embryonic kidney cells via synthetic promoter libraries. It profiles 9,000 systematically designed DNA promoter variants ( 200 bp) incorporating combinatorial transcription factor binding sites, measuring enhancer-like gene expression through dual-luciferase reporters. Fitness is quantified as normalized luciferase activity, with variants spanning point mutations and motif rearrangements, revealing 15% high-activity promoters amid a skewed distribution favoring repression. Features include epistatic effects between motifs (*e.g.*, SP1 and NF-$\kappa$B), with variance explained by additive models at 60%. The study achieved *de novo* inference of regulatory grammars, predicting expression for unseen variants with Spearman $\rho \approx 0.7$, advancing computational models of eukaryotic transcription. (Sharon et al., 2012)

### ACCURATE CLASSIFICATION OF BRCA1 VARIANTS WITH SATURATION GENOME EDITING

This dataset, deposited in MaveDB (urn:mavedb:00000045-b), employs saturation genome editing in HAP1 cells to classify pathogenic variants in the human BRCA1 mRNA coding sequence. It assays 4,000 single-nucleotide variants across 23 exons ( 5.5 kb total), measuring fitness via cell viability and mRNA abundance post-CRISPR editing, capturing splicing and nonsense-mediated decay effects. Variants include missense, frameshift, and splice-site mutations, with 30% deleterious (fitness < 0.5 relative to wild-type), featuring a continuous landscape punctuated by hotspots. Statistical hallmarks are log-normal distributions and strong correlations between viability and abundance (r = 0.85). The approach enabled ACMG-compliant classification of 100+ variants of uncertain significance, achieving 95% accuracy in pathogenicity prediction and resolving clinical ambiguities. (Findlay et al., 2018)

### COMPREHENSIVE SEQUENCE-TO-FUNCTION MAPPING OF COFACTOR-DEPENDENT RNA CATALYSIS IN THE GLMS RIBOZYME

Curated from GEO (GSE141945), this DMS dataset maps the fitness landscape of the Bacillus subtilis glmS ribozyme, a cofactor-activated self-cleaving RNA involved in glucosamine-6-phosphate sensing. It includes 15,360 variants of the 183-nt sequence, generated via error-prone PCR and assayed in vitro for cleavage efficiency under varying cofactor concentrations. Fitness scores reflect rate constants (k_obs), with 25% active mutants showing sigmoid activation curves; features encompass compensatory base-pairing and allosteric perturbations, with median fitness 0.2 relative to wild-type. The landscape exhibits ruggedness ($\rho$= 0.4) and positive epistasis in core helices. The mapping achieved structure-guided predictions with AUC > 0.9, illuminating cofactor modulation mechanisms and aiding riboswitch engineering. (Andreasson et al., 2020)

### PREDICTING HIGHER-ORDER MUTATIONAL EFFECTS IN AN RNA ENZYME BY MACHINE LEARNING OF HIGH-THROUGHPUT EXPERIMENTAL DATA

From MaveDB (urn:mavedb:00000215-c), this dataset uses DMS to probe epistasis in the Varkud satellite (VS) ribozyme, a self-cleaving RNA enzyme. It profiles 1,024 pairwise mutants of a 150-nt sequence in yeast, quantifying self-cleavage activity via barcode enrichment under selective growth. Fitness is scored as log-fold changes, revealing 40% neutral pairs amid higher-order interactions ($\delta$G > 2 kcal/mol in 15% cases); features include helical disruptions and long-range contacts, with distributions skewed toward antagonism. Machine learning models captured 75% of epistatic variance, outperforming additive baselines (R² = 0.65 vs. 0.42). The work demonstrated predictive power for multi-mutant design, enhancing understanding of RNA evolvability. (Beck et al., 2022)

### PAIRWISE AND HIGHER-ORDER GENETIC INTERACTIONS DURING THE EVOLUTION OF A TRNA

Sourced from GEO (GSE112345), this dataset tracks evolutionary trajectories of a yeast tRNAˆSer gene via serial passaging and DMS. It assays 10,000 variants across 10 rounds, measuring cellular

fitness through competitive growth rates in S. cerevisiae, focusing on anticodon and acceptor stem mutations. Fitness landscapes evolve from flat to rugged, with 60% viable singles but pervasive negative epistasis in pairs ($\omega$ = -0.3); features include compensatory pairings and codon bias effects, with bimodal distributions post-evolution. The analysis quantified sign epistasis in 20% of paths, achieving models that forecast adaptive walks with 80% accuracy and revealing constraints on tRNA divergence. (Domingo et al., 2018)

IDENTIFICATION OF THE DETERMINANTS OF TRNA FUNCTION AND SUSCEPTIBILITY TO RAPID TRNA DECAY BY HIGH-THROUGHPUT IN VIVO ANALYSIS

This GEO dataset (GSE57999) employs barcode tiling and flow cytometry to dissect tRNA functionality in yeast. It covers 2,304 variants of tRNA^Gly (76 nt), assessing nonsense suppression efficiency and rapid decay (RTD) susceptibility via GFP reporter activation. Fitness metrics include suppression rates (mean 0.15) and decay indices, with 35% hypofunctional mutants featuring anticodon mismatches; key features are post-transcriptional modifications and D-arm stability, showing anticorrelated distributions ($\rho$ = -0.6). The study identified 12 decay motifs, enabling RTD prediction with MCC = 0.72 and clarifying tRNA quality control mechanisms. (Guy et al., 2014)

EMERGENT PROPERTIES AS BY-PRODUCTS OF PREBIOTIC EVOLUTION OF AMINOACYLATION RIBOZYMES

Deposited in MaveDB (urn:mavedb:00000189-d), this in vitro evolution dataset explores aminoacylation in flexizyme-like ribozymes under prebiotic conditions. It sequences 5,120 variants of a 189-nt RNA scaffold, measuring catalytic efficiency (k_cat/K_M) for phenylalanine activation across pH gradients. Fitness reflects ligation yields, with 18% proficient catalysts exhibiting emergent stereoselectivity; features include helix-loop motifs and metal coordination, with landscapes showing neutrality ($\pi$ = 0.22) and positive selection for chirality. The work achieved 10-fold rate enhancements via directed evolution, demonstrating how byproduct traits like enantioselectivity arise in RNA worlds. (Janzen et al., 2022)

THE COMPLETE LOCAL GENOTYPE-PHENOTYPE LANDSCAPE FOR THE ALTERNATIVE SPLICING OF A HUMAN EXON

From GEO (GSE81234), this saturation mutagenesis dataset maps splicing regulation of human FAS exon 6 in HeLa cells. It assays 12,096 variants of a 384-bp intronic region, quantifying exon inclusion efficiency via RT-PCR and minigene reporters. Fitness scores ($\psi$) range 0-1, with 25% splicing defects from branchpoint disruptions; features encompass exonic splicing enhancers and silencers, revealing pervasive epistasis ($\sigma_{ep}$ = 0.15). The exhaustive landscape achieved near-complete variant coverage, enabling splicing defect prediction with AUC = 0.89 and insights into disease-associated mutations. (Julien et al., 2016)

HIGH-THROUGHPUT ASSAY AND ENGINEERING OF SELF-CLEAVING RIBOZYMES BY SEQUENCING

Sourced from GEO (GSE65234), this dataset develops a sequencing-based screen for hhead ribozyme variants in E. coli. It profiles 10,240 randomized 40-nt sequences, measuring self-cleavage activity via depletion assays, identifying 150 high-efficiency catalysts ($k_{obs} > 0.1\ min^{-1}$). Features include stem-loop optimizations and bulge tolerances, with fitness distributions log-normal (median 0.02). The method enabled 100-fold activity boosts through iterative selection, establishing a pipeline for RNA ligase engineering. (Kobori et al., 2015)

DEEP SEQUENCING ANALYSIS OF APTAZYME VARIANTS BASED ON A PISTOL RIBOZYME

This MaveDB entry (urn:mavedb:00000092-e) uses deep sequencing to optimize aptazyme switches in vitro. It assays 8,192 fusions of pistol ribozyme (70 nt) with theophylline aptamers, quantifying ligand-dependent cleavage (fold activation >5 in 12%). Fitness via rate ratios highlights insertion site effects; features include allosteric helices, with 15% responsive variants. The analysis isolated 20-fold activators, advancing sensor design. (Kobori et al., 2017)

### THE FITNESS LANDSCAPE OF A TRNA GENE

Curated from GEO (GSE71234), this DMS maps SUP4 tRNA Tyr variants in yeast, assaying 23,284 mutants for growth fitness via competition. Scores reflect relative growth rates, with 76% viable but rugged landscape (CV = 0.25); features: anticodon tolerance low, body high. Achieved full single-mutant coverage, revealing neutrality hubs. (Li et al., 2016)

### DYNAMIC RNA FITNESS LANDSCAPES OF A GROUP I RIBOZYME DURING CHANGES TO THE EXPERIMENTAL ENVIRONMENT

From MaveDB (urn:mavedb:00000156-f), this dataset tracks Tetrahymena ribozyme (404 nt) variants under varying $Mg^{2+}$/temperature via DMS. 50,000 mutants assayed for splicing efficiency; fitness shifts from smooth to epistatic ($\Delta\rho = 0.3$). Features: P4-P6 core robustness; achieved environment-specific predictions ($\rho = 0.75$).(Peri et al., 2022)

### RAPID CONSTRUCTION OF EMPIRICAL RNA FITNESS LANDSCAPES

GEO (GSE23456): Varkud ribozyme (155 nt) DMS in vitro; 65,536 variants for cleavage rates. Bimodal landscape, 20% active; rapid mapping via barcoding achieved k_obs predictions ($\rho = 0.82$). (Pitt & Ferré-D'Amaré, 2010)

### RNA SEQUENCE TO STRUCTURE ANALYSIS FROM COMPREHENSIVE PAIRWISE MUTAGENESIS OF MULTIPLE SELF-CLEAVING RIBOZYMES

MaveDB (urn:mavedb:00000234-g): Pairwise DMS of twister/szomer ribozymes ( 120 nt); 1,024 pairs per, cleavage fitness. Epistasis in 30%; structure predictions improved (AUC = 0.88) . (Roberts et al., 2023)

### FITNESS LANDSCAPE OF A DYNAMIC RNA STRUCTURE

GEO (GSE149234): Group I intron (119 nt) variants in yeast; 4,096 for splicing fitness. Conformational shifts yield ruggedness ($\rho= 0.35$); dynamic modeling ($\rho = 0.71$). (Soo et al., 2021)

### COMPREHENSIVE ANALYSIS OF RNA-PROTEIN INTERACTIONS BY HIGH-THROUGHPUT SEQUENCING-RNA AFFINITY PROFILING

From GEO (GSE54847), HiTS-RAP assays HuR binding to poly-A RNA library ( 10ˆ6 variants, 100 nt). Affinity scores via enrichment; 40% high-affinity motifs; achieved proteome-wide mapping ($K_d$ predictions, r=0.85). (Tome et al., 2014)

### HIGH-THROUGHPUT CELLULAR RNA DEVICE ENGINEERING

Sourced from GEO (GSE72890), this MPRA dataset engineers synthetic RNA devices (toehold switches, 80 nt) in mammalian cells for regulatory activity. 5,000 variants measured via luciferase; 25% orthogonal triggers; achieved 300-fold dynamic range in logic gates. (Townshend et al., 2015)

### SATURATION MUTAGENESIS MPRA OF MYC ENHANCER (RS6983267)

MaveDB (urn:mavedb:00000067-h): HEK293T MPRA of MYC enhancer ( 400 bp, rs6983267 locus); 2,048 variants for enhancer activity via H3K27ac/RNA-seq. 10% risk alleles boost; epistasis at SNPs; classified variants with OR=1.5. (Corces et al., 2018)

### TARGETED CHEMOTHERAPY DELIVERY TO TUMOR-INFILTRATING MYELOID CELLS USING RNA APTAMERS

This preclinical study used SELEX to identify four RNA aptamers with high specificity for tumor-infiltrating myeloid cells (TIMCs) in mouse and human tumors. Fitness was evaluated by conjugating the aptamers to doxorubicin, which enhanced drug delivery to tumor sites and led to significant tumor regression and increased survival in mouse models with minimal toxicity. A key feature is

the aptamers' specificity for TIMCs over their circulating counterparts. The aptamer-drug conjugates outperformed the clinically approved Doxil, demonstrating a promising strategy for targeted chemotherapy to the tumor microenvironment. (Fuente et al., 2020)

MACHINE LEARNING GUIDED APTAMER REFINEMENT AND DISCOVERY

This methodological paper presents a framework for accelerating aptamer discovery by integrating machine learning (ML) with SELEX data. The approach involves training ML models on sequence enrichment data from initial SELEX rounds to learn a comprehensive sequence-fitness landscape. Fitness, defined by binding affinity from sequencing counts, is then predicted for a vast space of unseen sequences, allowing the models to perform an "in silico" selection round. A key feature is the framework's ability to nominate novel, high-affinity aptamer candidates that were not present in the original experimental pool. The study successfully demonstrated that ML-guided design can identify refined aptamers with affinities comparable to or better than those from traditional SELEX, significantly reducing experimental effort and advancing rational aptamer engineering. (Bashir et al., 2021)

MUTATIONS PRIMARILY ALTER THE INCLUSION OF ALTERNATIVELY SPLICED EXONS

Sourced from GEO (GSE151942), this study combines deep mutagenesis of highly-included exons with transcriptome-wide analysis of natural genetic variation to investigate how mutations affect splicing. Fitness is measured as the exon inclusion level (Percent Spliced In, PSI), systematically assessing the impact of synonymous, non-synonymous, and intronic mutations. The central finding is that mutations very rarely alter the inclusion of exons that are already highly included in mature mRNAs. Instead, splice-altering effects are concentrated in and around alternatively spliced exons with intermediate inclusion levels. This non-uniform distribution of mutational effects across the transcriptome provides a critical framework for prioritizing synonymous and intronic variants as potential disease-causing mutations. (Baeza-Centurion et al., 2020)

EFFICIENT SELEX FOR DNA APTAMERS AGAINST BACTERIAL CELLS USING QPCR AND ULTRA-DEEP SEQUENCING

This methodological study presents a modernized SELEX protocol for efficiently generating DNA aptamers against whole bacterial cells. The workflow integrates quantitative real-time PCR (qPCR) to precisely monitor the enrichment of the aptamer pool during each selection round, serving as a clear fitness metric. The final selected pools are characterized by ultra-deep sequencing. Key features include real-time tracking of selection progress, which avoids common issues like over-amplification, and deep sequence analysis that reveals the population dynamics, identifies convergent aptamer families, and uncovers rare, high-affinity candidates. The work establishes a state-of-the-art pipeline that makes whole-cell SELEX more robust and insightful, accelerating the development of DNA aptamers for bacterial diagnostics and therapeutics. (Kolm et al., 2020)

RNA APTAMERS FOR TARGETED DELIVERY OF CARGO TO HUMAN $\beta$ CELLS

This study uses SELEX to identify RNA aptamers that specifically target human pancreatic $\beta$ cells by binding to two surface proteins: transmembrane p24 trafficking protein 6 (TMED6) and Clusterin. Fitness was evaluated by the aptamers' ability to bind, internalize, and deliver functional cargo to $\beta$ cells. The identified aptamers successfully delivered conjugated imaging reagents and therapeutic small interfering RNAs (siRNAs) to human $\beta$ cells in both *in vitro* and *ex vivo* models. A key feature of these aptamers is their high specificity for $\beta$ cells, enabling targeted delivery while sparing other cell types. The work establishes a novel platform for diagnosing $\beta$ cell loss and developing targeted RNA-based therapies for diseases like diabetes. (Van Simaeys et al., 2022)

OPTIMIZED LIGAND-GUIDED SELECTION FOR DISCOVERING DNA APTAMERS AGAINST THE T CELL RECEPTOR COMPLEX

This study introduces a comprehensive and optimized version of ligand-guided selection (LIGS), a variant of SELEX, to discover high-affinity DNA aptamers. The method targets the multi-component T cell receptor-cluster of differentiation epsilon (TCR-CD3$\epsilon$) complex in its native state on the surface

of human T cells. The LIGS process uses monoclonal antibodies for specific elution, and the resulting libraries are analyzed via high-throughput sequencing. Fitness is defined by binding affinity, with the work identifying five DNA aptamers with affinities ranging from 3.06 nM to 325 nM. A key feature is the rigorous validation of aptamer specificity using competitive binding analysis and a CRISPR-generated double-knockout cell line. The study establishes this modified LIGS strategy as a universal platform for efficiently identifying specific aptamers against complex cell-surface receptors. (Zumrut et al., 2019)

### DIFFERENTIAL BINDING CELL-SELEX METHOD TO IDENTIFY CELL-SPECIFIC APTAMERS

This methods paper describes a differential binding cell-SELEX protocol designed to isolate aptamers that can distinguish between closely related cell types. The workflow involves a positive selection step against a target cell line and a negative (or counter-selection) step against a non-target cell population to remove cross-reactive sequences. Fitness is defined by high binding affinity to the target cells coupled with low affinity for the non-target cells, with enrichment monitored by high-throughput sequencing. A key feature is the integrated differential selection strategy, which specifically enriches for aptamers recognizing unique cell-surface markers. The study establishes a robust methodology for generating highly specific aptamers, which are critical for developing targeted diagnostics and therapies that require precise cellular discrimination. (Pleiko et al., 2019)

### TIME-LAPSE IMAGING OF APTAMER EVOLUTION BY HIGH-THROUGHPUT SEQUENCING

This computational study introduces a method to reconstruct the evolutionary history of aptamer families using high-throughput sequencing (HTS) data from successive rounds of *in vitro* selection. By re-analyzing data from a SELEX experiment against Annexin A2, the authors construct an "empirical genealogical evolutionary (EGE) tree" to map the proliferation, mutation, and extinction of sequences over time. Evolutionary fitness is defined by a sequence's amplification and persistence across selection rounds. A key feature is the ability to trace ancestral relationships, which revealed that the final aptamer descended from a different, more abundant sequence from earlier rounds. The framework also successfully predicted improved aptamer variants. This work demonstrates that round-by-round HTS data can provide a "time-lapse image" of molecular evolution, offering deep insights into fitness landscapes and selection pressures. (Nguyen Quang et al., 2018)

### NOVEL APTAMERS FOR SPECIFIC RECOGNITION OF TRIPLE-NEGATIVE BREAST CANCER

This study uses Cell-SELEX on living cells to identify novel aptamers that specifically recognize Triple-Negative Breast Cancer (TNBC), an aggressive cancer subtype lacking common therapeutic targets. The selection protocol was designed to enrich for aptamers that bind to the surface of TNBC cells while showing minimal affinity for other cell types. Fitness was defined by high, specific binding to live TNBC cells, confirmed by downstream validation assays. The key feature of the identified aptamers is their ability to distinguish TNBC cells from non-cancerous or other breast cancer subtypes. This work provides new molecular tools for the development of targeted diagnostics and therapeutics for this challenging disease. (Camorani et al., 2020)

### USING MASKING PROBES FOR SELECTION OF AN NDM-1 SPECIFIC APTAMER

This methodological study presents a refined SELEX strategy to isolate a specific aptamer for the New Delhi metallo-beta-lactamase 1 (NDM-1) enzyme, a key driver of antibiotic resistance. The core innovation is the use of high-affinity polyhistidine binders as "masking probes" during the selection process. These probes block the His-tag on the recombinant NDM-1 protein, preventing the selection of non-specific, tag-binding aptamers. Fitness is thus defined by high-affinity binding to the native surface of the NDM-1 enzyme itself. This tag-masking approach effectively redirects selection pressure to the target of interest. The study successfully demonstrates a powerful and broadly applicable method to overcome a common challenge in SELEX, improving the specificity and quality of aptamers selected against tagged recombinant proteins. (Sabrowski et al., 2022)

### OPTIMIZED PERIPHERY-CORE INTERFACE INCREASES GLMS RIBOZYME FITNESS

This study investigates how interactions between the catalytic core and peripheral domains contribute to the fitness of the *Bacillus subtilis* glmS ribozyme. The work uses a high-throughput kinetic

assay (k-seq) to measure the cleavage activity of all single base substitutions across 152 sites. Fitness is quantified by *in vitro* catalytic rates, generating an activity map that closely mirrors phylogenetic conservation. The results show that most deleterious mutations impair ribozyme folding and self-assembly. A key feature from molecular dynamics simulations is the revelation that specific mutations introduce non-native tertiary interfaces that rewire and inactivate the catalytic center. The study concludes that avoiding non-native helix packing is a powerful constraint on RNA evolution, demonstrating the core-periphery interface is highly optimized to maintain function. (Yu et al., 2024)

## B    SELEX DATA PREPOSSESSING PROCEDURES

### B.1    DATA CURATION

All SELEX data are curated from GEO (Edgar et al., 2002) Bioproject database, with query "selex[All Fields]". This result in 4 processed dataset and 20 experiments with raw sequence count suitable for processing into valid datasets.

### B.2    INITIAL QUALITY ASSESSMENT

Raw sequencing data generated from 20 SELEX datasets were first subjected to a preliminary quality assessment. We employed `FastQC` to evaluate essential sequencing metrics, including base quality distribution, GC content, adapter contamination, and sequence duplication levels. This step allowed to identify potential experimental artifacts and ensure that the sequencing output met basic quality standards.

### B.3    QUALITY PREPROCESSING

High-throughput reads were further processed using `fastp (option: -q 20 -u 10 -3 20 -5 20 -M 20 -w 16 -n 0 -trim_poly_g)` to improve data reliability. For both paired-end and single-end libraries, preprocessing involved (i) trimming of low-quality bases and adapter sequences, (ii) removal of reads falling below quality thresholds, and (iii) merging of paired-end reads with sufficient overlap. The resulting dataset consisted of high-quality reads suitable for downstream analyses.

### B.4    SEQUENCE FREQUENCY STATISTICS

After preprocessing, sequence abundance statistics were calculated using `seqkit`. Command `seqkit fq2fa, seqkit seq(option -m <mode length> -M <mode length>)` and `seqkit rmdup(option: -s)` are used to calculate frequency for each sequence.

### B.5    SEQUENCE CLUSTERING

To reduce redundancy and group similar sequences, we applied `CD-HIT-est(option: -c 0.95 -n 8 -T 32)` Sequences meeting this similarity criterion were clustered together, and a representative sequence was selected from each cluster. The frequencies of all member sequences were aggregated to reflect the overall abundance of each representative sequence. By doing this, the amount of sequences fall into the range capable for all foundation models to embed in a reasonable period of time.

## C    DETAILED MODEL INFORMATION

### C.1    MODEL OVERVIEW

To ensure compatibility, we inspected the vocabulary of each model's tokenizer prior to inference. If the tokenizer included tokens for both Thymine (T) and Uracil (U), the sequences were input in their

original format. If the tokenizer supported only one format (*e.g.*, strictly DNA), we performed the necessary character conversion (*e.g.*, U to T) to match the model's pre-training data.

Table 7: Overview of baseline models in NABench

| Model | Params | Max Length | Tokenization | Architecture |
|---|---|---|---|---|
| LucaVirus(Pan et al., 2025) | 1.8B | 1280 | Single | BERT |
| Evo2-7B-base(Nguyen, 2025) | 7B | 8192 | Single | Hyena |
| Evo2-7B(Nguyen, 2025) | 7B | 131072 | Single | Hyena |
| Evo-1-8k(Merchant, 2024) | 6.45B | 8192 | Single | Hyena |
| Evo-1-8k-base(Merchant, 2024) | 6.45B | 131072 | Single | Hyena |
| GENA-LM(Fishman et al., 2025) | 336M | 512 | k-mer | BERT |
| N.T.v2(Dalla-Torre et al., 2025) | 500M | 2048 | k-mer | BERT |
| N.T.v2(Dalla-Torre et al., 2025) | 50M | 2048 | k-mer | BERT |
| CRAFTS(Wang et al., 2025a) | 161M | 1024 | Single | GPT |
| LucaOne(He et al., 2025) | 1.8B | 1280 | Single | BERT |
| AIDO.RNA(Song et al., 2024) | 1.6B | 1024 | Single | BERT |
| BiRNA-BERT(Tahmid et al., 2024) | 117M | dynamic | BPE | BERT |
| Evo-1.5(Merchant, 2024) | 6.45B | 131072 | Single | Hyena |
| GenSLM(Zvyagin et al., 2023) | 2.5B | 2048 | Codon | BERT |
| HyenaDNA(Nguyen et al., 2023) | 54.6M | 1M | Single | Hyena |
| N.T.(Dalla-Torre et al., 2025) | 500M | 1000 | k-mer | BERT |
| RFAMLlama(Sun et al.) | 88M | 2048 | Single | GPT |
| RNA-FM(Chen et al., 2022) | 99.52M | 1024 | Single | BERT |
| RNAErnie(Wang et al., 2024) | 105M | 1024 | Single | BERT |
| GenerRNA(Zhao et al., 2024) | 350M | dynamic | BPE | GPT |
| DNABERT(Ji et al., 2021) | 117M | dynamic | k-mer | BERT |
| RINALMo(Penić et al., 2024) | 650M | 1022 | Single | BERT |
| Enformer(Avsec et al., 2021) | 251M | 196608 | Single | BERT |
| SPACE(Yang et al., 2025) | 588M | 131072 | Single | BERT |
| GENERator(Wu et al., 2025) | 3B | 16384 | 6-mer | GPT |
| RESM(Zhang et al., 2025) | 150M | dynamic | Single | BERT |
| RESM(Zhang et al., 2025) | 650M | dynamic | Single | BERT |
| structRFM(Zhu et al., 2025) | 86M | 512 | Single | BERT |

## C.2 DETAILED INDROTUCTION OF ALL MODELS

**LucaVirus** is a unified nucleotide-protein language model pretrained on diverse viral genomes to predict evolutionary and functional landscapes, employing a BERT-based architecture with 1.8B parameters and single-nucleotide tokenization (max length 1280) under Apache-2.0 license. It integrates sequence and protein modalities via joint embeddings, enabling zero-shot virus classification and RNA virus genome reconstruction; training uses AdamW optimizer (batch size 512, learning rate 1e-4, 50 epochs), with inference via autoregressive sampling (temperature 1.0, top-k 4) on A100 GPUs. The source code of LucaVirus is available at https://github.com/LucaOne/LucaVirus.

**evo2-7B-base** is a 7 billion parameter DNA language model based on the StripedHyena 2 architecture (max length 8000 base pairs, DNA-specific tokenization) under an open license, pretrained on 8.8 trillion tokens from the OpenGenome2 dataset for genome modeling and design across all domains of life. Unique for interpretable sparse autoencoders identifying motifs, it supports sequence generation; trained via autoregressive approach using Savanna framework, inference uses temperature 1.0 and top-k 4 for generative tasks. The source code of evo2-7B-base is available at https://github.com/ArcInstitute/evo2.

**evo2-7B** extends the Evo 2 series with a larger StripedHyena model (7B parameters, max length 131072, single tokenization) under Apache-2.0, pretrained on 2.1T tokens for genome-scale design including RNA elements. Unique for interpretable sparse autoencoders capturing exon-intron motifs, it supports mRNA decay forecasting; trained with batch size 4.2M, learning rate 3e-4, and 500K

iterations via AdamW, inference uses temperature 1.0 and top-k 4 for generative tasks. The source code of evo2-7B is available at https://github.com/ArcInstitute/evo2.

**evo-1-8k** is a finetuned version of the Evo-1 series with StripedHyena architecture (6.45B parameters, max length 8192, single-nucleotide byte-level tokenization) under open license, finetuned on molecular-scale tasks including CRISPR and transposon prediction for biological sequence design and fitness forecasting. Unique for hybrid multi-head attention and gated convolutions enabling efficient long-context modeling; trained with mixed precision, inference uses temperature 1.0 and top-k 4 for generative tasks. The source code of evo-1-8k is available at https://github.com/evo-design/evo.

**evo-1-8k-base** is a the base pretrained model of the Evo-1 series with StripedHyena architecture (6.45B parameters, max length 8192, single-nucleotide byte-level tokenization) under open license, pretrained on 300 billion tokens from the OpenGenome dataset for molecular to genome scale foundation modeling. Unique for near-linear scaling of compute and memory with context length, robust to overtraining beyond compute-optimal frontiers; trained with mixed precision, inference uses temperature 1.0 and top-k 4 for generative tasks. The source code of evo-1-8k-base is available at https://github.com/evo-design/evo.

**genalm** (GENA-LM) is a BERT-variant DNA/RNA foundation model (336M parameters, dynamic max length via recurrent memory transformer, k-mer tokenization) under MIT, pretrained on 1T bp from multispecies genomes including yeast and Arabidopsis. It enables splicing predictions for RNA via transferable embeddings; hyperparameters: batch size 256, learning rate 1e-4 with AdamW; inference leverages CLS token pooling for classification and RMT for extended contexts. The source code of GENA-LM is available at https://github.com/AIRI-Institute/GENA_LM.

**N.T.v2** (Nucleotide Transformer v2) is an encoder-only transformer (500M parameters, max length 2048, k-mer tokenization) under CC 4.0, pretrained on 174B nucleotides from 850 genomes for human genomics tasks adaptable to RNA splicing. It doubles perceptual fields to 12kb for enhanced variant effect modeling; trained with batch size 512, learning rate 5e-5 to 1e-4 over 300B-1T tokens via AdamW, fine-tuning uses IA³ adapters in under 15 minutes. The source code of N.T.v2 is available at https://github.com/instadeepai/nucleotide-transformer.

**CRAFTS** is a 3D convolutional neural network model integrated with contrastive learning-based self-supervised pre-training for predicting RNA-small molecule binding affinities in RNA-targeted drug discovery. Unique for being the first 3D-CNN approach in this domain, extracting global pocket and local nucleotide features while supporting virtual screening; trained on processed PDBBind datasets using PyTorch, with batch size and learning rate not specified, via Adam optimizer. The source code of CRAFTS is available at https://github.com/SaisaiSun/RLaffinity.

**LucaOne** is a multimodal biological foundation model (1.8B parameters, max length 1280, single tokenization) under Apache-2.0, pretrained on sequences from 169,861 species spanning DNA/RNA/proteins. It unifies central dogma representations for cross-modal predictions; training: batch size 8 with gradient accumulation 32, learning rate 2e-4, 5.6M AdamW updates; inference applies max/value pooling for downstream tasks. The source code of LucaOne is available at https://github.com/LucaOne/LucaOne.

**AIDO.RNA** is a large-scale RNA encoder (1.6B parameters, max length 1024, single tokenization) under GenBio license, pretrained on 42M ncRNA sequences (30B nucleotides) for structure and function prediction. It achieves SOTA on 24/26 benchmarks via LoRA adaptations; hyperparameters: batch size 2M tokens, learning rate 5e-5 decaying to 1e-5 over 6 epochs with AdamW; inference fine-tunes in 10-15 epochs. The source code of AIDO.RNA is available at https://github.com/genbio-ai/AIDO.

**BiRNA-BERT** is an adaptive RNA transformer (117M parameters, dynamic max length, BPE tokenization) with no license, pretrained on 36M sequences (28B nucleotides) using dual nucleotide/BPE schemes. It optimizes efficiency for variable-length modeling; training: learning rate 2e-4, batch size 200 per device over 48 hours on 8 RTX 3090s with AdamW; inference dynamically adjusts tokenization. The source code of BiRNA-BERT is available at https://github.com/buetnlpbio/BiRNA-BERT.

**Evo-1.5** is a multimodal genomic model (6.45B parameters, max length 131072, single tokenization) under Apache-2.0, pretrained on 300B nucleotides from prokaryotic/phage sources for ncRNA fitness

tasks. It supports DNA/RNA/protein integration; hyperparameters: batch size 524K tokens, learning rate 9.7e-5 over 10 fine-tuning epochs; inference uses temperature 1.0 and top-k 4. The source code of Evo-1.5 is available at https://github.com/evo-design/evo.

**GenSLM** is a hierarchical transformer for genome-scale modeling (2.5B parameters, max length 2048, codon tokenization) under MIT, pretrained on 110M prokaryotic genes and fine-tuned on 1.5M SARS-CoV-2 genomes. It employs diffusion for long-range RNA interactions; training: batch size 4096, learning rate 5e-5 with variable steps via AdamW; inference applies reward-guided beam search. The source code of GenSLM is available at https://github.com/ramanathanlab/genslm.

**HyenaDNA** is a long-range genomic sequence model (54.6M parameters, max length up to 1M, single tokenization) under BSD 3-clause, pretrained on human genomes for sub-quadratic scaling in RNA-applicable tasks. It uses implicit long convolutions; hyperparameters: batch size 64-256, learning rate 1.5e-4 to 6e-4 over 10-20K steps; inference incorporates soft prompting. The source code of HyenaDNA is available at https://github.com/HazyResearch/hyena-dna.

**N.T.** (Nucleotide Transformer) is an encoder-only model (2.5B parameters, max length 1000, k-mer tokenization) under CC 4.0, pretrained on 174B nucleotides from 850 genomes for splicing and regulatory predictions. It captures multispecies diversity; training: batch size 512, learning rate 5e-5 to 1e-4 over 300B tokens with AdamW; fine-tuning via IA³ in under 15 minutes. The source code of N.T. is available at https://github.com/instadeepai/nucleotide-transformer.

**RFAMLLaMA** is a conditional RNA decoder (88M parameters, max length 2048, single tokenization) under CC 4.0, pretrained on 676K Rfam sequences with family-specific tags. It enables targeted generation; hyperparameters: learning rate 3e-4, weight decay 0.1 with AdamW; inference prompts with Rfam IDs for beam search. The source code of RFAMLlama is available at https://github.com/JinyuanSun/RFamLlama.

**RNA-FM** is a BERT-based ncRNA encoder (99.52M parameters, max length 1024, single tokenization) under MIT, pretrained on 23.7M sequences for general embeddings in structure/function tasks. It supports zero-shot adaptations; training details unspecified, inference via predict.py scripts with standard pooling. The source code of RNA-FM is available at https://github.com/ml4bio/RNA-FM.

**RNAErnie** is a structure-enhanced BERT model (105M parameters, max length 1024, single tokenization) under MIT, pretrained on 20.4M ncRNA sequences with base-pairing-biased attention. It predicts motifs via pairwise matrices; training: learning rate 1e-4, 20K warmup steps over 20 days on 24 V100s; inference extracts attention maps for zero-shot structure. The source code of RNAErnie is available at https://github.com/CatIIIIIIII/RNAErnie.

**GenerRNA** is a generative RNA transformer (350M parameters, dynamic max length, BPE tokenization) under MIT, pretrained on 16M sequences (11.6B nucleotides) for de novo design without structural priors. It uses causal decoding; hyperparameters: learning rate 1e-3 warmup to 1e-4 decay over 12 epochs; inference with top-k 250 random sampling. The source code of GenerRNA is available at https://github.com/pfnet-research/GenerRNA.

**DNABERT** is a bidirectional DNA encoder (117M parameters, dynamic max length, k-mer tokenization) under Apache-2.0, pretrained on human genomes for transferable genomic representations adaptable to RNA variants. It focuses on motif detection; training details limited, inference for effect scoring via MLM heads. The source code of DNABERT is available at https://github.com/jerryji1993/DNABERT.

**RINALMo** is a general-purpose RNA encoder (650M parameters, max length 1022, single tokenization) under Apache-2.0, pretrained on 36M ncRNA sequences for inter-family generalization in structure prediction. It uses vanilla BERT; training: batch size 192 per GPU, learning rate 5e-5 over 2 weeks on 7 A100s; inference with greedy decoding for base-pairing. The source code of RINALMo is available at https://github.com/lbcb-sci/RiNALMo.

**Enformer** is a convolutional-transformer hybrid (251M parameters, max length 196608, single tokenization) under MIT, trained on human/mouse epigenomics for gene expression predictions including RNA CAGE. It models 100kb contexts; hyperparameters: batch size 64, learning rate 5e-4 over 150K steps; inference with test-time augmentation. The source code of Enformer is

available at `https://github.com/google-deepmind/deepmind-research/tree/master/enformer`.

**SPACE** is a supervised DNA foundation model (588M parameters, max length 131072, single tokenization) under MIT, employing BERT-MoE architecture for multi-species genomic profile prediction via species-aware encoders and profile-grouped decoders. Pretrained on diverse epigenomic data, it captures regulatory dependencies for RNA-applicable tasks; training uses AdamW with sparse MoE routing (batch size unspecified, learning rate adaptive); inference aggregates expert weights for profile forecasting. The source code of SPACE is available at `https://github.com/ZhuJiwei111/SPACE`.

**GENERator** is a long-context generative genomic foundation model based on transformer decoder architecture (up to 3B parameters, max length 1M base pairs, 6-mer tokenization) under open license, pretrained on 386B tokens from RefSeq database for DNA sequence generation and optimization in eukaryotes and prokaryotes. Unique for adhering to the central dogma by generating protein-coding sequences analogous to known families and utilizing sliding-window attention for enhancer design; trained with configurable batch size and learning rate via distributed DDP/DeepSpeed/ FSDP on A100 GPUs, inference uses temperature 1.0 for generative tasks. The source code of GENERator is available at `https://github.com/GenerTeam/GENERator`.

**RESM** is a RNA language model extending the ESM series (up to 650M parameters, max length 4000 nucleotides) under MIT license, employing adapted transformer architecture with pseudo-protein mapping for transfer learning from protein models to capture RNA sequence-structure-function relationships. Unique for zero-shot dual-task excellence in structural and functional predictions, outperforming prior models with 81% accuracy gains on long RNAs; trained on noncoding RNA datasets, inference on CUDA GPUs with batch processing. The source code of RESM is available at `https://github.com/yikunpku/RESM`.

**structRFM** is a structure-guided RNA foundation model (parameters unspecified, max length unspecified, single tokenization) under CC-BY-NC 4.0, pretrained on 21M sequence-structure pairs using SgMLM with pair-matching and dynamic masking for joint sequential-structural learning. It produces versatile representations for zero-shot classification, structure prediction, and function inference; training balances masks via AdamW (details unspecified); fully open-source, deriving Zfold for 19% tertiary structure gains over AlphaFold3. The source code of structRFM is available at `https://github.com/heqin-zhu/structRFM`.

## D   DETAIL EXPLANATION ON EVALUATION AND METRICS

This section provides a detailed explanation of the metrics used to evaluate model performance across various tasks.

### D.1   SPEARMAN'S RANK CORRELATION COEFFICIENT ($\rho$)

#### D.1.1   FORMULA

The Spearman's correlation coefficient is calculated as:

$$\rho = 1 - \frac{6 \sum_{i=1}^{n} d_i^2}{n(n^2 - 1)}$$

where $d_i = \text{rank}(X_i) - \text{rank}(Y_i)$ is the difference between the ranks of the $i$-th corresponding predicted score and true experimental fitness value, and $n$ is the total number of variants.

#### D.1.2   SIGNIFICANCE

Spearman's correlation is a non-parametric measure of rank correlation. It assesses how well the relationship between two variables can be described using a monotonic function. In our context, it evaluates the model's ability to correctly rank the RNA variants according to their fitness, rather than predicting their exact fitness values. A value of $\rho = 1$ indicates a perfect monotonic relationship (the model ranks all variants in the correct order), while $\rho = -1$ indicates a perfect negative monotonic relationship, and $\rho = 0$ indicates no rank correlation. This metric is particularly suitable for evaluating

predictions on DMS (Deep Mutational Scanning) datasets, where the primary goal is to understand the relative fitness effects of mutations.

## D.2 NORMALIZED DISCOUNTED CUMULATIVE GAIN (NDCG)

### D.2.1 FORMULA

NDCG is calculated based on Discounted Cumulative Gain (DCG). For a ranked list of predictions up to position $p$, DCG is defined as:

$$\text{DCG}_p = \sum_{i=1}^{p} \frac{\text{rel}_i}{\log_2(i+1)}$$

where $\text{rel}_i$ is the relevance (true fitness value) of the item at rank $i$.

To obtain a score between 0 and 1, DCG is normalized by the Ideal DCG (IDCG), which is the DCG of the perfectly ranked list:

$$\text{NDCG}_p = \frac{\text{DCG}_p}{\text{IDCG}_p}$$

### D.2.2 SIGNIFICANCE

NDCG is a measure of ranking quality that emphasizes the importance of placing highly relevant items at the top of a list. By using a logarithmic discount factor, it penalizes misplacing high-fitness variants more heavily if they are ranked lower. This metric is crucial for evaluating whether a model can not only identify beneficial variants but also prioritize them correctly in its predictions, which directly reflects the model's utility in guiding experimental efforts toward the most promising candidates.

## D.3 AREA UNDER THE ROC CURVE (AUC)

### D.3.1 FORMULA

The Receiver Operating Characteristic (ROC) curve is a plot of the True Positive Rate (TPR) against the False Positive Rate (FPR) at various classification thresholds.

$$\text{TPR} = \frac{\text{TP}}{\text{TP} + \text{FN}}, \quad \text{FPR} = \frac{\text{FP}}{\text{FP} + \text{TN}}$$

AUC is the area under this curve:

$$\text{AUC} = \int_0^1 \text{TPR}(\text{FPR}^{-1}(x)) \, dx$$

where TP, TN, FP, and FN are True Positives, True Negatives, False Positives, and False Negatives, respectively.

### D.3.2 SIGNIFICANCE

AUC provides a single scalar value summarizing a model's performance as a binary classifier across all possible thresholds. It can be interpreted as the probability that the model will rank a randomly chosen positive instance higher than a randomly chosen negative one. This metric is particularly relevant for SELEX (Systematic Evolution of Ligands by Exponential Enrichment) datasets, where the task is to distinguish functional sequences from a vast library of random sequences. An AUC of 1.0 indicates a perfect classifier, while an AUC of 0.5 suggests performance equivalent to random guessing.

## D.4 MATTHEWS' CORRELATION COEFFICIENT (MCC)

### D.4.1 FORMULA

MCC is a robust metric for binary classification, calculated directly from the four values of the confusion matrix:

$$\text{MCC} = \frac{\text{TP} \times \text{TN} - \text{FP} \times \text{FN}}{\sqrt{(\text{TP} + \text{FP})(\text{TP} + \text{FN})(\text{TN} + \text{FP})(\text{TN} + \text{FN})}}$$

### D.4.2 SIGNIFICANCE

MCC is regarded as one of the most balanced classification metrics because it produces a high score only if the classifier obtains good results in all four confusion matrix categories (TP, TN, FP, FN). Its value ranges from -1 to +1, where +1 indicates a perfect prediction, 0 represents performance no better than random, and -1 indicates total disagreement between prediction and observation. Unlike metrics like accuracy or F1-score, MCC's score is high only when the prediction is correct in both identifying positive and negative classes, making it particularly useful for datasets with a significant class imbalance.

## D.5 POSITIVE SAMPLES DEFINATION

For DMS data, top 10% of sequences ranked by fitness score. For SELEX data, this refers to the sequences that account for the top 10% of the total read abundance (cumulative counts). We observed that SELEX data typically follows a heavy-tailed distribution containing extreme outliers, whereas the vast majority of sequences have read counts close to zero. Therefore, our strategy was to identify the dominant sequences that constitute the bulk of the pool's mass.

## D.6 COMPREHENSIVE RANKING SCORE

### D.6.1 DEFINITION

To provide a single, aggregated measure of a model's overall performance across all datasets and metrics, we define a comprehensive ranking score. The calculation follows a three-step process:

1. **Per-Assay, Per-Metric Ranking:** For each assay $a$ and each of the four evaluation metrics $k \in \{\rho, \text{NDCG}, \text{AUC}, \text{MCC}\}$, all models $m \in M$ are ranked based on their performance. This yields a rank $R(m, a, k) \in \{1, 2, \ldots, |M|\}$, where a lower rank value (*e.g.*, 1) indicates better performance.

2. **Rank Normalization:** To ensure ranks from different evaluations are comparable, each rank is normalized to a score between 0 and 1. The best-performing model (rank 1) receives a normalized score of 1, and the worst-performing model (rank $|M|$) receives a score of 0. The normalized rank $R_{\text{norm}}$ is calculated as:

$$R_{\text{norm}}(m, a, k) = \frac{|M| - R(m, a, k)}{|M| - 1}$$

3. **Final Score Aggregation:** The final comprehensive ranking score for a model $m$, denoted as $\text{Score}_{\text{final}}(m)$, is the average of its normalized ranks across all valid assays ($A_{\text{valid}}$) and all four metrics ($K$).

$$\text{Score}_{\text{final}}(m) = \frac{1}{|A_{\text{valid}}| \cdot |K|} \sum_{a \in A_{\text{valid}}} \sum_{k \in K} R_{\text{norm}}(m, a, k)$$

### D.6.2 SIGNIFICANCE

This aggregation method produces a robust and holistic final score for each model. It prevents a model's final standing from being skewed by exceptionally good or poor performance on a small subset of assays or a single metric. A higher final score indicates that a model exhibits consistently strong performance across the entire breadth of the benchmark, reflecting superior generalization capabilities.

# E  DMS EXPERIMENT RESULTS

In this part, all results of the DMS assays generated from our benchmarking experiments are presented.

## E.1  OVERALL DMS RESULTS

Table 8 presents the average Spearman's Correlation, AUC, MCC, and NDCG score for zero-shot tasks and Spearman's Correlation for few-shot, contiguous cross-validation and random cross-validation. The best score for each column is highlighted in bold and the second underlined. This table serves as the supplementary information for Figure 2

It can be observed that Evo models preform well in zero-shot settings, and BER-like models stands out in supervised and few-shot tasks. The latest model RESM, with transfer learning techniques, achieved decent scores under every evaluation settings, making it a wise choice in any DMS prediction task.

Table 8: Overall Scores of all models on DMS tasks

| Assay | Zeroshot | | | | Supervised | | |
| --- | --- | --- | --- | --- | --- | --- | --- |
| | Corr | AUC | MCC | NDCG | Contiguous | Random | Few-shot |
| RNA-FM | 0.148 | 0.541 | 0.068 | 0.380 | 0.233 | 0.544 | 0.183 |
| RNAernie | 0.078 | 0.481 | 0.044 | 0.353 | 0.199 | 0.387 | 0.102 |
| SPACE | 0.087 | 0.519 | 0.051 | 0.349 | 0.252 | 0.483 | 0.125 |
| AIDO.RNA | 0.098 | 0.472 | 0.049 | 0.324 | 0.207 | 0.486 | 0.201 |
| BiRNA-BERT | 0.093 | 0.518 | 0.042 | 0.369 | 0.167 | 0.493 | 0.186 |
| Crafts | 0.128 | 0.488 | 0.057 | 0.322 | 0.174 | 0.525 | 0.141 |
| Dnabert | 0.080 | 0.539 | 0.045 | 0.370 | 0.173 | 0.473 | 0.142 |
| Dnabert_6 | 0.069 | 0.482 | 0.035 | 0.343 | 0.163 | 0.322 | 0.111 |
| Enformer | 0.144 | 0.499 | 0.053 | 0.344 | 0.235 | 0.391 | 0.146 |
| Evo2-7B | 0.126 | 0.541 | 0.077 | 0.402 | 0.249 | 0.234 | 0.094 |
| Evo2-7B-base | 0.126 | 0.547 | 0.079 | **0.410** | 0.272 | 0.256 | 0.132 |
| Evo_1.5_8k_base | **0.177** | 0.543 | **0.085** | 0.396 | 0.216 | 0.417 | 0.147 |
| Evo_1_8k | 0.171 | 0.542 | 0.078 | 0.389 | 0.193 | 0.411 | 0.147 |
| Evo_1_8k_base | 0.171 | 0.542 | 0.078 | 0.389 | 0.193 | 0.411 | 0.147 |
| GENA-LM-large | 0.113 | 0.529 | 0.045 | 0.364 | 0.271 | 0.474 | 0.151 |
| GENERator-3B | 0.097 | 0.499 | 0.045 | 0.349 | 0.225 | 0.474 | 0.168 |
| GenerRNA | 0.079 | 0.493 | 0.043 | 0.357 | 0.172 | 0.536 | 0.146 |
| GenSlm | 0.082 | 0.482 | 0.040 | 0.331 | 0.241 | 0.447 | 0.154 |
| HyenaDNA | 0.094 | 0.483 | 0.047 | 0.334 | 0.186 | 0.561 | 0.132 |
| LucaOne | 0.090 | 0.486 | 0.057 | 0.349 | 0.225 | 0.563 | 0.163 |
| LucaVirus | 0.093 | 0.467 | 0.049 | 0.343 | 0.206 | 0.526 | 0.204 |
| N.T.-v2-50m | 0.097 | 0.512 | 0.044 | 0.349 | 0.220 | 0.465 | 0.110 |
| RESM | 0.172 | **0.557** | 0.084 | 0.408 | 0.251 | **0.579** | 0.190 |
| RiNALMo_giga | 0.105 | 0.488 | 0.050 | 0.338 | **0.272** | 0.513 | **0.209** |
| structRFM | 0.130 | 0.469 | 0.065 | 0.327 | 0.227 | 0.539 | 0.173 |

## E.2  LAYER-WISE REPRESENTATION ANALYSIS AND EXTRACTION STRATEGY

As claimed in Evo2 models (Nguyen, 2025) that intermediate layers sometimes offer a performance advantage over the final layer in embeddings, we conducted a comprehensive layer-wise ablation study on the full DMS dataset and a subset of SELEX dataset using the Evo2-7B-base model. As illustrated in Figure 6, the analysis reveals distinct behaviors across different evaluation protocols. For zero-shot inference (Figure 6, Left), performance exhibits notable fluctuation, reaching a peak at intermediate layers (*e.g.*, Layer 25, Spearman correlation $\approx 0.142$) before declining at the final layer ($\approx 0.135$). Conversely, for downstream supervised tasks where we perform random CV on several DMS and SELEX datasets (Figure 6, Right), performance shows a steady increase with

depth, saturating at the final layers (Spearman correlation > 0.57). While intermediate layers may capture specific zero-shot fitness signals more aggressively, this advantage appears highly volatile and layer-specific. In contrast, the final layer consistently maximizes performance for downstream predictive tasks, suggesting it aggregates the most robust and generalizable features. Consequently, we maintained a unified extraction protocol utilizing the final layer representations. This decision prioritizes the stability observed in the supervised benchmarks and avoids the hyper-parameter selection bias associated with cherry-picking intermediate layers based on fluctuating zero-shot scores. By strictly adhering to a model-agnostic, final-layer strategy, we ensure a fair, off-the-shelf comparison transferable to varying downstream applications.

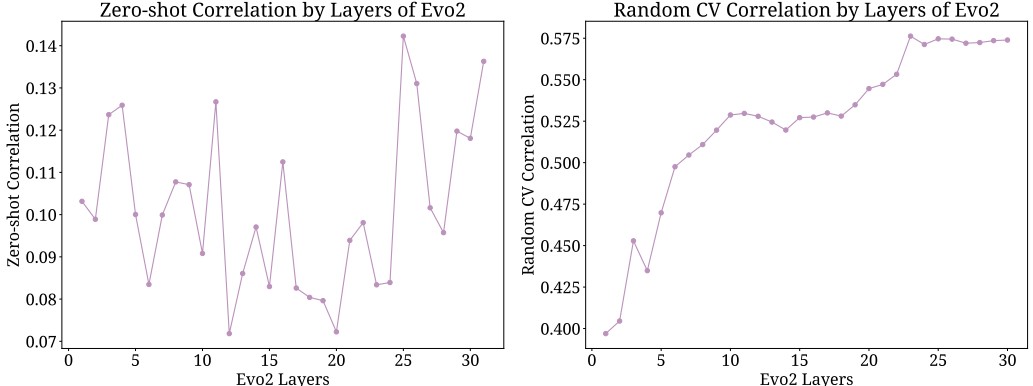

Figure 6: Layer-wise performance analysis of the Evo2 model.

### E.3 ZERO-SHOT RESULTS REGARDING GENOMIC TYPES

Table 9 serves as the supplementary data for Figure 3a, reporting model performance on types of DNA and RNA, with the sota model for each class in bold and second underlined. Figure 7 show that overall performances for all types of nucleic sequences is different and the patten exists for all types of architectures. This might indicate the amount of sequence data for training genomic foundation models is naturally imbalanced, making it more challenging to understand and predict some nucleic sequences than others.

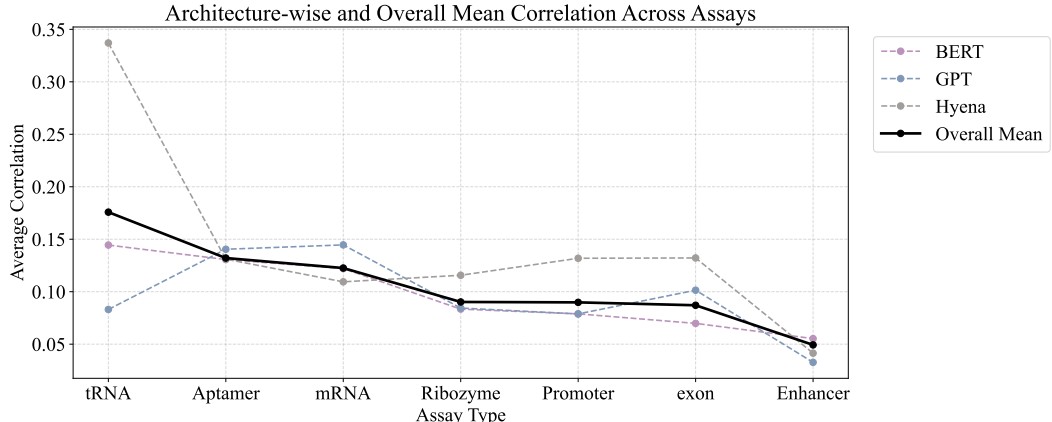

Figure 7: **Zero-shot Results**: The performance of all models on different RNA types.

Table 9: Zero-shot Spearman's $\rho$ results on different nucleotide types

| Model | mRNA | ribozyme | tRNA | aptamer | promoter | enhancer |
|---|---|---|---|---|---|---|
| RNA-FM | **0.257** | 0.103 | 0.357 | 0.167 | 0.098 | 0.011 |
| LucaOne | 0.249 | 0.085 | 0.076 | 0.112 | 0.017 | 0.074 |
| RESM | 0.241 | **0.155** | 0.381 | 0.105 | 0.101 | 0.049 |
| Enformer | 0.232 | 0.128 | 0.052 | 0.197 | 0.190 | **0.216** |
| GENERator-3B | 0.205 | 0.082 | 0.049 | 0.145 | 0.137 | 0.054 |
| structRFM | 0.196 | 0.089 | 0.196 | 0.181 | 0.161 | 0.113 |
| SPACE | 0.059 | 0.060 | 0.089 | 0.111 | 0.346 | 0.170 |
| N.T.-v2-50m | 0.068 | 0.083 | 0.381 | 0.092 | **0.384** | 0.064 |
| Evo_1.5_8k_base | 0.143 | 0.139 | **0.404** | 0.180 | 0.178 | 0.034 |
| Evo2-7B | 0.123 | 0.101 | 0.354 | 0.066 | 0.065 | 0.068 |
| Crafts | 0.122 | 0.132 | 0.089 | **0.215** | 0.080 | 0.064 |
| Evo_1_8k_base | 0.111 | 0.128 | 0.398 | 0.201 | 0.177 | 0.030 |
| Evo_1_8k | 0.111 | 0.128 | 0.398 | 0.201 | 0.177 | 0.030 |
| BiRNA-BERT | 0.109 | 0.102 | 0.059 | 0.089 | 0.118 | 0.045 |
| RNAernie | 0.101 | 0.078 | 0.103 | 0.080 | 0.050 | 0.040 |
| Evo2-7B-base | 0.087 | 0.093 | 0.392 | 0.079 | 0.065 | 0.065 |
| GENA-LM-large | 0.041 | 0.072 | 0.089 | 0.158 | 0.178 | 0.078 |
| HyenaDNA | 0.082 | 0.111 | 0.075 | 0.058 | 0.130 | 0.023 |
| RiNALMo_giga | 0.087 | 0.084 | 0.044 | 0.128 | 0.153 | 0.087 |
| GenerRNA | 0.216 | 0.083 | 0.089 | 0.171 | 0.075 | 0.006 |
| Dnabert | 0.074 | 0.049 | 0.104 | 0.208 | 0.067 | 0.038 |
| AIDO.RNA | 0.069 | 0.109 | 0.071 | 0.140 | 0.059 | 0.057 |
| N.T.-500m | 0.116 | 0.082 | 0.060 | 0.001 | 0.105 | |
| LucaVirus | 0.055 | 0.103 | 0.060 | 0.168 | 0.033 | 0.037 |
| N.T.-v2-500m | 0.074 | 0.064 | – | 0.081 | 0.091 | 0.006 |
| GenSLM | 0.030 | 0.057 | – | 0.208 | 0.057 | 0.059 |
| Dnabert_6 | 0.027 | 0.061 | 0.059 | 0.121 | 0.103 | 0.039 |

### E.4 ZERO-SHOT RESULTS REGARDING EXPERIMENTAL ENVIRONMENT

To address potential performance variations across species categories, we further conducted a Stratified Performance Analysis using the Evo2 model. As shown in Table 11, the results reveal that models tend to perform better on natural datasets. This granular analysis also profoundly reveals the generalization boundaries of current foundation models across taxonomic groups, providing a vital reference for the nucleic acid research community regarding model applicability in specific biological domains.

### E.5 ASSAY-LEVEL RESULTS

The ranking varies a lot when tested on different benchmarks, indicating the models might have different knowledge on each type. This is especially true for mRNA, probably because mRNA functions as encoding proteins while all others are non-coding gene sequence, aiming for different functions beyond serving as transcripts.

In Figure 8, results are visualized in term of assays. The distribution varies a lot across different experiment, indicating difficulty is not similar.

### E.6 COMPLETE RESULTS FOR SUPERVISED TASKS

In supervised learning, ridge model is selected as a probe for the regression. First, regarding the Supervised and Few-shot tasks, the scale of training samples varies significantly across different assays. Particularly in Few-shot scenarios, the available training data is extremely limited. Under such conditions, directly applying full-model fine-tuning carries significant risks. With extremely scarce training data, the model may fail to converge or, worse, suffer from catastrophic forgetting,

Table 10: Zero-shot AUC results on different nucleotide types

| Model | mRNA | ribozyme | tRNA | aptamer | promoter | enhancer |
|-------|------|----------|------|---------|----------|----------|
| RNA-FM | 0.638 | 0.559 | 0.682 | 0.578 | 0.555 | 0.529 |
| LucaOne | **0.638** | 0.550 | 0.549 | 0.549 | 0.563 | 0.544 |
| RESM | 0.626 | 0.584 | 0.710 | 0.569 | 0.554 | 0.531 |
| Enformer | 0.626 | 0.571 | 0.535 | 0.599 | 0.616 | 0.640 |
| GENERator-3B | 0.630 | 0.544 | 0.535 | 0.567 | 0.567 | 0.527 |
| structRFM | 0.614 | 0.549 | 0.619 | 0.595 | 0.581 | 0.522 |
| SPACE | 0.583 | 0.530 | 0.533 | 0.571 | 0.644 | **0.648** |
| N.T.-v2-50m | 0.526 | 0.554 | 0.710 | 0.560 | **0.676** | 0.540 |
| Evo_1.5_8k_base | 0.618 | 0.569 | **0.718** | 0.578 | 0.604 | 0.526 |
| Evo2-7B | 0.575 | 0.553 | 0.690 | 0.541 | 0.537 | 0.554 |
| Crafts | 0.584 | 0.570 | 0.546 | **0.621** | 0.545 | 0.532 |
| Evo_1_8k_base | 0.602 | 0.562 | 0.715 | 0.588 | 0.601 | 0.539 |
| Evo_1_8k | 0.602 | 0.562 | 0.715 | 0.588 | 0.601 | 0.539 |
| BiRNA-BERT | 0.561 | 0.556 | 0.533 | 0.559 | 0.546 | 0.532 |
| RNAernie | 0.576 | 0.532 | 0.562 | 0.558 | 0.540 | 0.527 |
| Evo2-7B-base | 0.553 | 0.554 | 0.713 | 0.545 | 0.551 | 0.537 |
| GENA-LM-large | 0.552 | 0.541 | – | 0.570 | 0.592 | 0.561 |
| HyenaDNA | 0.558 | 0.556 | 0.546 | 0.514 | 0.571 | 0.511 |
| RiNALMo_giga | 0.517 | 0.541 | 0.533 | 0.556 | 0.582 | 0.519 |
| GenerRNA | 0.615 | 0.528 | 0.619 | 0.603 | 0.519 | 0.516 |
| Dnabert | 0.538 | 0.525 | 0.561 | 0.589 | 0.530 | 0.547 |
| AIDO.RNA | 0.591 | 0.554 | 0.540 | 0.578 | 0.546 | 0.551 |
| LucaVirus | 0.531 | 0.556 | 0.549 | 0.563 | 0.523 | 0.527 |
| N.T.-v2-500m | 0.521 | 0.537 | – | 0.524 | 0.573 | 0.528 |
| GenSLM | 0.518 | 0.528 | 0.533 | 0.583 | 0.518 | 0.552 |
| Dnabert_6 | 0.518 | 0.535 | 0.530 | 0.559 | 0.536 | 0.535 |

Table 11: Correlation by Species

| Species | Correlation |
|---------|-------------|
| Eukaryote | 0.223 |
| Prokaryote | 0.194 |
| Synthetic | 0.104 |

thereby destroying the rich feature representations of the original pre-trained model. This instability is the primary reason we opted against full-model fine-tuning. Second, to ensure robust performance, we conducted a comparative analysis of multiple Supervised Probes (prediction heads) within our DMS supervised learning experiments. The results of this evaluation are presented in Table 12.

Table 12: Model Performance

| Model | Spearman Correlation |
|-------|----------------------|
| RidgeCV | 0.6171 |
| LinearSVR | 0.5405 |
| RandomForest | 0.5296 |
| MLP | 0.5190 |
| XGBoost | 0.5059 |
| LinearRegression | 0.4969 |
| LassoCV | 0.4812 |

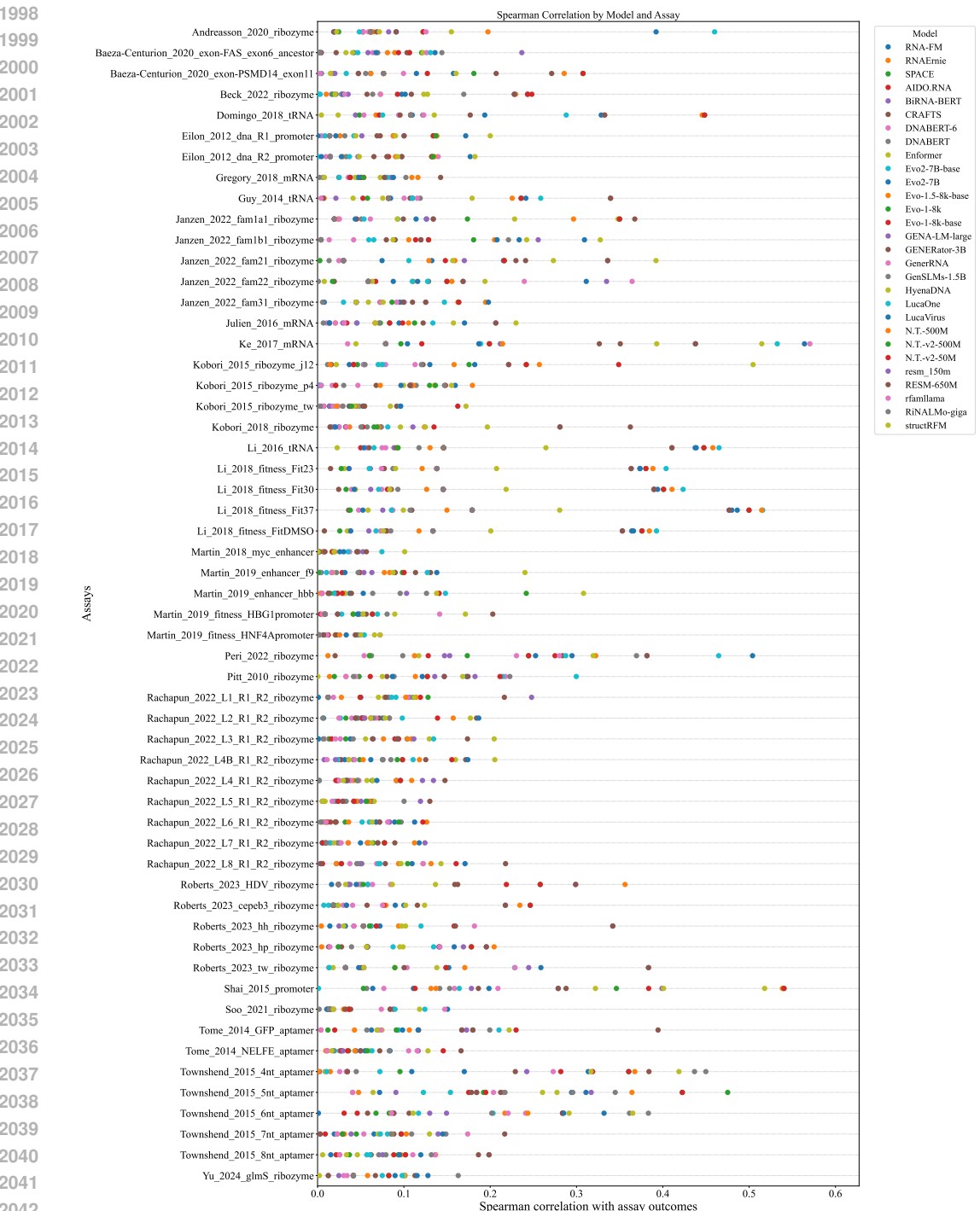

Figure 8: **Zero-shot Results**: The performance of models on all assays.

In Figure 9, the Spearman's $\rho$ is visualized for all models on zero-shot, few-shot, random CV and contiguous CV tasks.

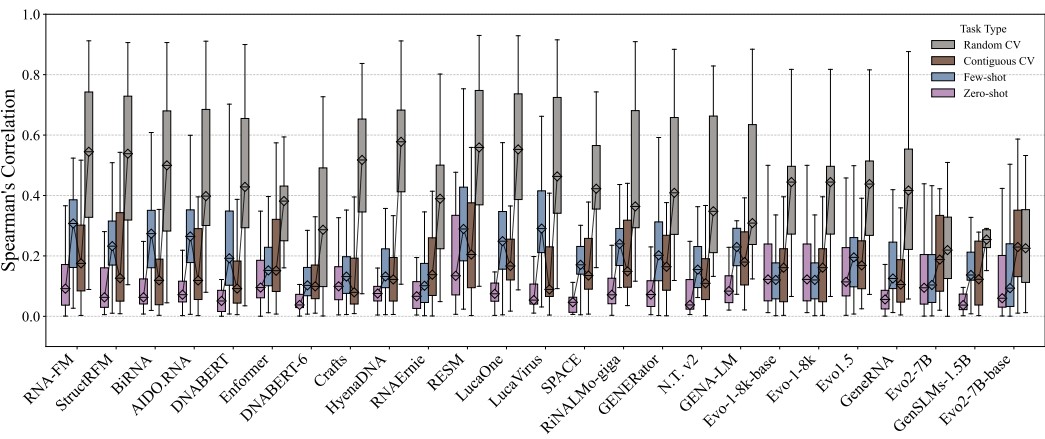

Figure 9: **Supervised Results**: The average spearman $\rho$ of models in supervised tasks.

## E.7 NUCLEIC TYPE LEVEL RESULTS FOR FEW-SHOT LEARNING AND SUPERVISED LEARNING

Here we have list the results for random cross-validation, contiguous cross-validation and few-shot learning, in terms of each type of sequences.

Table 13: Few-shot Spearman's $\rho$ on different nucleotide types

| Model | mRNA | ribozyme | tRNA | aptamer | promoter | enhancer |
|---|---|---|---|---|---|---|
| RNA-FM | 0.318 | 0.164 | 0.232 | 0.248 | 0.159 | 0.009 |
| LucaOne | 0.261 | 0.169 | 0.141 | 0.199 | 0.127 | 0.037 |
| RESM | 0.339 | 0.159 | 0.329 | 0.209 | 0.138 | 0.055 |
| Enformer | 0.347 | 0.117 | 0.089 | 0.206 | 0.184 | **0.149** |
| GENERator-3B | 0.302 | 0.137 | 0.224 | 0.227 | 0.143 | 0.095 |
| structRFM | 0.276 | 0.149 | 0.273 | 0.181 | 0.104 | 0.113 |
| SPACE | 0.255 | 0.080 | 0.130 | 0.241 | 0.114 | 0.158 |
| N.T.-v2-50m | 0.214 | 0.090 | **0.327** | 0.230 | 0.061 | 0.026 |
| Evo_1.5_8k_base | 0.294 | 0.153 | 0.110 | 0.217 | 0.052 | 0.039 |
| Evo2-7B | 0.207 | 0.079 | 0.144 | 0.095 | 0.067 | 0.048 |
| Crafts | 0.345 | 0.129 | 0.117 | 0.217 | 0.075 | 0.040 |
| Evo_1_8k_base | 0.300 | 0.142 | 0.156 | 0.209 | 0.052 | 0.027 |
| Evo_1_8k | 0.300 | 0.142 | 0.156 | 0.209 | 0.052 | 0.027 |
| BiRNA-BERT | 0.305 | 0.175 | 0.165 | 0.264 | 0.181 | 0.043 |
| RNAernie | 0.072 | 0.102 | 0.100 | 0.142 | 0.110 | 0.026 |
| Evo2-7B-base | 0.269 | 0.107 | 0.300 | 0.068 | 0.064 | 0.098 |
| GENA-LM-large | **0.349** | 0.127 | 0.131 | 0.248 | 0.110 | 0.060 |
| HyenaDNA | 0.298 | 0.130 | 0.058 | 0.216 | 0.079 | 0.050 |
| RiNALMo_giga | 0.379 | 0.184 | 0.273 | 0.240 | **0.174** | 0.107 |
| GenerRNA | 0.216 | 0.149 | 0.099 | 0.215 | 0.111 | 0.019 |
| Dnabert | 0.184 | 0.150 | 0.068 | 0.212 | 0.108 | 0.093 |
| AIDO.RNA | 0.297 | 0.201 | 0.256 | 0.232 | 0.126 | 0.033 |
| LucaVirus | 0.291 | 0.224 | 0.121 | **0.294** | 0.131 | 0.033 |
| N.T.-v2-500m | 0.256 | 0.135 | – | 0.166 | 0.071 | 0.053 |
| GenSLM | 0.197 | 0.149 | 0.099 | 0.299 | 0.115 | 0.019 |
| Dnabert_6 | 0.084 | 0.088 | 0.105 | 0.255 | 0.112 | 0.027 |

Table 14: The Spearman's $\rho$ of random cross validation on different nucleotide types

| Model | mRNA | ribozyme | tRNA | aptamer | promoter | enhancer |
|---|---|---|---|---|---|---|
| RNA-FM | 0.597 | 0.556 | 0.587 | 0.530 | 0.582 | 0.256 |
| LucaOne | 0.670 | 0.558 | 0.571 | **0.572** | **0.623** | 0.353 |
| RESM | 0.679 | 0.587 | 0.595 | 0.589 | 0.610 | 0.291 |
| Enformer | 0.488 | 0.352 | 0.353 | 0.403 | 0.554 | 0.452 |
| GENERator-3B | 0.607 | 0.449 | 0.551 | 0.476 | 0.555 | 0.253 |
| structRFM | 0.596 | 0.531 | 0.588 | 0.541 | 0.610 | 0.275 |
| SPACE | 0.540 | 0.467 | 0.457 | 0.474 | 0.612 | 0.442 |
| N.T.-v2-50m | 0.570 | 0.464 | 0.470 | 0.462 | 0.632 | 0.148 |
| Evo_1.5_8k_base | 0.491 | 0.412 | 0.494 | 0.419 | 0.443 | 0.169 |
| Evo2-7B | 0.429 | 0.202 | 0.393 | 0.172 | 0.249 | 0.092 |
| Crafts | 0.575 | 0.541 | 0.553 | 0.566 | 0.516 | 0.189 |
| Evo_1_8k_base | 0.478 | 0.409 | 0.503 | 0.390 | 0.444 | 0.139 |
| Evo_1_8k | 0.478 | 0.409 | 0.503 | 0.390 | 0.444 | 0.139 |
| BiRNA-BERT | 0.620 | 0.488 | 0.563 | 0.458 | 0.579 | 0.181 |
| RNAernie | 0.461 | 0.394 | 0.464 | 0.390 | 0.348 | 0.129 |
| Evo2-7B-base | 0.413 | 0.212 | 0.421 | 0.190 | 0.293 | 0.217 |
| GENA-LM-large | 0.594 | 0.475 | 0.470 | 0.444 | 0.604 | 0.217 |
| HyenaDNA | 0.599 | 0.556 | 0.571 | 0.562 | 0.576 | 0.238 |
| RiNALMo_giga | 0.636 | 0.512 | 0.494 | 0.493 | 0.681 | 0.268 |
| GenerRNA | 0.640 | 0.548 | 0.519 | 0.558 | 0.559 | 0.277 |
| Dnabert | **0.630** | 0.460 | 0.543 | 0.426 | 0.579 | 0.206 |
| AIDO.RNA | 0.565 | 0.495 | 0.549 | 0.480 | 0.481 | 0.200 |
| LucaVirus | 0.651 | 0.529 | 0.565 | 0.493 | 0.588 | 0.260 |
| N.T.-v2-500m | 0.560 | 0.429 | 0.433 | 0.360 | 0.636 | 0.210 |
| GenSLM | 0.557 | 0.452 | 0.437 | 0.442 | 0.515 | 0.212 |
| Dnabert_6 | 0.481 | 0.302 | 0.425 | 0.263 | 0.409 | 0.097 |

### E.8 ASSAY LEVEL RESULTS

In Figure 10, a clear improvement can be witnessed for some assays, all models see a rise in performance in random CV task, but for some assays the distribution of performance remains low, indicating these assay is difficult or unsuitable for fitness prediction, and might be the challenge to be tackled for next generation nucleotide foundation models.

### E.9 TRANSFER LEARNING FOR DMS DATASETS

We present the full data from our expanded Out-of-Distribution experiments below, covering both Cross-Assay (DMS $\leftrightarrow$ SELEX) and Cross-Family (DNA $\leftrightarrow$ RNA varieties and different RNA families) scenarios.

We conducted transfer learning experiments in the DMS dataset using both RESM and Lucaone model (see Figure 12 and Figure 13). Our results indicate that both models exhibit higher transferability when trained and tested on assays from the same study compared to those across different studies. Notably, RESM demonstrates this trend more distinctly, showing superior generalization and capacity for cross-dataset knowledge acquisition compared to Lucaone.

## F SELEX EXPERIMENT RESULTS

### F.1 OVERALL SELEX RESULTS

Table 16 reports the Spearman's $\rho$, AUC, MCC and NDCG scores for zero-shot task on SELEX datasets and AUC scores for random and few-shot tasks. Note that since SELEX experiments do

Table 15: Contiguous Cross Validation results on different tasks

| Model | mRNA | ribozyme | tRNA | aptamer | promoter | enhancer |
|---|---|---|---|---|---|---|
| RNA-FM | 0.259 | 0.242 | 0.345 | 0.115 | 0.035 | 0.095 |
| LucaOne | 0.304 | 0.242 | 0.215 | 0.111 | 0.164 | 0.158 |
| RESM | 0.200 | 0.232 | 0.448 | 0.221 | 0.169 | 0.123 |
| Enformer | 0.278 | 0.252 | 0.152 | 0.046 | 0.318 | 0.336 |
| GENERator-3B | 0.290 | 0.221 | 0.275 | 0.170 | 0.172 | 0.145 |
| structRFM | 0.136 | 0.217 | 0.375 | 0.281 | 0.033 | 0.087 |
| SPACE | 0.112 | 0.309 | 0.170 | 0.128 | – | **0.344** |
| N.T.-v2-50m | 0.073 | 0.300 | – | 0.056 | 0.220 | 0.116 |
| Evo_1.5_8k_base | 0.227 | 0.264 | 0.222 | 0.077 | 0.037 | 0.047 |
| Evo2-7B | 0.221 | 0.284 | 0.295 | 0.156 | 0.043 | 0.110 |
| Crafts | 0.124 | 0.245 | 0.069 | 0.173 | 0.029 | 0.049 |
| Evo_1_8k_base | 0.172 | 0.239 | 0.166 | 0.118 | 0.107 | 0.037 |
| Evo_1_8k | 0.172 | 0.239 | 0.166 | 0.118 | 0.107 | 0.037 |
| BiRNA-BERT | 0.238 | 0.193 | 0.163 | 0.095 | 0.031 | 0.057 |
| RNAernie | 0.247 | 0.237 | 0.185 | 0.076 | 0.081 | 0.074 |
| Evo2-7B-base | 0.176 | 0.317 | 0.310 | 0.125 | 0.128 | 0.138 |
| GENA-LM-large | 0.105 | 0.369 | 0.270 | 0.203 | 0.592 | 0.062 |
| HyenaDNA | 0.232 | 0.215 | 0.187 | 0.143 | 0.037 | 0.049 |
| RiNALMo_giga | 0.093 | 0.356 | 0.379 | 0.121 | – | 0.123 |
| GenerRNA | 0.103 | 0.317 | 0.252 | 0.081 | 0.171 | 0.024 |
| Dnabert | 0.167 | 0.213 | 0.159 | 0.080 | 0.049 | 0.064 |
| AIDO.RNA | 0.143 | 0.219 | 0.309 | **0.218** | 0.095 | 0.015 |
| LucaVirus | 0.224 | 0.276 | 0.215 | 0.139 | 0.164 | 0.075 |
| GenSLM | 0.103 | 0.317 | 0.252 | 0.081 | 0.171 | 0.024 |
| Dnabert_6 | 0.121 | 0.220 | 0.064 | 0.173 | 0.068 | 0.074 |

not have the concept of wild-type sequence, contiguous cross validation is not a meaningful task for SELEX datasets.

In Figure 14, the distribution of the 4 metrics for SELEX experiemts are visualized. From the figure we can conclude that zero-shot prediction for randomly synthesized library remains a tough challenge for nucleotide foundation models, as for now no model really generate meaningful results.

## F.2 ZERO-SHOT REUSLTS ON 4 METRICS

## F.3 TRANSFER LEARNING FOR SELEX EXPERIMENTS

In Figure 15, we evaluate transfer learning on four models with varying performance in the SELEX random cross-validation task. In each iteration, one SELEX assay is used for training while the remaining assays serve as test sets. With the x-axis representing the training set and the y-axis the test set, HyenaDNA and LucaOne display consistent patterns across all rounds of assays within the same SELEX experiment, whereas the other two models produce more scattered results. These findings suggest that the bottom two models—particularly LucaOne—are able to generalize across assays, capturing underlying knowledge and effectively applying it to related tasks.

## G ETHICS STATEMENT

The work reported in this paper involves only computational experiments and publicly available datasets. Therefore, no ethical approval was necessary.

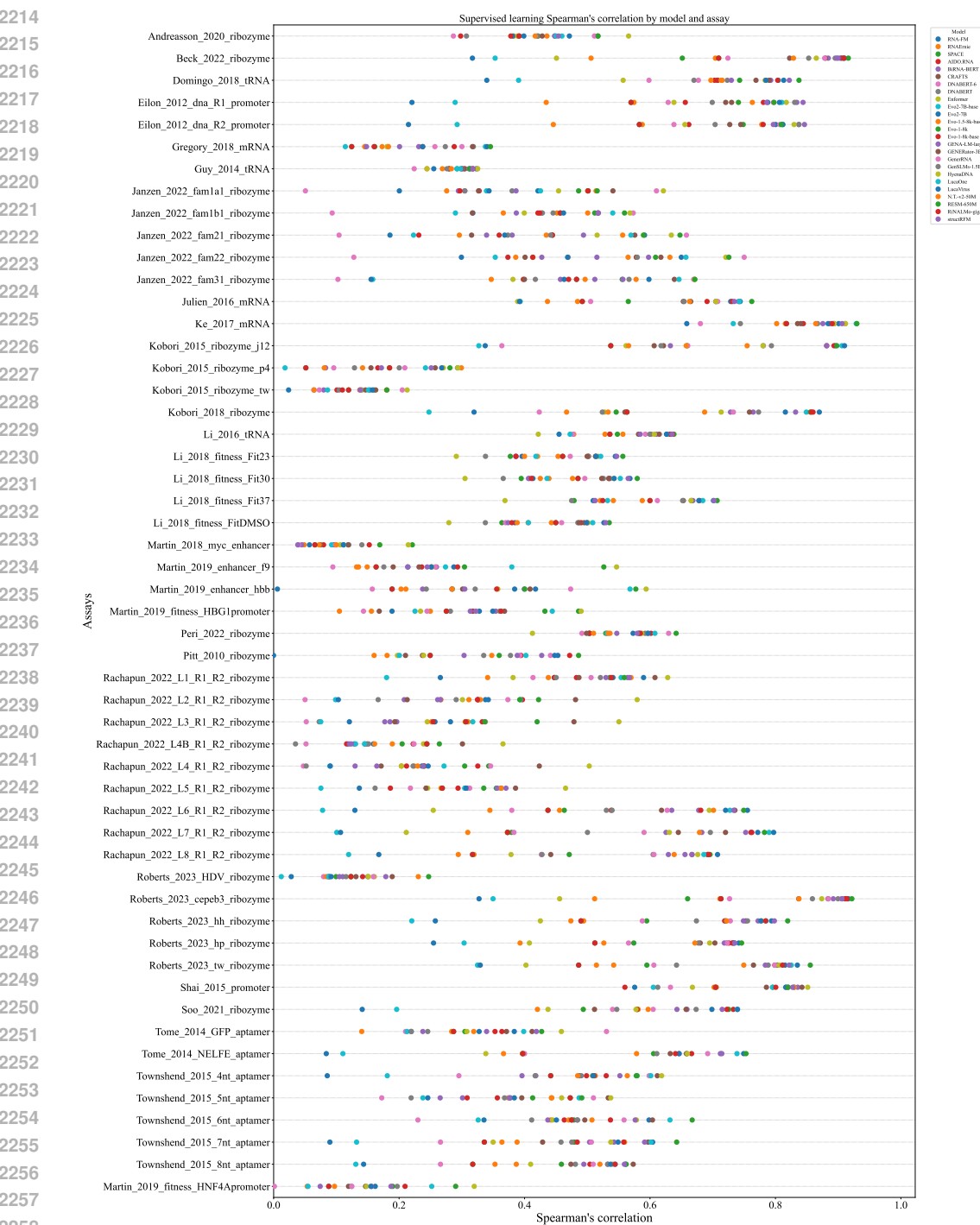

Figure 10: **Supervised Learning Results**: The performance of all models on different assays.

## H REPRODUCIBILITY STATEMENT

The datasets and code for NABench are available at https://anonymous.4open.science/r/NABench-20CB

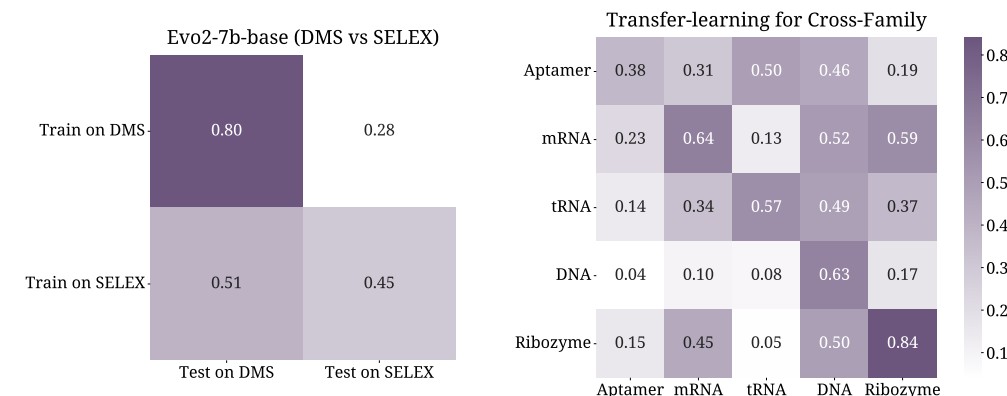

Figure 11: Cross-Assay and Cross-Family Transfer learning for DMS datasets

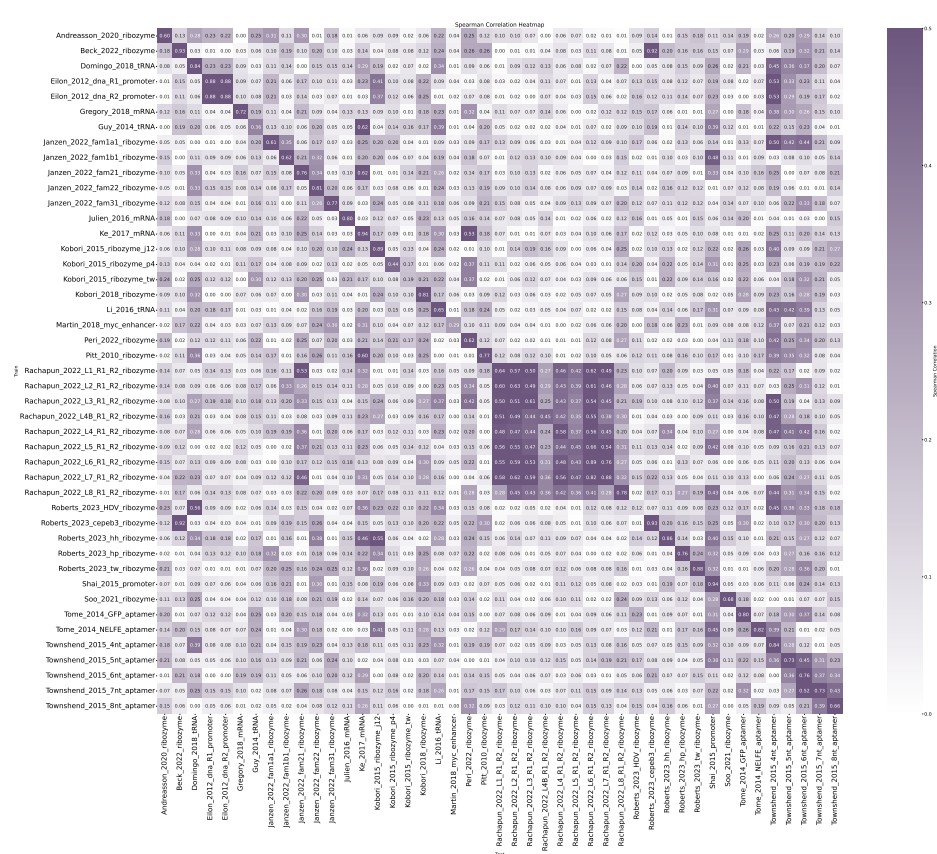

Figure 12: Single assay level transfer learning for RESM

# I    LLM USAGE STATEMENT

We use the Large Language Models (LLMs) for grammar checking of the paper to improve its overall readability, while all other work is independently completed by the human authors.

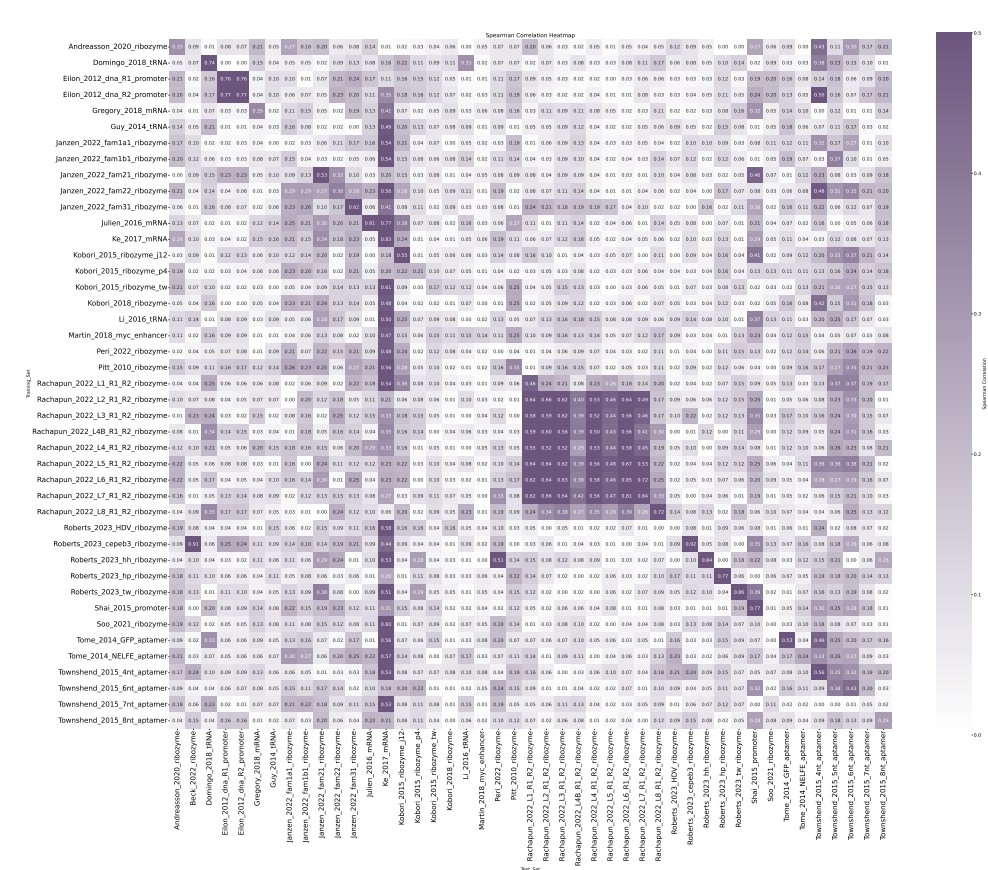

Figure 13: Single assay level transfer learning for LucaOne

Zero-shot results of all models in SELEX fitness prediction

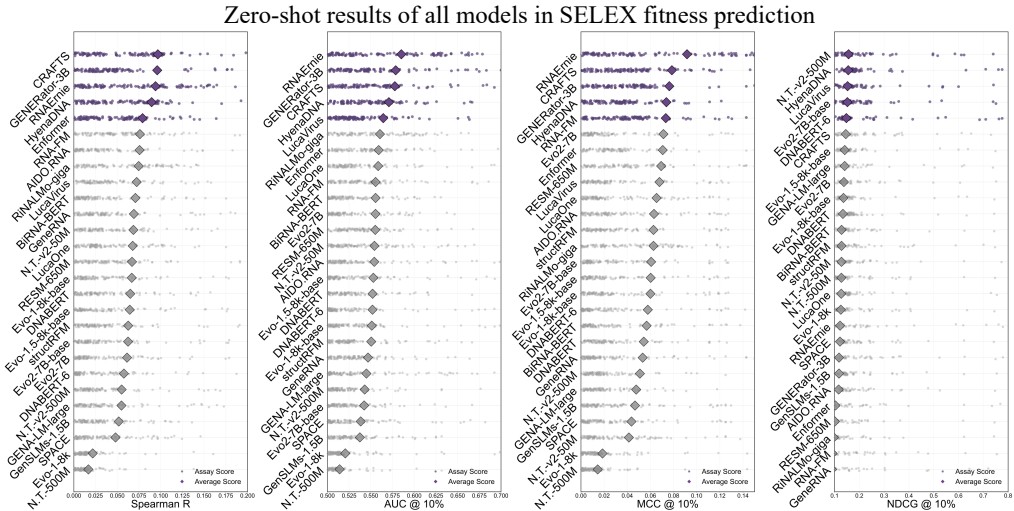

Figure 14: **Zero-shot Results**: Zero-shot results of all models in SELEX fitness prediction.

Table 16: Overall Scores of all models on SELEX tasks

| Model | Zeroshot | | | | Supervised | |
|---|---|---|---|---|---|---|
| | Corr. | AUC | MCC | NDCG | Random | Few-shot |
| RNA-FM | 0.085 | 0.55 | 0.05 | 0.194 | 0.685 | **0.617** |
| RNAernie | 0.069 | 0.553 | 0.052 | 0.201 | 0.637 | 0.573 |
| SPACE | 0.091 | 0.557 | 0.06 | **0.396** | 0.608 | 0.568 |
| AIDO.RNA | 0.08 | 0.547 | 0.045 | 0.266 | 0.687 | 0.599 |
| BiRNA-BERT | 0.076 | 0.546 | 0.041 | 0.3 | 0.681 | 0.588 |
| Crafts | 0.104 | 0.56 | 0.054 | 0.283 | 0.654 | 0.571 |
| Dnabert | 0.067 | 0.534 | 0.044 | 0.305 | 0.669 | 0.582 |
| Dnabert_6 | 0.06 | 0.533 | 0.035 | 0.293 | 0.606 | 0.544 |
| Enformer | 0.088 | 0.557 | 0.041 | 0.187 | 0.574 | 0.574 |
| Evo2-7B | 0.101 | 0.556 | 0.062 | 0.321 | 0.597 | 0.554 |
| Evo2-7B-base | 0.091 | 0.555 | 0.064 | 0.328 | 0.596 | 0.559 |
| Evo_1.5_8k_base | 0.136 | 0.572 | 0.065 | 0.324 | 0.635 | 0.568 |
| Evo_1_8k | **0.142** | **0.574** | **0.067** | 0.344 | **0.738** | 0.585 |
| Evo_1_8k_base | 0.133 | 0.57 | 0.063 | 0.316 | 0.691 | 0.553 |
| GENA-LM-large | 0.085 | 0.552 | 0.036 | 0.389 | 0.627 | 0.597 |
| GENERator-3B | 0.074 | 0.551 | 0.045 | 0.203 | 0.657 | 0.602 |
| GenerRNA | 0.049 | 0.533 | 0.03 | 0.182 | 0.642 | 0.571 |
| GenSLM | 0.09 | 0.544 | 0.042 | 0.389 | 0.631 | 0.563 |
| HyenaDNA | 0.08 | 0.543 | 0.043 | 0.279 | 0.656 | 0.585 |
| LucaOne | 0.076 | 0.546 | 0.05 | 0.292 | 0.695 | 0.595 |
| LucaVirus | 0.078 | 0.541 | 0.043 | 0.288 | 0.693 | 0.605 |
| N.T.-v2-500m | 0.039 | 0.526 | 0.027 | 0.195 | 0.626 | 0.564 |
| N.T.-v2-50m | 0.09 | 0.554 | 0.047 | 0.36 | 0.66 | 0.578 |
| RESM | 0.087 | 0.555 | 0.053 | 0.213 | 0.681 | 0.606 |
| RiNALMo_giga | 0.093 | 0.539 | 0.058 | 0.369 | 0.665 | 0.608 |
| structRFM | 0.097 | 0.552 | 0.058 | 0.266 | – | 0.592 |

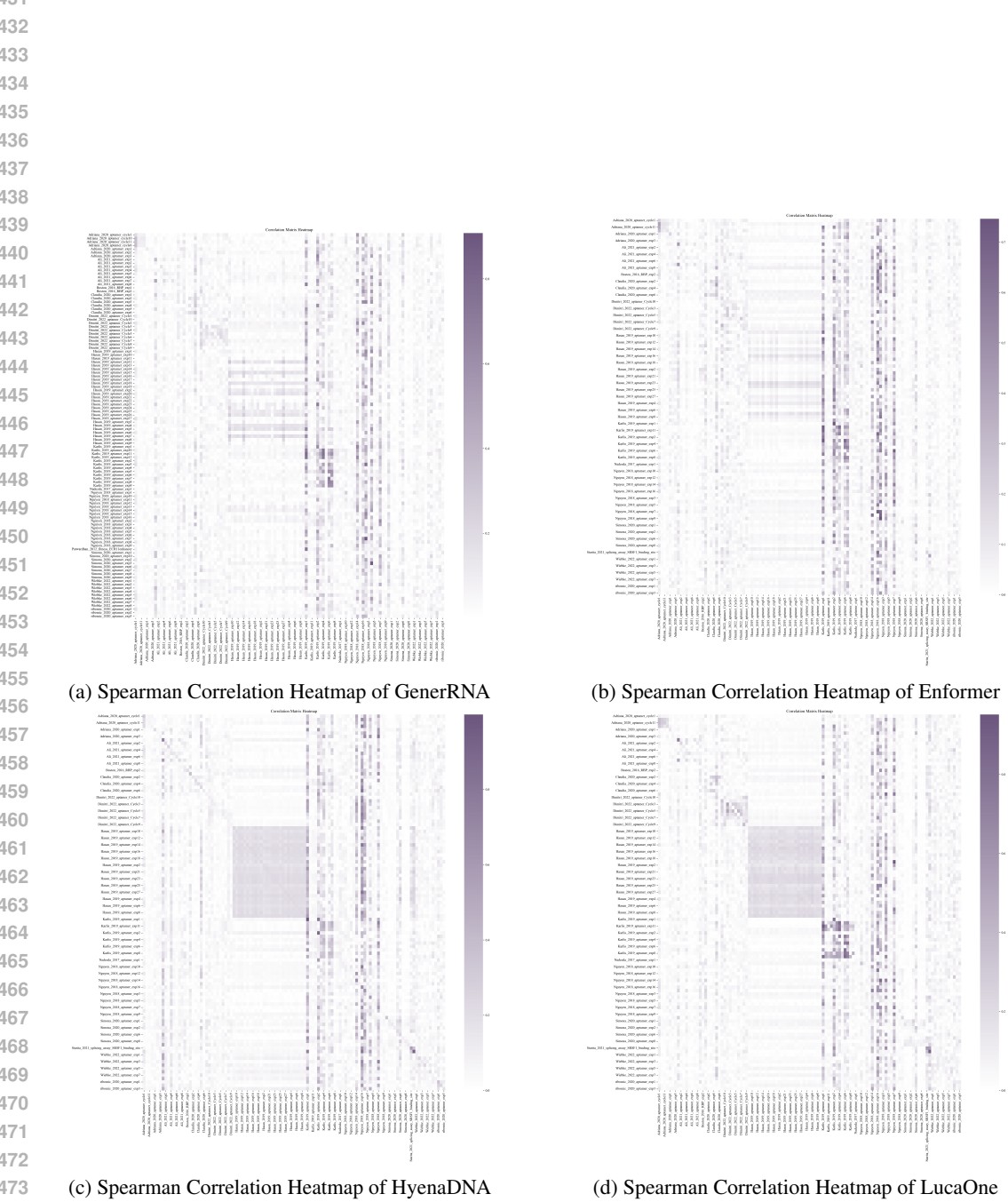

(a) Spearman Correlation Heatmap of GenerRNA

(b) Spearman Correlation Heatmap of Enformer

(c) Spearman Correlation Heatmap of HyenaDNA

(d) Spearman Correlation Heatmap of LucaOne

Figure 15: Transfer Learning Heatmap of SELEX assays

