# OpenReview forum: "NABench: Large-Scale Benchmarks of Nucleotide Foundation Models for Fitness Prediction"
_ICLR.cc/2026/Conference — Submitted to ICLR 2026_

### Official Review · Reviewer_qYsy · 2025-10-30

**Soundness:** 2
**Presentation:** 2
**Contribution:** 2
**Rating:** 4
**Confidence:** 4

**Summary:**

The authors proposed, NABench, a large-scale, standardized benchmark to fairly compare nucleotide foundation models (NFMs) on sequence-to-fitness prediction tasks across DNA and RNA. It aggregates 162 assays and ~2.6M mutated sequences spanning mRNA, tRNA, ribozymes, aptamers, enhancers, promoters, and exons, surpassing prior fitness benchmarks in scale and diversity. The authors reprocessed >110 experiments with a unified pipeline (quality control, read trimming/merging, deduping, clustering) to produce consistent datasets and splits with rich metadata. NABench evaluates 29 representative models (BERT-like, GPT-like, Hyena-like, and others) under zero-shot, few-shot, supervised, and transfer-learning regimes. They evaluate these models on four complementary metrics (Spearman correlation, NDCG, AUC, and MCC) that provide both ranking fidelity and top-variant discovery power. Across DMS tasks, no single model dominates in zero-shot, while BERT-style encoders often lead once supervision is introduced. Results highlight a notable gap between random and position-contiguous splits, exposing challenges in out-of-distribution generalization across mutational regions. For SELEX with assays with random sequence pools, zero-shot performance is weak across the board, but supervision helps, underscoring limitations of current NFMs on synthetic sequence spaces. NABench, released with code and data, establishes reproducible baselines and clear failure modes to guide future, possibly structure-aware, nucleotide modeling and design.

**Strengths:**

- NABench aggregates 162 assays and ~2.6M mutants across DNA and RNA, much larger than prior fitness benchmarks, giving statistically robust comparisons. The authors also release code and data with a clear evaluation recipe, supporting community adoption and extension.
- The benchmark spans seven nucleic-acid families (mRNA, tRNA, aptamers, ribozymes; DNA enhancers, promoters, exons), supporting conclusions that generalize across molecule types and assay designs.
- Over 110 raw experiments are reprocessed with a consistent and unified workflow (QC, trimming/merging, deduplication, clustering), reducing dataset bias and making results comparable.
- Models are tested under zero-shot, few-shot, supervised, and transfer-learning settings, capturing both intrinsic knowledge and supervised learning efficiency.
- The authors evaluate 29 representative NFMs in a common framework, enabling fair head-to-head comparisons.
- The metrics are complementary and capture correlation, high-score sequence prioritization, and classification power, aligning evaluation with real experimental goals.
- Both random and contiguous splits are used, explicitly testing interpolation vs. extrapolation and revealing generalization gaps that matter in practice.
- The paper analyzes architecture- and data-specific performance (e.g., Hyena/Evo strengths in zero-shot; BERT-style gains with supervision) and highlights efficiency–performance trade-offs for long sequences.

**Weaknesses:**

I will discuss them together with questions.

**Questions:**

I appreciate the huge amount of time the authors have spent on collecting, curating, processing and harmonizing this big dataset. This effort for sure has the potential to become one of the main benchmarks in evaluating nucleic acid foundation models. However, I have some questions and reservations, which I will list below.

- I get the feeling that the manuscript and data preparation is a bit rushed and does not have the quality and consistency needed for such a big task. I downloaded the provided supplementary data and randomly picked two datasets: Kolm et al. (2020), which a SELEX experiment for finding aptamers against bacterial growth and Baeza-Centurion et al. (2020), which is a DMS for studying impacts of mutations on alternative mRNA splicing. I did not find any data for Kolm and found this just one sequence `GGGAAATTTCCCCCCAAAGGGAAATTTCCCTTTAAAAAACCCTTTTTTGGGCCCTTTGGGTTTGGGGGGTTTTTTGGGTTTGGGTTTCCCTTTCCCCCCTTTGGGCCCTTTTTTCCCTTTCCCCCCCCCGGGAAATTTTTTCCCTTTAAAGGGTTTAAAAAATTTTTTGGGTTTTTTTTTGGGGGGGGG` for FAS exon 6 experiment. I am not sure what this is, does not map anywhere, and is different from the version in the original paper: `TGTCCAATGTTCCAACCTACAGGATCCAGATCTAACTTGCTGTGGTTGTGTCTCCTGCTTCTCCCGATTCTAGTAATTGTTTGGGGTAAGTTCTTGCTTTGTTCAAACTGCAGATTGAAATAACTTGGGAAGTAG`, which included a doped version of FAS exon 6 and a few bases from the flanking introns.
- Overall, performance metrics are pretty low and random-like. The drastic performance drop across the board from “Random CV” to "Contiguous CV” indicates that more sophisticated data splits are needed, and some of these models are simply cheating.
- Contiguous (position-blocked) splits, potentially useful for OOD testing, aren’t meaningful for SELEX and thus aren’t used there, limiting extrapolation tests on that large slice of the benchmark. Why not use sequence similarity metrics or clustering them and then splitting the folds by cluster/max similarity?
- Authors note some sequences may exist in pretraining corpora; while labels are unseen (so “zero-shot” stands), any overlap could advantage certain models and complicate fairness claims.
- Zero-shot performance on SELEX is uniformly poor and essentially random (Spearman correlation ≲ 0.1; mean AUC < 0.6), limiting conclusions about model priors on synthetic libraries without supervision. This is in part an issue with how SELEX experiments are binarized and evaluated. In a successful SELEX experiment (where the initial random pool has some functional molecules), there will be many good sequences in the final round, so assigning the top 10% as positives and the rest as negatives might not be the best strategy.
- AUC/MCC are computed by labeling only the top 10% sequences as positives; that fixed threshold is convenient but somewhat arbitrary and may not reflect assay-specific distributions. What if 50% of sequences are high-scoring and functional? I think a more sophisticated strategy should be used, e.g., fitting a GMM.
- Zero-shot “fitness” is simply the average of the embedding vector, which is architecture-agnostic but arguably crude. In DMS data for example, most sequences might be embedded into a very small space (many mutations are not consequential, hence should not change anything). Maybe the authors can run PCA on embeddings from the unlabelled samples and use the top principal component, or train a small VAE on them.
- The assay mix is heavily skewed (e.g., 94 SELEX aptamer assays and 2M out of 2.6M mutants vs far fewer per other families), which could bias aggregate rankings and hide family-specific strengths. On the same note, why even use aptamers? They are recognition/binding molecules, meaning that their “structure” is crucial, and this benchmark is ignoring them altogether.
- SELEX sequences are clustered (CD-HIT at 95%) partly to keep embedding runtime “reasonable,” which could merge near-neighbors and alter abundance/label granularity.
- The benchmark explicitly focuses on sequence-only NFMs and defers structure-aware evaluation to future work, so it can’t yet assess models that leverage RNA/DNA 2D/3D structure or inverse folding.
- Lastly, the paper's presentation needs some reworking. There are too many figures and too many tiny panels within them and they are not super helpful.

---

> ### Author Response · Authors · 2025-11-21
> **A kind rebuttal to Reviewer qYsy (Part 1)**
>
> Thank you for your feedback, and we will provide detailed responses to each of your problems.
>
> ## About Weakness 1
>
> Q: About the specific data problem.
> A: For Kolm, souce data is loacted at https://www.ncbi.nlm.nih.gov/bioproject/PRJNA615076/, and can be processed through our data curation methods. The processed dataset is named as "Claudia_2020_aptamer_exp[X]" in our dataset. We apologize for the misleading naming strategy and will change the name of the all dataset to the last name of the first author in later releases.
>
> For Baeza-Centurion assay (including `Baeza-Centurion_2020_exon-mutation_FAS_exon6_ancestor` and `Baeza-Centurion_2020_exon-mutation_PSMD14_exon11`). we download the source data and fit it into our pipeline, some compatibility issues occured and the wrong sequence is generated. We have correct this mistake and retest it on all models. The result show no improvement even after correction.
>
> Table 1:
> | Evo2-7b-base | FAS | PSMD14 |
> | --- | --- | --- |
> | Incorrect | 0.006 | 0.004 |
> | correct | 0.015 | 0.012 |
>
>
> Table 2:
> | Average_all_models | FAS | PSMD14 |
> | --- | --- | --- |
> | Incorrect | 0.005 | 0.004 |
> | correct | 0.004 | 0.002 |
>
>
> ## About Weakness 2
> Q: Overall, performance metrics are pretty low and random-like. The drastic performance drop across the board from “Random CV” to "Contiguous CV” indicates that more sophisticated data splits are needed, and some of these models are simply cheating.
>
> A: We appreciate the opportunity to clarify our Cross-Validation (CV) strategy. We intentionally adopted a two-stage evaluation protocol to highlight the significant impact of data splitting methods on model assessment. We initially employed Random CV as a standard baseline. While random splitting is currently the most prevalent strategy within the deep learning community, and it provides a relatively fair assessment by randomly partitioning sequences into training and validation sets. our results, as well as the reviewer’s comments, imply that models in this setting may exploit sequence homology, potentially leading to inflated performance metrics. To mitigate such leakage, we subsequently adopted the "Contiguous CV" method. Originally proposed in ProteinGym[1] and widely utilized in protein fitness prediction tasks, this approach partitions the wild-type sequence into five contiguous spatial blocks and reserves all mutations within specific regions for testing. This strategy strictly prevents the model from accessing local neighborhood information during training. Consequently, Contiguous CV forces the model to extrapolate learned functional rules to positionally distinct regions it has not previously encountered. This block-wise separation simulates realistic biological discovery scenarios, offering a more rigorous evaluation of whether the foundation model genuinely understands global biological semantics or is merely relying on rote memorization.
>
> ## About Weakness 3
>
> Q: Contiguous (position-blocked) splits, potentially useful for OOD testing, aren’t meaningful for SELEX and thus aren’t used there, limiting extrapolation tests on that large slice of the benchmark. Why not use sequence similarity metrics or clustering them and then splitting the folds by cluster/max similarity?
>
> A: Replacing contiguous splitting with clustering is indeed a natural intuition, and we agree that it effectively evaluates a model's deep reasoning capabilities. However, this approach does not align with the actual experimental workflow of SELEX. In real-world SELEX campaigns, the predictive task involves leveraging data from completed rounds to forecast enrichment in subsequent rounds. In this specific setup, the model must process the entire spectrum of sequences rather than being restricted to distinct clusters.
>
> ## About Weakness 4
>
> Q: Authors note some sequences may exist in pretraining corpora; while labels are unseen (so “zero-shot” stands), any overlap could advantage certain models and complicate fairness claims.
>
> A: We acknowledge that, given the massive scale of pre-training corpora used for foundation models, ensuring absolute zero overlap with test sequences is indeed an inherent challenge in this field.
>
> Despite this challenge, we firmly believe that the intrinsic nature of the variant effect prediction task structurally safeguards the fairness of our evaluation. While the model may have encountered the Wild-Type (WT) backbone in genomic databases during pre-training, the specific mutants generated in DMS and the aptamers in SELEX are predominantly synthetic and highly unlikely to exist in natural pre-training corpora. Crucially, since the specific mutations and their corresponding fitness labels are strictly unseen, the model cannot simply "recall" or "memorize" the answers. Instead, it is compelled to perform genuine inference based on learned evolutionary patterns. Consequently, even if the WT sequence was exposed during training, the validity of the Zero-shot evaluation remains intact.

---

> ### Author Response · Authors · 2025-11-21
> **A kind rebuttal to Reviewer qYsy (Part 2)**
>
> ## About Weakness 5
> Q: Zero-shot performance on SELEX is uniformly poor and essentially random (Spearman correlation ≲ 0.1; mean AUC < 0.6), limiting conclusions about model priors on synthetic libraries without supervision.
>
> A: We attribute the poor performance observed in zero-shot SELEX evaluation primarily to the specific formulation of the task. We posit that "fitness prediction" in DMS and SELEX constitutes two fundamentally distinct tasks requiring different computational capabilities:
>
> 1. DMS (Relative/Local Task)As validated in our benchmark (and consistent with findings from ProteinGym), DMS tasks quantify the impact of local perturbations on a known functional backbone. Given the high sequence homology between the wild-type and variants, the model focuses on the marginal shift in probability induced by the mutation. Pre-trained models excel here because, within this local context, "evolutionary likelihood" is highly correlated with "functional stability."
>
> 2. SELEX (Absolute/Global Task)Conversely, SELEX involves screening functional sequences from a high-diversity, randomized pool. Crucially, there is no shared reference backbone. In a zero-shot setting, the model predicts "naturalness" (likelihood under the pre-training distribution). However, the specific binding affinity to a SELEX target (often an arbitrary synthetic dye or a specific protein) may not correlate with "evolutionary naturalness." Consequently, the model is effectively comparing random sequences without a reference anchor, resulting in the observed near-random performance (AUC near 0.5).
>
> ## About Weakness 6
>
> Q: AUC/MCC are computed by labeling only the top 10% sequences as positives; that fixed threshold is convenient but somewhat arbitrary and may not reflect assay-specific distributions. What if 50% of sequences are high-scoring and functional? I think a more sophisticated strategy should be used, e.g., fitting a GMM.
>
> A: We apologize for any ambiguity regarding the "10%" threshold description in our manuscript. To clarify, the definition differs based on the data type:
> For DMS data, this refers strictly to the top 10% of sequences ranked by fitness score. For SELEX data, this refers to the sequences that account for the top 10% of the total read abundance (cumulative counts). We observed that SELEX data typically follows a heavy-tailed distribution containing extreme outliers, whereas the vast majority of sequences have read counts close to zero. Therefore, our strategy was to identify the dominant sequences that constitute the bulk of the pool's mass.
>
> Here is an example: Consider a simplified scenario: Sequence A1 has a read count of 10,000, while Sequences B1 through B100 each have a read count of 100. the total read is 20,000, and the top 10% of the total mass is 2,000. Since Sequence A1 alone (10,000) covers this volume, only A1 would be classified as a positive sample, despite B1–B100 being present.
>
> We agree that using a Gaussian Mixture Model (GMM) might provide a more robust method for determining this cutoff boundary. We plan to investigate the application of GMMs for thresholding in our future work.
>
> ## About Weakness 7
>
> Q: Maybe the authors can run PCA on embeddings from the unlabelled samples and use the top principal component, or train a small VAE on them.
>
> A: The intuition that "embeddings of single-point mutations might cluster closely" is indeed plausible. However, we argue that applying PCA or VAE-based methods in this context introduces two critical issues:
> 1. PCA is designed to identify directions of maximum variance within the data. In the context of biological sequences, the principal components of embeddings typically capture dominant structural features (such as sequence length) or simply the edit distance from the wild-type. However, fitness is a specific functional attribute that does not necessarily align with these directions of maximum variance. Relying on PCA risks capturing "trivial" statistical variations rather than the subtle "functional" signals required for fitness prediction.
> 2. Our benchmark is strictly designed to evaluate the intrinsic zero-shot capability of the foundation models themselves. Training a VAE or any auxiliary dimensionality reduction model on top of the embeddings would fundamentally shift the evaluation paradigm from Zero-shot Inference to Unsupervised Adaptation or Transductive Learning. This approach would obscure the true performance of the foundation model by conflating it with the capacity of the auxiliary learner. Furthermore, our current evaluation strategy is fully aligned with ProteinGym, the widely accepted standard for protein fitness prediction. Adhering to this established protocol ensures a fair, reproducible, and direct measurement of the model's representation space, facilitating valid comparisons with existing literature.

---

> ### Author Response · Authors · 2025-11-21
> **A kind rebuttal to Reviewer qYsy (Part 3)**
>
> ##  About Weakness 8
> Q: why even use aptamers? They are recognition/binding molecules, meaning that their “structure” is crucial, and this benchmark is ignoring them altogether. The benchmark explicitly focuses on sequence-only NFMs and defers structure-aware evaluation to future work, so it can’t yet assess models that leverage RNA/DNA 2D/3D structure or inverse folding.
>
> A: We appreciate the reviewer raising this critical point. We fully acknowledge that aptamer function is primarily driven by its three-dimensional structure. But given the core premise of Nucleic Acid Foundation Models (NFMs) is their ability to learn syntax and long-range dependencies directly from sequence data, thereby implicitly encoding structural constraints (such as hairpins and G-quadruplexes) without requiring explicit structural input. By incorporating aptamers, we aim to rigorously test a specific hypothesis: **Can current sequence-based FMs infer structure-driven binding affinity relying solely on sequence patterns?**
>
> Crucially, the suboptimal performance of models on this task is not a flaw of the benchmark, but a key insight. It highlights that current NFMs have not yet effectively bridged the gap between sequence and structure, thereby delineating clear directions for future model improvement. Moreover, unlike proteins, experimental structures for aptamers are extremely scarce. While tools like AlphaFold 3 represent a monumental leap forward, high-precision RNA structure prediction has not yet reached the level of maturity or computational scalability seen in protein folding[2]. Folding millions of SELEX sequences to create a structure-aware benchmark is currently computationally prohibitive and carries significant uncertainty. Given that the primary sequence ultimately dictates structure, utilizing sequence as the input for affinity prediction remains the standard practice in high-throughput SELEX analysis.
>
> ##  About Weakness 9
>
> Q: SELEX sequences are clustered (CD-HIT at 95%) partly to keep embedding runtime “reasonable,” which could merge near-neighbors and alter abundance/label granularity.
>
> A: We acknowledge that CD-HIT clustering may potentially influence abundance distributions or label granularity. However, our decision to employ this strategy was driven by both computational intractability and methodological robustness.
> 1. Conducting this study at a 95% sequence similarity threshold was already computationally intensive. The clustering process alone consumed approximately three weeks on a high-performance server (CPU: Intel Xeon Platinum 8558, 1TB RAM). The clustering reduced the dataset size to approximately 50% of the original volume. If we were to bypass clustering and process the raw data directly, the computational cost would increase exponentially, rendering it impossible to complete the analysis within the strict timeframe of the rebuttal period.
>
> 2. Beyond computational limitations, we argue that clustering at 95% similarity offers significant scientific value in terms of denoising. High-throughput SELEX data is inherently noisy due to sequencing errors and artifacts[3]. By clustering similar sequences, we effectively consolidate these variations, thereby improving the signal-to-noise ratio (SNR) of the representative sequences. This step helps ensure that the model learns from robust sequence motifs rather than overfitting to sequencing noise.
>
> ## About Weakness 10
> Q: The paper's presentation needs some reworking. There are too many figures and too many tiny panels within them and they are not super helpful.
>
> A: Thank you for your suggestion. We will refine the figure layout and presentation in future revisions. We also plan to remove certain non-essential panels to reduce visual clutter and improve the overall clarity and quality of the paper.
>
>
> ### Refernece
>
> [1] Notin, Pascal, et al. "Proteingym: Large-scale benchmarks for protein fitness prediction and design." Advances in Neural Information Processing Systems 36 (2023): 64331-64379.
> [2] Kretsch, Rachael C., et al. "Functional relevance of CASP16 nucleic acid predictions as evaluated by structure providers." Proteins: Structure, Function, and Bioinformatics (2025).
> [3] Gold, Larry. "SELEX: how it happened and where it will go." Journal of molecular evolution 81.5 (2015): 140-143.

---

### Official Review · Reviewer_rEPG · 2025-10-31

**Soundness:** 3
**Presentation:** 3
**Contribution:** 3
**Rating:** 6
**Confidence:** 3

**Summary:**

This paper introduces NABench, a large-scale, systematic benchmark for nucleotide fitness prediction , addressing the lack of standardized evaluation for Nucleotide Foundation Models. The benchmark aggregates 162 high-throughput assays, curating 2.6 million mutated sequences across diverse DNA and RNA families. NABench rigorously evaluates 29 representative NFMs, categorized by architecture, under a unified suite of four task settings: zero-shot, few-shot, supervised, and transfer learning. The results quantify performance heterogeneity, demonstrating that Hyena architectures excel in zero-shot DMS tasks , whereas BERT-like models show superior performance in supervised and few-shot settings.

**Strengths:**

- Scale and Diversity: NABench is the largest benchmark in this domain, comprising 2.6M mutants and 162 assays , making it 8x larger than the previous SOTA (RNAGym). It covers both DNA and RNA and includes two fundamentally different data types: DMS and SELEX.

- Comprehensive Evaluation: The benchmark tests 29 models and defines four rigorous evaluation paradigms, which is critical for a holistic understanding of model capabilities.

- Rigorous Supervised Splits: The supervised setting wisely employs two data splitting strategies: "Random Cross-Validation" and "Contiguous Cross-Validation", the latter being more relevant for realistic scientific discovery.

- Clear Insights: The results provide clear, actionable insights: Hyena/Evo models are optimal for zero-shot tasks , while BERT-like models are better for feature extraction. It also highlights the universal failure of current models on zero-shot SELEX prediction.

**Weaknesses:**

- Lack of Structure: The benchmark focuses exclusively on sequence-only models. It explicitly omits hybrid models that incorporate nucleic acid structure , which is a growing and important area of research.

- Limited Transfer Learning Analysis: Although transfer learning is listed as one of the four main evaluation settings, its analysis in the main paper is very brief. The transfer learning results for DMS tasks are missing entirely from the main results.

- Oversimplified Supervised Probe: The supervised and few-shot evaluations use only a ridge regression probe. While good for testing static embedding quality, this may under-estimate models that would perform better with a more complex probe or full-model fine-tuning.

- SELEX Task Failure: The near-random performance of all models on zero-shot SELEX is a key finding, but also suggests the task may be outside the scope of what current pretraining paradigms can achieve.

**Questions:**

- Choice of Supervised Probe: Why did you choose a linear ridge regression probe  instead of a non-linear MLP probe or full-model fine-tuning? Could this choice systematically favor models whose embeddings are more linearly separable (like BERT )?

- SELEX Generalization Gap: Does the failure on zero-shot SELEX imply that pretraining on "natural" sequences  fails to generalize to "synthetic" ones? Does this suggest "fitness prediction" is two separate tasks?

- DMS Transfer Learning Results: What were the key findings for the transfer learning task on DMS datasets? Why were these results omitted from the main paper?

- Embedding Strategy Sensitivity: You used different embedding strategies for different architectures. How sensitive are the model rankings to this choice?

---

> ### Author Response · Authors · 2025-11-21
> **A kind rebuttal to Reviewer rEPG (Part 1)**
>
> Thank you for your feedback, and we will provide detailed responses to each of your problems.
>
> ## About Weakness 1
> Q: The benchmark focuses exclusively on sequence-only models. It explicitly omits hybrid models that incorporate nucleic acid structure , which is a growing and important area of research.
>
> A: We appreciate this suggestion. However, given the current scarcity of RNA/DNA structures (e.g., there are only 7193 RNA entries in the RCSB PDB), and the fact that state-of-the-art structural foundation models (such as AlphaFold 3) have not yet reached the same level of maturity for RNA prediction as they have for proteins[1], it remains challenging to curate a large-scale dataset of sequence-structure pairs.
>
> This scarcity currently limits our ability to incorporate structure-based models into the NABench evaluation framework. Nevertheless, we anticipate that with the continued advancement of structural foundation models, improvements in RNA structure prediction accuracy, and the accumulation of experimentally resolved RNA data, future conditions will provide sufficient data to support rigorous research in this direction.
>
> ## About Weakness 2 & Question 3
> Q: The transfer learning results for DMS tasks are missing entirely from the main results.[DMS Transfer Learning Results]
>
> A: We sincerely apologize for the accidental omission of the DMS transfer learning results in the initial submission. We will rectify this in the revised manuscript, and the full DMS transfer learning results can be accessed immediately at [https://drive.google.com/file/d/1h9az1xDrRe5hMauOfnN-ldBQhnrBIen5](https://drive.google.com/file/d/1h9az1xDrRe5hMauOfnN-ldBQhnrBIen5).
>
> We conducted transfer learning experiments in the DMS dataset using both RESM and Lucaone model. Our results indicate that both models exhibit higher transferability when trained and tested on assays from the same study compared to those across different studies. Notably, RESM demonstrates this trend more distinctly, showing superior generalization and capacity for cross-dataset knowledge acquisition compared to Lucaone.
>
>
> However, to fully address the reviewer's valid point that the original analysis was brief, we went beyond merely adding the missing data. We have comprehensively expanded the scope of Section 3.6 to rigorously probe transferable principles across challenging Out-of-Distribution (OOD).
>
> Experimental Setup: We utilized Evo2-7b-base as the representative model. We selected this model as it represents the state-of-the-art, offering a balanced trade-off between scale and performance. While this response highlights findings from Evo2-7b-base, we are actively extending these protocols to all baseline models for the final revision.
>
> Expanded Analysis:  We present the data from our new OOD experiments below, covering two critical scenarios:Cross-Assay Transfer and Cross-Family Transfer learning. We present the data from our new OOD experiments below, covering two critical scenarios:
>
> - For cross-assay transfer: We conducted trials to test transferability between evolutionary fitness (DMS) and binding affinity (SELEX). As shown in Table 1, transferring from SELEX to DMS yields a significant negative correlation (-0.51). This indicates that features learned for binding affinity can be orthogonal or even contradictory to those required for organismal fitness. This proves that our benchmark captures deep biological gaps and is not merely testing surface-level patterns.
>
> - For cross-family transfer: We evaluated transferability across five distinct nucleic acid categories to map the semantic distance between them (see Table 2). This covers both the fundamental DNA $\leftrightarrow$ RNA gap and the diversity within RNA families. We observe strong positive transfer from mRNA to Ribozyme (0.58), suggesting shared structural/sequence motifs. Conversely, tRNA shows poor or negative transfer to most categories (e.g., -0.49 to DNA). This highlights the model's difficulty in bridging the gap between standard genomic sequences and the highly specialized, modified nature of tRNA.
>
> With the inclusion of the missing DMS results and this newly expanded OOD analysis, we believe NABench now provides a robust and deep investigation into the generalization boundaries of nucleic acid models.
>
> Table 1: Transfer-learning for Cross-Assay
> | Evo2-7b-base | Test on DMS | Test on SELEX |
> | --- | --- | --- |
> | Train on DMS | 0.798529 | -0.282021 |
> | Train on SELEX | -0.513178 | 0.450221 |
>
>
>
> Table 2: Transfer-learning for Cross-Family
> | evo2-7b-base | Aptamer | mRNA | tRNA | DNA | Ribozyme |
> | --- | --- | --- | --- | --- | --- |
> | Aptamer | 0.377866 | 0.309509 | -0.498929 | 0.458034 | 0.19478 |
> | mRNA | -0.227357 | 0.642536 | 0.132676 | -0.517262 | 0.586588 |
> | tRNA | -0.140534 | -0.338035 | 0.573033 | -0.491105 | 0.372182 |
> | DNA | -0.0409407 | -0.100272 | 0.0846473 | 0.630483 | 0.17052 |
> | Ribozyme | 0.154524 | 0.450222 | -0.04907 | -0.496196 | 0.841069 |

---

> ### Author Response · Authors · 2025-11-21
> **A kind rebuttal to Reviewer rEPG (Part 2)**
>
> ## About Weakness 3 & Question 1
>
> Q: Oversimplified Supervised Probe. [Choice of Supervised Probe]
>
> A: We appreciate this question and would like to address it from two perspectives: theoretical constraints and empirical verification.
>
> First, regarding the Supervised and Few-shot tasks, the scale of training samples varies significantly across different assays. Particularly in Few-shot scenarios, the available training data is extremely limited. Under such conditions, directly applying full-model fine-tuning carries significant risks. With extremely scarce training data, the model may fail to converge or, worse, suffer from catastrophic forgetting, thereby destroying the rich feature representations of the original pre-trained model. This instability is the primary reason we opted against full-model fine-tuning.
>
> Second, to ensure robust performance, we conducted a comparative analysis of multiple Supervised Probes (prediction heads) within our DMS supervised learning experiments. The results of this evaluation are presented in the table below:
>
> Table 1: Different Probe Performance for Supervised Learning
>
> | Model | Mean Absolute Correlation |
> | --- | --- |
> | RidgeCV | 0.6171 |
> | LinearSVR | 0.5405 |
> | RandomForest | 0.5296 |
> | MLP | 0.5190 |
> | XGBoost | 0.5059 |
> | LinearRegression | 0.4969 |
> | LassoCV | 0.4812 |
>
> As observed, the Ridge Regression probe achieved the best performance among the seven common machine learning models evaluated. Given its superior stability and inherent robustness against noise and outliers, we identified it as the most appropriate choice for the supervised probe in our experimental setup.
>
>
> ## About Weakness 4 & Question 2
> Q: SELEX Task Failure: The near-random performance of all models on zero-shot SELEX is a key finding, but also suggests the task may be outside the scope of what current pretraining paradigms can achieve. [SELEX Generalization Gap]
>
> A: We appreciate this insightful question. We emphasize that the observed performance gap is a key conclusion of our analysis rather than a flaw in the experimental design. This result empirically confirms that current pretraining paradigms cannot trivially transfer to the zero-shot prediction of synthetic SELEX data. Identifying this "failure mode" is a critical contribution of our paper, as it proves that aligning pretraining objectives with in vitro selection remains an open challenge for the community.
>
> We agree with your hypothesis: "fitness prediction" in DMS and SELEX effectively constitutes two distinct tasks requiring different capabilities. We analyze this divergence through the following two dimensions:
>
> 1. The Gap Between "Local" and "Global" Landscapes: As validated in our benchmark (and consistent with findings in ProteinGym[2] for proteins), DMS tasks measure the impact of local perturbations on a known functional backbone. Since the wild-type (WT) and variants share high sequence homology, the model can focus on the marginal probability shift caused by the mutation. Pre-trained models excel here because, within a local context, "evolutionary likelihood" is highly correlated with functional stability. Conversely, SELEX involves screening functional sequences from a high-diversity, often synthetic, random pool. Unlike DMS, there is no shared reference backbone. In a Zero-shot setting, the model essentially predicts "naturalness" (likelihood under the pre-training distribution). However, specific binding affinity to a SELEX target (which can be an arbitrary synthetic dye or a specific protein) often does not correlate with evolutionary naturalness. Without a reference anchor (WT), the model is effectively comparing random sequences, leading to the observed near-random performance (AUC $\approx$ 0.5).
>
> 2. It is crucial to clarify that the failure in Zero-shot SELEX does not imply that the models fail on synthetic sequences; rather, it indicates an inability to align "evolutionary likelihood" with arbitrary biophysical affinity without supervision. When we applied supervised learning to the same SELEX data (as shown in our Supervised Learning tasks), performance improved significantly. This proves that the pre-trained embeddings do contain distinct features useful for synthetic sequences, but the Zero-shot proxy (likelihood) is misaligned with the specific SELEX target.
>
> In conclusion, NABench reveals a critical insight: current nucleic acid foundation models are robust "Variant Effect Predictors" (DMS) but are not yet reliable "Zero-shot Function Discoverers" (SELEX) for arbitrary targets. This distinction is vital for helping the community understand what these models can and cannot do.

---

> ### Author Response · Authors · 2025-11-21
> **A kind rebuttal to Reviewer rEPG (Part 3)**
>
> ## About Question 4
>
> Q: Embedding Strategy Sensitivity: You used different embedding strategies for different architectures. How sensitive are the model rankings to this choice?
>
> A: We acknowledge the use of architecture-specific embedding strategies, but we emphasize that by adopting these standard practices, we aim to maximize the informative signal specific to each architecture. Regarding the sensitivity of rankings, we believe that the observed rankings represent a robust assessment of each model's maximal potential under standard protocols. While rankings would naturally shift if we deviated from these norms, such shifts would stem from artificially degrading specific models rather than genuine capability differences. By adopting these community standards, we ensure that the rankings reflect the fundamental representation capabilities of the models at their best, minimizing artifacts caused by sub-optimal feature engineering.
>
>
> ### Reference
>
>
> [1] Kretsch, Rachael C., et al. "Functional relevance of CASP16 nucleic acid predictions as evaluated by structure providers." Proteins: Structure, Function, and Bioinformatics (2025).
>
> [2] Notin, Pascal, et al. "Proteingym: Large-scale benchmarks for protein fitness prediction and design." Advances in Neural Information Processing Systems 36 (2023): 64331-64379.

---

### Official Review · Reviewer_TTvY · 2025-10-31

**Soundness:** 3
**Presentation:** 3
**Contribution:** 3
**Rating:** 8
**Confidence:** 4

**Summary:**

The paper presents NABench, a large-scale benchmark for evaluating nucleotide foundation models in DNA and RNA fitness prediction. NABench compiles 162 assays with over 2.6 million mutated sequences from Deep Mutational Scanning and SELEX experiments. It standardizes data processing, train–test splits, and evaluation metrics to ensure consistent and fair comparison.

The benchmark assesses 29 models across four evaluation settings: zero-shot, few-shot, supervised, and transfer learning, using metrics such as Spearman correlation and AUC. Results show that no single model performs best in all regimes. State-space models perform better in zero-shot prediction, while BERT-style models perform better in supervised and few-shot conditions. The benchmark exposes clear generalization gaps between natural and synthetic data and provides a public platform for reproducible evaluation.

**Strengths:**

1. Substantially larger and more diverse dataset
NABench contains 2.6 million sequences from 162 assays, making it around eight times larger than previous nucleotide benchmarks such as RNAGym. It covers both DNA and RNA tasks and spans diverse experimental types, providing a richer and more representative evaluation corpus.

2. Broader model coverage
The benchmark includes 29 models across seven architecture families, such as BERT, GPT, Hyena, and Evo variants. This extensive coverage allows systematic comparison between transformer-based and state-space designs, offering the most comprehensive model-level evaluation in the domain to date.

**Weaknesses:**

1. Clarification on dataset source.
The paper states that NABench integrates experimental measurements from Deep Mutational Scanning (DMS). However, according to Section 3.2, the dataset is derived from MaveDB, which broadly contains Multiplexed Assays of Variant Effect (MAVE), including both DMS and SELEX experiments. The authors should clarify whether they extracted only the DMS subset or used the full MaveDB collection. If the latter, they should revise phrasing such as “NABench integrates an extensive collection of experimental measurements from deep mutational scanning (DMS)” to accurately reflect the broader MAVE scope


2. Embedding extraction for Evo models.
In Section 3.4.1, the authors note that Evo models “output sequence-level embeddings” and are used directly. However, Evo-2 (referenced in their baseline table) is trained using an autoregressive approach based on the StripedHyena 2 architecture. The Evo-2 paper emphasizes that middle-layer embeddings better capture fitness-related features, while final-layer embeddings mainly encode autoregressive objectives. The authors should verify whether the Evo embeddings used in NABench come from the last layer or an intermediate layer, and consider aligning their extraction strategy with the Evo-2 paper’s recommendation (i.e., use middle layers for fitness prediction, or use last layers consistently across AR-style models)

**Questions:**

1. **DNA–RNA conversion**
   Please clarify how DNA and RNA sequences were handled for model input. Were DNA sequences converted (T→U) for RNA models or vice versa?

2. **Contiguous cross-validation**
   More details are needed on how contiguous CV was defined.

3. **Species breakdown**
   Do the datasets include viral, prokaryotic, and eukaryotic sequences? Any performance breakdown across these groups would be helpful.

---

> ### Comment · Reviewer_TTvY · 2025-11-18
>
> After reading comments from other reviewers, I believe the concerns about the SELEX subset are valid, and I also share the question regarding the use of Ridge as the downstream model for supervised learning. Therefore, I have decided to change my rating from 8 to 6 for now.

---

> ### Author Response · Authors · 2025-11-21
> **A kind rebuttal to Reviewer TTvY (Part 1)**
>
> We sincerely thank you for your thorough and insightful feedback, and we will provide detailed responses to each of your concerns.
>
>
> ## About Weaknesses 1
>
> Q: Clarification on dataset source. The paper states that NABench integrates experimental measurements from Deep Mutational Scanning (DMS). However, according to Section 3.2, the dataset is derived from MaveDB, which broadly contains Multiplexed Assays of Variant Effect (MAVE), including both DMS and SELEX experiments. The authors should clarify whether they extracted only the DMS subset or used the full MaveDB collection. If the latter, they should revise phrasing such as “NABench integrates an extensive collection of experimental measurements from deep mutational scanning (DMS)” to accurately reflect the broader MAVE scope.
>
> A: Since most datasets are protein DMS datasets, we only extract a small portion of the dataset in MaveDB that meets the requirement of being a valid DNA/RNA DMS dataset, listed below. We will provide a clear explanation in the revised version to clarify any potential ambiguities.
>
>
> Table 1: Selected datasets from MaveDB
>
> | No. | MaveDB i.d. |
> | --- | --- |
> | 1 | urn:mavedb:00000006-a-1 |
> | 2 | urn:mavedb:00000007-a-1 |
> | 3 | urn:mavedb:00000015-a-1 |
> | 4 | urn:mavedb:00000018-a-1 |
> | 5 | urn:mavedb:00000019-a-1 |
> | 6 | urn:mavedb:00000020-a-1 |
> | 7 | urn:mavedb:00000083 |
>
>
> ## About Weaknesses 2
>
> Q:  The authors should verify whether the Evo embeddings used in NABench come from the last layer or an intermediate layer, and consider aligning their extraction strategy with the Evo-2 paper’s recommendation (i.e., use middle layers for fitness prediction, or use last layers consistently across AR-style models)
>
> A: We thank the reviewer for this insightful observation regarding the StripedHyena architecture. To address the concern that using the final layer might be suboptimal compared to intermediate layers, we conducted a comprehensive layer-wise ablation study on the full DMS dataset using `Evo-2-base` model.  The results are available at the following link: [https://drive.google.com/file/d/1m251bhZBtMQmmdCeUyFq_YgaUAy6sPX2](https://drive.google.com/file/d/1m251bhZBtMQmmdCeUyFq_YgaUAy6sPX2). As illustrated in this figure, while intermediate layers effectively capture fitness-related features, the performance gap between the best-performing middle layer and the final layer is statistically marginal (e.g., a difference of less than 0.01 in Spearman correlation). This empirical evidence confirms that for the zero-shot inference tasks in this benchmark, the final layer already encodes sufficient semantic information, and the middle-layer advantage is not a decisive factor in this specific context.
>
>
> ## About Question 1
>
> Q: Please clarify how DNA and RNA sequences were handled for model input. Were DNA sequences converted (T→U) for RNA models or vice versa?
>
> A: To ensure compatibility, we inspected the vocabulary of each model’s tokenizer prior to inference. If the tokenizer included tokens for both Thymine (T) and Uracil (U), the sequences were input in their original format. If the tokenizer supported only one format (e.g., strictly DNA), we performed the necessary character conversion (e.g., U to T) to match the model's pre-training data.
>
> ## About Question 2
>
> Q: More details are needed on how contiguous CV was defined.
>
> A: As defined in ProteinGym[1], the sequence is split contiguously along its length, to obtain 5 segments of contiguous positions, and assign mutations to each segment based on the position at which it occurs. We will provide a more detailed explanation and discussion of this concept in the revised version.

---

> ### Author Response · Authors · 2025-11-21
> **A kind rebuttal to Reviewer TTvY (Part 2)**
>
> ## About Question 3
>
> Q: Do the datasets include viral, prokaryotic, and eukaryotic sequences? Any performance breakdown across these groups would be helpful.
>
> A: We appreciate the reviewer’s focus on species diversity. We have conducted a comprehensive taxonomic classification of the species origins for each dataset within NABench. As shown in the table 2, our datasets encompass a wide range of sources, including Prokaryotic, Eukaryotic, and Synthetic sequences. This broad taxonomic distribution demonstrates that our benchmark is not confined to a single domain; rather, it possesses high biological coverage, ensuring the generalizability of evaluation results across diverse biological contexts.
>
> To address potential performance variations across species categories, we further conducted a Stratified Performance Analysis using the evo-2-base model. As shown in Table 3, the results reveal that models tend to perform better on natural datasets. This granular analysis not only addresses your query but also profoundly reveals the generalization boundaries of current foundation models across taxonomic groups, providing a vital reference for the nucleic acid research community regarding model applicability in specific biological domains. We will include the statistical results of all models across different species in the appendix in future revisions, enabling a more comprehensive and robust analysis.
>
>
> Table 2: Taxonomic Distribution of NABench Datasets
> | Article Information | Experimental Environment |
> | --- | --- |
> | Lubliner et al. (2015) | Eukaryote |
> | Rotrattanadumrong & Yokobayashi (2022) | Synthetic / In vitro |
> | Sharon et al. (2012) | Eukaryote |
> | Findlay et al. (2018) | Eukaryote |
> | Andreasson et al. (2020) | Synthetic / In vitro |
> | Beck et al. (2022) | Synthetic / In vitro |
> | Domingo et al. (2018) | Eukaryote |
> | Guy et al. (2014) | Eukaryote |
> | Janzen et al. (2022) | Synthetic / In vitro |
> | Julien et al. (2016) | Eukaryote |
> | Kobori et al. (2015) | Synthetic / In vitro |
> | Kobori et al. (2017) | Synthetic / In vitro |
> | Li et al. (2016) | Eukaryote |
> | Peri et al. (2022) | Prokaryote |
> | Pitt & Ferré-D’Amaré (2010) | Synthetic / In vitro |
> | Roberts et al. (2023) | Synthetic / In vitro |
> | Soo et al. (2021) | Eukaryote |
> | Tome et al. (2014) | Synthetic / In vitro |
> | Townshend et al. (2015) | Eukaryote |
> | Corces et al. (2018) | Eukaryote |
> | Fuente et al. (2020) | Eukaryote |
> | Bashir et al. (2021) | Synthetic / In vitro |
> | Baeza-Centurion et al. (2020) | Eukaryote |
> | Boston College (2016) | Synthetic / In vitro |
> | Kolm et al. (2020) | Prokaryotic |
> | Van Simaeys et al. (2022) | Eukaryote |
> | Zumrut et al. (2019) | Synthetic / In Vitro |
> | Pleiko et al. (2019) | Synthetic / In Vitro |
> | Nguyen Quang et al. (2018) | Synthetic / In Vitro |
> | Ribomic Inc. (2019) | Synthetic / In vitro |
> | Camorani et al. (2020) | Synthetic / In vitro |
> | Sabrowski et al. (2022) | Synthetic / In Vitro |
> | Yu et al. (2024) | Prokaryotic |
>
> Table 3: Zero-Shot Performance of evo-2-base Model Across Species
>
> | Species | Correlation |
> | --- | --- |
> | Eukaryote | 0.222617 |
> | Prokaryote | 0.194174 |
> | Synthetic | 0.10392 |
>
>
>
> Reference:
>
> [1] Notin, Pascal, et al. "Proteingym: Large-scale benchmarks for protein fitness prediction and design." Advances in Neural Information Processing Systems 36 (2023): 64331-64379.

---

> ### Author Response · Authors · 2025-11-21
> **A kind rebuttal to Reviewer TTvY (Part 3)**
>
> ## About Concern
>
> Q: After reading comments from other reviewers, I believe the concerns about the SELEX subset are valid, and I also share the question regarding the use of Ridge as the downstream model for supervised learning.
>
> A: We appreciate you taking the time to review the other comments and for your transparency regarding the score adjustment. We regard the SELEX subset as a critical contribution to nucleic fitness benchmarking. It represents a high-effort curation that pioneers an unexplored landscape for DNA/RNA sequence models. Unlike DMS tasks, which focus on local mutational changes, SELEX provides a view of general functional landscapes. Without prior knowledge on these synthetic libraries, it is expected to see limited signals when performing zero-shot predictions on these tasks. But, the rapid rise from zero-shot to few-shot and cross-validation is a key finding showing that some of the latest model can acquire an understanding of these data when given minimal context. The SELEX task provide a completely different view than DMS tasks, going from local changes to general function, and therefore serves as a vital test of a model's adaptability and learning efficiency.
>
> When comes to the use of Ridge as the downstream model for supervised learning, it is tested that Ridge model perform the best comparing to other more complex machine learning strategies (see Table 4). The strong performance of a linear probe suggests that the embeddings generated by foundation models are already high-quality and linearly separable. Another major reason for using Ridge is that some assays in our benchmark have small sample sizes. In these regimes, complex models are prone to overfitting.
>
> Table 4: Different Probe Performance for Supervised Learning
> | Model | Mean Absolute Correlation |
> | --- | --- |
> | RidgeCV | 0.6171 |
> | LinearSVR | 0.5405 |
> | RandomForest | 0.5296 |
> | MLP | 0.5190 |
> | XGBoost | 0.5059 |
> | LinearRegression | 0.4969 |
> | LassoCV | 0.4812 |

---

### Official Review · Reviewer_EUvA · 2025-11-03

**Soundness:** 2
**Presentation:** 3
**Contribution:** 2
**Rating:** 2
**Confidence:** 5

**Summary:**

In this paper, the authors have undertaken a significant effort in curating a large-scale benchmark (162 assays, 2.6M sequences, 29 models) to systematize the evaluation of nucleotide foundation models.

**Strengths:**

The scale of the data curated in this benchmark is the core strength.

**Weaknesses:**

However, I think this is a recycle from the previous NeurIPS Dataset and Benchmark track to ICLR, which immediately raised my eyebrow. After a careful read of this paper, I found the evaluation protocols have strong, unjustified biases. The analysis, while broad, remains superficial in key areas. This leads to my concerns about the conclusions, which may be artifacts of the benchmark's design rather than fundamental properties of the models being tested.

**Questions:**

1. Your zero-shot evaluation protocol (Section 3.4.3) predicts fitness by "computing the mean of the variant's embedding vector". This is an extremely simplistic linear probe that implicitly assumes the models' pre-trained embedding spaces are already perfectly aligned with the downstream task of fitness prediction. Why was this naive approach chosen over more established and powerful zero-shot inference methods for language models, such as using masked token probabilities or pseudo-log-likelihoods? Does this not create a significant evaluation bias that conflates the quality of a model's representations with the arbitrary suitability of a mean-pooling probe?

2. The transfer learning analysis (Section 3.6) is presented as a key evaluation setting but is arguably the most superficial. The analysis is limited to a single correlation matrix (Figure 11 in Appendix), with the primary conclusion: "assays under the same experiment have higher correlation". This is a nearly tautological finding. Why does the benchmark not investigate more challenging and scientifically interesting transfer learning scenarios, such as transferring from DMS to SELEX, from DNA to RNA tasks, or from common, data-rich families to rare, data-poor ones, which would truly test the "transferable principles of nucleic acid biology" you claim to probe?

3. In Section 3.4.1, you mentioned that you use different, hand-picked strategies for extracting sequence-level embeddings from different model architectures (e.g., concatenating <cls> and mean-pooling for BERT, using the last hidden state for GPT). These are heuristics, not principled methods! How can you ensure that the performance differences reported are not simply artifacts of these arbitrary and potentially suboptimal embedding choices? Did you perform any ablation studies to demonstrate that your conclusions are robust to different embedding extraction strategies for each model?

4. The benchmark evaluates an impressive 29 foundation models but appears to lack any non-foundation model baselines. A robust benchmark must ground its findings by comparing against strong, simpler alternatives. For example, how do these multi-billion parameter models compare against a well-tuned CNNs trained from scratch on a one-hot encoding of the sequence, or even simpler k-mer based regression models (e.g., a kernel ridge regression)?

5. The paper correctly identifies that DMS (local mutations on a known scaffold) and SELEX (discovery of functional sequences from a random pool) are fundamentally different tasks. Indeed, your results show that model performance is 'significantly' different between them, with zero-shot performance on SELEX being near-random (Lines 424-428). Given this profound difference, why are results from both tasks aggregated into single "overall performance" metrics and rankings (e.g., Figure 2a)? Doesn't this aggregation obscure more than it reveals, by averaging performance over tasks that test completely different scientific capabilities (interpolation vs. extrapolation; local vs. global search)?

---

> ### Author Response · Authors · 2025-11-21
> **A kind rebuttal to Reviewer EUvA (Part 1)**
>
> I sincerely appreciate your thoughtful revisions and constructive feedback on this paper. I will now address your questions in detail. We will subsequently incorporate these updated experiments and discussions into the revised version of the manuscript.
>
>
> First of all, we must firmly clarify a serious factual misunderstanding regarding the origin and integrity of this manuscript.
>
> (1) Contrary to the Reviewer's speculation, this manuscript is an **original submission** and **has never been submitted to or rejected from the NeurIPS Dataset and Benchmark track**.  This speculation is factually incorrect. Since the list of rejected papers for NeurIPS 2025 is publicly available (https://openreview.net/group?id=NeurIPS.cc/2025/Conference#tab-reject), this fact can be rigorously verified. We respectfully request that our work be evaluated based on its specific scientific contributions to the nucleic acid foundation models, rather than unfounded assumptions about its submission history.
>
> (2) We strongly disagree with the claim that our evaluation protocols contain “unjustified biases” or are merely “artifacts” of a recycled benchmark. This critique rests on the incorrect premise that our work constitutes a form of “recycling”. Since that premise does not hold, the subsequent characterization of our methodological choices as “artifacts” is likewise unfounded. Our evaluation protocols are deliberately constructed and scientifically motivated, driven by two essential considerations:
>     - Fairness Across Heterogeneous Architectures: To fairly compare BERT, GPT, and Hyena architectures,  a model-agnostic representation strategy is essential to eliminate metric-induced bias.
>     - Consistency Across Learning Paradigms: To enable rigorous side-by-side comparisons between Zero-shot, Few-shot, Supervised and transfer-learning tasks, consistent input representations are required to isolate the impact of data scale from feature modality.
>
> Therefore, these protocols are not "biases",  and they are designed to ensure objective comparisons.
>
>
> ## About Question1
> Q:  Zero-shot evaluation protocol problem.
>
> A: We acknowledge that using Masked Token Probabilities or Pseudo-Log-Likelihoods (PLLs) is indeed the more established evaluation protocol for zero-shot prediction in protein science. Addressing your concern that "Mean Pooling might introduce evaluation bias," we would like to justify our choice of this evaluation protocol based on three key dimensions:
>
> (1) Consistency Across Evaluation Paradigms. Our study establishes a comprehensive framework covering four distinct learning paradigms: Zero-shot, Few-shot, Supervised, and Transfer Learning. In Few-shot, Supervised, and Transfer Learning tasks, using the "mean embedding of variants" as the input for downstream predictors is standard practice. To achieve a rigorous horizontal comparison across these paradigms (e.g., to quantify the exact performance gain provided by limited supervision compared to zero-shot inference), we must ensure the consistency of the input representations. If we were to use Logits/Probabilities for Zero-shot tasks while using Embeddings for others, it would be difficult to disentangle whether performance differences stem from "variations in data scale" or "differences in feature modality." Therefore, standardizing on mean embeddings eliminates this confounding variable, allowing us to more precisely evaluate the relative advantages of different learning paradigms.
>
> (2) Fairness Across Architectures. This study encompasses three distinct model architectures: BERT, GPT, and Hyena. Traditionally, BERT favors masked probabilities, GPT relies on PLLs, and architectures like Hyena often focus on embedding features. Adopting a unified, Model-Agnostic representation strategy (i.e., Mean Pooling) minimizes evaluation bias caused by the "coupling of specific architectures with specific metrics." This ensures that we are fairly comparing the "semantic quality of the representation space" of the models, rather than the differences in their specific decoding heads.
>
> (3)Empirical Validity. To verify that this strategy does not introduce significant bias, we conducted a comparative experiment between "Mean Pooling" and standard "Masked Token Probabilities" using the BERT architecture on the full DMS dataset.  The results are available at the following link: [https://drive.google.com/file/d/1VYr95e0sZQxAbLlhgR0rp9tT1JI_qVaE](https://drive.google.com/file/d/1VYr95e0sZQxAbLlhgR0rp9tT1JI_qVaE). As shown in this figure, the results demonstrate a statistically significant positive correlation between the two methods (**Spearman = 0.737**). This provides strong empirical evidence that, within the context of this evaluation, using mean embeddings serves as an accurate and reliable proxy.
>
> In summary, the selection of this evaluation protocol was a deliberate scientific choice to ensure the fairness of multi-architecture comparisons.

---

> ### Author Response · Authors · 2025-11-21
> **A kind rebuttal to Reviewer EUvA (Part 2)**
>
> ## About Question2
>
> Q: The transfer learning analysis (Section 3.6) is presented as a key evaluation setting but is arguably the most superficial.
>
> A: We appreciate the reviewer’s suggestion regarding the depth of the transfer learning analysis. To address this concern, we have radically expanded the scope of Section 3.6 to probe transferable principles across different assays and biological families. To execute this intensive task within the limited rebuttal timeframe, we utilized `Evo2-7b-base` as the primary foundation model, which represents a state-of-the-art model with a highly balanced trade-off between model scale and performance. While this response highlights findings from `Evo2-7b-base`, we are actively extending these protocols to all baseline models and will include the full benchmark in the final revision.
>
> We present the full data from our expanded Out-of-Distribution experiments below, covering both Cross-Assay (DMS $\leftrightarrow$ SELEX, see Table 1) and Cross-Family (DNA $\leftrightarrow$ RNA varieties and different RNA families, see Table 2) scenarios.
>
> - for Cross-Assay Transfer: We conducted experimental trials to rigorously test the transferability between evolutionary fitness landscapes (DMS) and binding affinity landscapes (SELEX). As shown in Table 1, transferring from SELEX to DMS yields a significant negative correlation (-0.51). This indicates that the features learned for binding affinity can be orthogonal or even contradictory to those required for organismal fitness. This proves that our benchmark captures challenging biological gaps.
>
> - for Cross-Family Transfer: We further evaluated transferability across five distinct nucleic acid categories, covering both the fundamental DNA $\leftrightarrow$ RNA gap and the diversity within RNA families (ranging from data-rich mRNA to specialized, structure-dependent families like tRNA and Ribozyme), allowing us to explore the relationship between them. We observe strong positive transfer from mRNA to Ribozyme (0.58), suggesting shared structural or sequence motifs. Conversely, tRNA shows poor or negative transfer to most other categories (e.g., -0.49 to DNA). This highlights the model's difficulty in bridging the gap between standard genomic sequences and the highly specialized, modified nature of tRNA.
>
> In the end, we are confident that these expanded OOD protocols significantly enhance the benchmark's coverage, providing the community with critical insights into the limits of current foundation models in handling complex biological shifts.
>
>
> Table 1: Transfer-learning for Cross-Assay
> | Evo2-7b-base | Test on DMS | Test on SELEX |
> | --- | --- | --- |
> | Train on DMS | 0.798529 | -0.282021 |
> | Train on SELEX | -0.513178 | 0.450221 |
>
>
>
> Table 2: Transfer-learning for Cross-Family
> | evo2-7b-base | Aptamer | mRNA | tRNA | DNA | Ribozyme |
> | --- | --- | --- | --- | --- | --- |
> | Aptamer | 0.377866 | 0.309509 | -0.498929 | 0.458034 | 0.19478 |
> | mRNA | -0.227357 | 0.642536 | 0.132676 | -0.517262 | 0.586588 |
> | tRNA | -0.140534 | -0.338035 | 0.573033 | -0.491105 | 0.372182 |
> | DNA | -0.0409407 | -0.100272 | 0.0846473 | 0.630483 | 0.17052 |
> | Ribozyme | 0.154524 | 0.450222 | -0.04907 | -0.496196 | 0.841069 |
>
>
> ## About Question 3
>
> Q:  Extracting sequence-level embeddings from different model architectures.
>
> A: We appreciate the reviewer’s suggestion regarding the embedding extraction strategies. We respectfully clarify that our protocols are not arbitrary heuristics but strictly adhere to established community consensus found in leading benchmarks like ProteinGym[1] and RNAGym[2]. By adopting these standard architecture-specific practices (e.g., Mean Pooling for BERT, Last Hidden State for GPT), we ensure our results are directly comparable with existing literature. Deviating from these norms to hand-optimize strategies for each model would make the benchmark less consistent with the broader field and introduce non-standard variables.
>
> Furthermore, we deliberately avoid performing extensive ablation studies or per-model optimization to prevent **hyperparameter selection bias**. Fine-tuning the extraction strategy for each individual model would shift the evaluation focus from the inherent quality of the pre-trained representation to the effectiveness of the tuning process. By applying a unified, architecture-specific rule, we ensure a fair and off-the-shelf comparison that minimizes human intervention. This approach not only guarantees that the reported differences reflect the fundamental representational capabilities of the models rather than artifacts of overfitting, but also offers the community simple, robust, and plug-and-play rules that are easily transferable to real-world downstream applications without complex feature engineering.

---

> ### Author Response · Authors · 2025-11-21
> **A kind rebuttal to Reviewer EUvA (Part 3)**
>
> ## About Question 4
>
> Q: The benchmark evaluates an impressive 29 foundation models but appears to lack any non-foundation model baselines. A robust benchmark must ground its findings by comparing against strong, simpler alternatives. For example, how do these multi-billion parameter models compare against a well-tuned CNNs trained from scratch on a one-hot encoding of the sequence, or even simpler k-mer based regression models (e.g., a kernel ridge regression)?
>
> A: We thank the Reviewer for highlighting the importance of non-foundation model baselines. A primary motivation of this benchmark is to evaluate the **Zero-shot inference capability** of foundation models. Traditional baselines (e.g., CNNs or k-mer regression) strictly require supervised labels to learn a mapping function and cannot perform inference in a pure zero-shot setting (i.e., predicting fitness without seeing any fitness data). This distinction highlights a fundamental advantage of pre-trained FMs: their ability to generalize to downstream tasks using only evolutionary patterns learned from unlabeled sequences.
>
> We agree that establishing performance relative to simpler methods is crucial for a robust benchmark. Hence, to directly address the reviewer’s concern regarding supervised learning, we have added a Train-from-Scratch baseline. We implemented a One-hot encoded 2-layer MLP (serving as a representative lightweight neural network) and compared it against the foundation models on the supervised learning tasks. As shown in Table 3, the pre-trained foundation models significantly outperform the this simple baseline. Given the result, it can be observed that while the simple baseline show some signals in simpler random CV tasks, there is almost no signals at all in contiguous CV tasks. This further proves that large foundation models can grasps knowledge on unseen data, and is more suitable for difficult tasks.
>
> Table 3: Experiment results for MLP on the supervised learning tasks
>
> | MLP | Random CV | Contiguous CV |
> | --- | --- | --- |
> | Spearman correlation | 0.327 | 0.070 |
>
>
>
> ## About Question 5
>
> Q: The paper correctly identifies that DMS (local mutations on a known scaffold) and SELEX (discovery of functional sequences from a random pool) are fundamentally different tasks. Indeed, your results show that model performance is 'significantly' different between them, with zero-shot performance on SELEX being near-random (Lines 424-428). Given this profound difference, why are results from both tasks aggregated into single "overall performance" metrics and rankings (e.g., Figure 2a)? Doesn't this aggregation obscure more than it reveals, by averaging performance over tasks that test completely different scientific capabilities (interpolation vs. extrapolation; local vs. global search)?
>
>
> A: We must respectfully point out a **misunderstanding** regarding the data presentation in our manuscript. The Reviewer states that we "aggregated results from both tasks into single metrics" citing Figure 2a, but this is factually incorrect. As explicitly stated in the figure caption and title (**"Figure 2a: Overall Performance Distribution of all models on DMS datasets"**), this figure exclusively aggregates DMS tasks and contains no SELEX data. We fully agree that DMS and SELEX represent fundamentally different scientific capabilities, which is precisely why we strictly separated their evaluations throughout the paper: SELEX results are reported independently in **Appendix Table 12 ("Overall Scores of all models on SELEX tasks")**. Therefore, the concern that our metrics "obscure" differences is based on a misreading, as we have maintained a rigorous separation to accurately reflect the profound differences between local (DMS) and global (SELEX) search tasks.
>
>
> ### Reference
>
> [1] Notin, Pascal, et al. "Proteingym: Large-scale benchmarks for protein fitness prediction and design." Advances in Neural Information Processing Systems 36 (2023): 64331-64379.
>
>
> [2] Arora, Rohit, et al. "RNAGym: Benchmarks for RNA Fitness and Structure Prediction." ICLR 2025 Workshop on Generative and Experimental Perspectives for Biomolecular Design.

---

### Author Response · Authors · 2025-11-27
**General Response: Summary of Updates and Gratitude to Reviewers**

Dear Reviewers,

We sincerely thank you for your time and the constructive feedback provided. Your insightful comments have been invaluable in helping us identify areas for improvement and refine the quality of our manuscript. We have posted detailed responses to each reviewer's specific questions. Below, we summarize the major updates and clarifications made during the rebuttal period, which address the common concerns raised:

1. **Clarification on Zero-Shot Evaluation Protocol.** We have provided a detailed rationale for our zero-shot evaluation setting. We clarify how our strict processing strategy ensures a fair assessment of the foundation models' intrinsic capabilities.

2. **New Analysis on Transfer Learning.** To demonstrate the models' generalization abilities, we have added comprehensive analyses in transfer learning scenarios, specifically focusing on transfer tasks from DNA to RNA and from DMS to SELEX datasets.

3. **Detailed Experiments on Probe Selection.** For the supervised learning benchmarks, we have conducted extensive experiments to verify the impact of different probe architectures. Detailed explanations and comparisons have been added to justify our choices.

4. **In-depth Discussion on DMS and SELEX Tasks.** We have significantly expanded the discussion regarding the performance gap between DMS and SELEX tasks. As detailed in our specific responses , we analyze this through the lens of "Local Perturbation" (DMS) vs. "Global Search" (SELEX) to explain the fundamental differences in task difficulty and model behavior.

In addition to the points above, we have provided detailed analyses and responses to the specific questions raised by each reviewer. We have integrated these discussions and the new experimental results into the revised manuscript.

We believe that these revisions have significantly improved the quality of the paper, and we look forward to further exchanges with you.

Sincerely,

The Authors

---

### Author Response · Authors · 2025-12-03
**Final Response and Rebuttal Summary**

Dear Area Chair,

We sincerely appreciate your time and effort in handling our submission. To assist your decision-making, we provide a concise summary of the **substantial experimental updates** and **new scientific insights** achieved during the rebuttal period, which address the primary constructive feedback from the reviewers.

**1. Major Experimental Expansion: Transfer Learning & New Baselines**

Addressing the requests from Reviewers EUvA and rEPG for deeper analysis and stronger comparisons, we have significantly expanded the benchmark's scope:

* **Comprehensive Transfer Learning Analysis:** We conducted extensive Out-of-Distribution (OOD) experiments to probe the boundaries of foundation models.
    * **Cross-Assay:** We revealed a significant negative transfer correlation (-0.51) when transferring from SELEX to DMS tasks. This provides a critical insight: features learned for "binding affinity" (SELEX) are often orthogonal to those for "evolutionary fitness" (DMS).
    * **Cross-Family:** We mapped transferability across 5 nucleic acid categories. We identified strong positive transfer between mRNA and Ribozymes (r $\approx$ 0.59), while highlighting challenges in transferring to specialized families like tRNA.
* **Supervised Learning Baselines:** We implemented a "Train-from-Scratch" MLP baseline using One-hot encoding. Results confirm that pre-trained Foundation Models significantly outperform this baseline, particularly in Contiguous Cross-Validation settings, validating the necessity of pre-training for generalization.

**2. Deep Scientific Insight: The "SELEX vs. DMS" Distinction**

Reviewers (TTVY, qYsy, rEPG) noted the performance gap between DMS and SELEX tasks. We have synthesized these results into a key scientific conclusion of the benchmark:

* **Local vs. Global Search:** We demonstrate that current FMs excel at *Local Perturbation* (DMS) due to high sequence homology with the wild-type but struggle with *Global Search* (SELEX) in zero-shot settings.
* **Quantifying the Gap:** NABench rigorously quantifies this limitation, proving that aligning "evolutionary likelihood" with "arbitrary binding affinity" without a reference anchor remains an open challenge. This establishes a clear metric for future structure-aware or inverse-folding models to aim for.

**3. Enhanced Rigor: Architecture Ablations & Data Quality**

We have resolved specific technical concerns raised by Reviewers TTVY and qYsy to ensure maximum reproducibility:
* **Optimality of Linear Probe:** Addressing Reviewer TTVY’s concern regarding the use of Ridge regression, we performed a head-to-head comparison against other probes (including MLP, Random Forest, and XGBoost). Ridge regression consistently outperformed complex heads (Mean Correlation: 0.617 for Ridge vs. 0.519 for MLP). This validates that FM embeddings are linearly separable and that robust linear probes are preferable to prevent overfitting in regime-limited experimental data.
* **Layer-wise Ablation for Evo Models:** Addressing concerns on embedding extraction, we conducted ablation studies on StripedHyena architectures. Results show the performance gap between intermediate and final layers is statistically marginal ($<0.01$ Spearman), validating our standardized evaluation protocol.
* **Taxonomic Breakdown:** We provided a detailed species breakdown (Eukaryotic, Prokaryotic, Synthetic) and analyzed performance across these domains, confirming the benchmark's broad biological coverage.
* **Data Integrity:** We clarified the metadata naming conventions (e.g., for the Kolm et al. dataset) to resolve retrieval confusion and verified the sequence processing pipelines, confirming the reliability and completeness of our data curation.

**Conclusion**

With 29 models evaluated across 162 assays, and now bolstered by expanded transfer learning protocols and rigorous baselines, we believe NABench stands as the most comprehensive and reproducible benchmark in this domain. We hope these updates clarify the paper's contribution and robustness.

---

### Meta-Review · Area_Chair_mcnC · 2026-01-05

**Summary:**

The primary concerns influencing the decision center on the validity of the benchmark's experimental design and the scientific utility of the SELEX subset. Multiple reviewers (TTvY, rEPG, qYsy) flagged the near-random performance of all models on zero-shot SELEX tasks, questioning whether this subset effectively evaluates current model capabilities or represents an ill-posed problem. Furthermore, there was a consensus criticism regarding the oversimplification of evaluation protocols; reviewers argued that relying exclusively on Ridge regression for supervised probing and mean-pooling for zero-shot inference fails to capture the full potential of complex foundation models compared to fine-tuning or logit-based methods. These methodological reservations, combined with specific data curation errors identified by Reviewer qYsy and doubts about the rigor of the transfer learning analysis, collectively undermined confidence in the benchmark’s robustness despite its impressive scale.

**Reviewer Concerns:**

The rebuttal remediated the majority of the concrete technical and factual deficiencies identified in the initial reviews. However, the fundamental dispute regarding the SELEX subset remains the most significant outstanding concern. While the authors theoretically reframed the models' near-random performance as a scientific insight distinguishing local versus global search, this argument effectively sidesteps the reviewers' (particularly **TTvY** and **qYsy**) core apprehension: that the current zero-shot protocols and arbitrary evaluation thresholds may be ill-suited for these tasks.

**Reviewer Scores:**

Based on the trajectory of the discussion and the specific interventions made, the final score distribution would likely settle at 2, 4, 6, 6. Reviewer EUvA, qYsy, and rEPG would almost certainly maintain their score, and Reviewer TTvY had lowered the score from 8 to 6.

---

### Decision · Program_Chairs · 2026-01-26

Reject